# Unconditional stability of a recurrent neural circuit implementing divisive normalization

**Shivang Rawat**[1,2]     **David J. Heeger**[3,4]     **Stefano Martiniani**[1,2,5]

[1] Courant Institute of Mathematical Sciences, NYU
[2] Center for Soft Matter Research, Department of Physics, NYU
[3] Department of Psychology, NYU     [4] Center for Neural Science, NYU
[5] Simons Center for Computational Physical Chemistry, Department of Chemistry, NYU

`{sr6364, david.heeger, sm7683}@nyu.edu`

## Abstract

Stability in recurrent neural models poses a significant challenge, particularly in developing biologically plausible neurodynamical models that can be seamlessly trained. Traditional cortical circuit models are notoriously difficult to train due to expansive nonlinearities in the dynamical system, leading to an optimization problem with nonlinear stability constraints that are difficult to impose. Conversely, recurrent neural networks (RNNs) excel in tasks involving sequential data but lack biological plausibility and interpretability. In this work, we address these challenges by linking dynamic divisive normalization (DN) to the stability of "oscillatory recurrent gated neural integrator circuits" (ORGaNICs), a biologically plausible recurrent cortical circuit model that dynamically achieves DN and that has been shown to simulate a wide range of neurophysiological phenomena. By using the indirect method of Lyapunov, we prove the remarkable property of unconditional local stability for an arbitrary-dimensional ORGaNICs circuit when the recurrent weight matrix is the identity. We thus connect ORGaNICs to a system of coupled damped harmonic oscillators, which enables us to derive the circuit's energy function, providing a normative principle of what the circuit, and individual neurons, aim to accomplish. Further, for a generic recurrent weight matrix, we prove the stability of the 2D model and demonstrate empirically that stability holds in higher dimensions. Finally, we show that ORGaNICs can be trained by backpropagation through time without gradient clipping/scaling, thanks to its intrinsic stability property and adaptive time constants, which address the problems of exploding, vanishing, and oscillating gradients. By evaluating the model's performance on RNN benchmarks, we find that ORGaNICs outperform alternative neurodynamical models on static image classification tasks and perform comparably to LSTMs on sequential tasks.

## 1   Introduction

Deep neural networks (DNNs) have found widespread use in modeling tasks from experimental systems neuroscience. The allure of DNN-based models lies in their ease of training and the flexibility they offer in architecting systems with desired properties [1–3]. In contrast, neurodynamical models like the Wilson-Cowan [4] or the Stabilized Supralinear Network (SSN) [5] are more biologically plausible than DNNs, but these models confront considerable training challenges due to the lack of stability guarantees for high-dimensional problems. Training recurrent neural networks (RNNs), by comparison, is more straightforward thanks to ad hoc regularization techniques like layer nor-

malization, batch normalization, and gradient clipping/scaling, which help stabilize training without imposing strict stability constraints. Conversely, neurodynamical models require enforcing hard stability constraints while maintaining biological plausibility. In lower dimensions, it is relatively straightforward to derive constraints on model parameters that ensure a dynamically stable system [6, 7]. However, for high-dimensional systems, this becomes significantly more challenging, as integrating these hard constraints into the optimization problem is more complex [8, 9]. Stability is generally advantageous in DNNs, as it is linked to improved generalization, mitigation of exploding gradient problems, increased robustness to input noise, and simplified training techniques [10].

The divisive normalization (DN) model was developed to explain the responses of neurons in the primary visual cortex (V1) [11–14], and has since been applied to diverse cognitive processes and neural systems [15–24]. Therefore, DN has been proposed as a canonical neural computation [25] that is linked to many well-documented physiological [26, 27] and psychophysical [28, 29] phenomena. DN models various neural processes: adaptation [30, 31], attention [32], automatic gain control [33], decorrelation, and statistical whitening [34]. The defining characteristic of DN is that each neuron's response is divided by a weighted sum of the activity of a pool of neurons (Eq. 2, below) like when normalizing the length of a vector. Due to its wide applicability and ability to explain a variety of neurophysiological phenomena, we argue that this characteristic should be central to any neurodynamical model. Both the Wilson-Cowan and SSN models have been shown to approximate DN responses [5, 35], but only approximately in certain parameter regimes.

Normalization techniques have been extensively adopted for training DNNs, demonstrating their ability to stabilize, accelerate training, and enhance generalization [36–38]. Divisive normalization can be viewed as a comprehensive normalization strategy, with batch and layer normalization being specific instances [39]. Models implementing DN have shown superior performance compared to common normalization methods (Batch [36], Layer [37], Group [40]) in tasks such as image recognition with convolutional neural networks (CNNs) [41] and language modeling with RNNs [39, 42]. Despite the foundational role of these techniques in deep learning algorithms, their implementation is ad hoc, limiting their conceptual relevance. They serve as practical solutions addressing the limitations of current machine learning frameworks rather than offering principled insights derived from understanding cortical circuits.

It has been proposed that DN is achieved via a recurrent circuit [11, 13, 43–47]. Oscillatory recurrent gated neural integrator circuits (ORGaNICs) are rate-based recurrent neural circuit models that implement DN dynamically via recurrent amplification [47, 48]. Since ORGaNICs' response follows the DN equation at steady-state, its steady-state response captures the full range of aforementioned neural phenomena explained by DN [11–34]. ORGaNICs have further been shown to simulate key time-dependent neurophysiological and cognitive/perceptual phenomena under realistic biophysical constraints [47, 48]. Additional phenomena not explained by DN [49] can in principle be integrated into the model. In this paper, however, we focus on the effects of DN on the dynamical stability of ORGaNICs. Despite some empirical evidence that ORGaNICs are highly robust, the question of whether the model is stable for arbitrary parameter choices, and thus whether it can be robustly trained on ML tasks by backpropagation-through-time (BPTT), remains open.

Here, we establish the unconditional stability — applicable across all parameters and inputs — of a multidimensional two-neuron-types ORGaNICs model when the recurrent weight matrix is the identity. We prove this result, detailed in Section 4, by the indirect method of Lyapunov: we perform linear stability analysis around the model's analytically-known normalization fixed point and reduce the stability problem to that of a high-dimensional mechanical system, whose stability is defined in terms of a tractable quadratic eigenvalue problem. We then address the stability of the model with an arbitrary recurrent weight matrix in Section 5. While the indirect method of Lyapunov becomes intractable for such a system, we provide proof of unconditional stability for a two-dimensional circuit with an arbitrary recurrent weight and offer empirical evidence supporting the claim of stability for high-dimensional systems.

ORGaNICs can be viewed as biophysically plausible extensions of Long Short Term Memory units (LSTMs) [3] and Gated Recurrent Units (GRUs) [50], RNN architectures that have been widely used in ML applications [3, 51–54]. The main differences are that ORGaNICs operate in continuous time and have built-in dynamic normalization (via recurrent gain modulation) and built-in attention (via input gain modulation). Thus, we expect that ORGaNICs should be able to solve relatively sophisticated tasks [47]. Here, we demonstrate (Section 6) that by virtue of their intrinsic

stability, ORGaNICs can be trained on sequence modeling tasks by BPTT, in the same manner as traditional RNNs (unlike SSN that instead requires costly specialized training strategies [55]), despite implementing power-law activations [5]. Moreover, we show that ORGaNICs trained by naive BPTT (i.e., without gradient clipping/scaling or other ad hoc strategies) achieve performance comparable to LSTMs on the tasks that we consider, despite no systematic hyperparameter tuning.

## 2 Related Work

**Trainable biologically plausible neurodynamical models**: There have been several attempts to develop neurodynamical models that mimic the function of biological circuits and that can be trained on cognitive tasks. Song et al. [56] incorporated Dale's law into the vanilla RNN architecture, which was successfully trained across a variety of cognitive tasks. Building on this, Soo et al. [57] developed a technique for such RNNs to learn long-term dependencies by using skip connections through time. ORGaNICs is a model that is already built on biological principles and can learn long-term dependencies intrinsically by tuning the (intrinsic or effective) time constants, therefore it does not require the method used in [57]. Soo et al. [55] introduced a novel training methodology (dynamics-neural growth) for SSNs and demonstrated its utility for tasks involving static (time-independent) stimuli. However, this training approach is costly and difficult to scale (because SSNs, unlike ORGaNICs, are not unconditionally stable), and its applicability on tasks with dynamically changing inputs remains unclear.

**Dynamical systems view of RNNs:** The stability of continuous-time RNNs has been extensively studied and discussed in a comprehensive review by Zhang et al. [58]. Recent advancements have focused on designing architectures that address the issues of vanishing and exploding gradients, thereby enhancing trainability and performance. A central idea in these designs is to achieve better trainability and generalization by ensuring the dynamical stability of the network. Moreover to avoid the problem of vanishing gradients the key idea is to constrain the real part of the eigenvalues of the linearized dynamical system to be close to zero, which facilitates the propagation and retention of information over long durations of time. Chang et al. [59] and Erichson et al. [60] achieve this by imposing an antisymmetric constraint on the recurrent weight matrix. Meanwhile, Rusch et al. [61, 62] propose an architecture based on coupled damped harmonic oscillators, resulting in a second-order system of ordinary differential equations that behaves similarly to how ORGaNICs behave in the vicinity of the normalization fixed point, as we show in Section 4. Despite their impressive performance on various sequential data benchmarks, these models lack biological plausibility due to their use of saturating nonlinearities (instead of normalization) and unrealistic weight parameterizations.

## 3 Model description

In its simplest form, the two-neuron-types ORGaNICs model [47, 48] with $n$ neurons of each type can be written as,

$$\boldsymbol{\tau}_y \odot \dot{\mathbf{y}} = -\mathbf{y} + \mathbf{b} \odot \mathbf{z} + \left(\mathbf{1} - \mathbf{a}^+\right) \odot \left(\mathbf{W}_r \left(\sqrt{\mathbf{y}^+} - \sqrt{\mathbf{y}^-}\right)\right)$$
$$\boldsymbol{\tau}_a \odot \dot{\mathbf{a}} = -\mathbf{a} + \mathbf{b}_0^2 \odot \boldsymbol{\sigma}^2 + \mathbf{W} \left(\left(\mathbf{y}^+ + \mathbf{y}^-\right) \odot \mathbf{a}^{+2}\right)$$
(1)

where $\mathbf{y} \in \mathbb{R}^n$ and $\mathbf{a} \in \mathbb{R}^n$ are the membrane potentials (relative to an arbitrary threshold potential that we take to be 0) of the excitatory ($\mathbf{y}$) and inhibitory ($\mathbf{a}$) neurons, evolving according to the dynamical equations defined above with $\dot{\mathbf{y}}$ and $\dot{\mathbf{a}}$ denoting the time derivatives. The notation $\odot$ denotes element-wise multiplication of vectors, and squaring, rectification, square-root, and division are also performed element-wise. $\mathbf{1}$ is an n-dimensional vector with all entries equal to 1. $\mathbf{z} \in \mathbb{R}^n$ is the input drive to the circuit and is a weighted sum of the input, $\mathbf{x} \in \mathbb{R}^m$, i.e., $\mathbf{z} = \mathbf{W}_{zx}\mathbf{x}$. The firing rates, $\mathbf{y}^\pm = \lfloor \pm\mathbf{y} \rfloor^2$ and $\mathbf{a}^+ = \sqrt{\lfloor \mathbf{a} \rfloor}$ are rectified ($\lfloor . \rfloor$) power functions of the underlying membrane potentials. For the derivation of a general model with arbitrary power-law exponents, including the Eq. 1, see Appendix A. Note that the term $\sqrt{\mathbf{y}^+} - \sqrt{\mathbf{y}^-}$ serves the purpose of defining a mechanism for reconstructing the membrane potential (which can be negative, depending on the sign of the input) from the firing rates $\mathbf{y}^\pm$ that are strictly nonnegative. $\mathbf{y}^+$ and $\mathbf{y}^-$ are the firing rates of neurons with complementary receptive fields such that they encode inputs with positive and negative signs, respectively. Note that only one of these neurons fires at a given time. In ORGaNICs, these neurons have a single dynamical equation for their membrane potentials, where the sign of $\mathbf{y}$

indicates which neuron is active. Neurons with such complementary (anti-phase) receptive fields are found adjacent to each other in the visual cortex [63], and we hypothesize that such complementary neurons are ubiquitous throughout the neocortex. $\mathbf{b} \in \mathbb{R}_*^{+n}$ and $\mathbf{b}_0 \in \mathbb{R}_*^{+n}$ are the input gains for the external inputs $\mathbf{z}$ and $\boldsymbol{\sigma}$ fed to neurons $\mathbf{y}$ and $\mathbf{a}$, respectively. $\mathbb{R}_*^+$ is the set of positive real numbers, $\{x \in \mathbb{R} \mid x > 0\}$. $\boldsymbol{\sigma} \in \mathbb{R}_*^{+n}$ determines the semisaturation of the responses of neurons $\mathbf{y}$ by contributing to the depolarization of neurons $\mathbf{a}$. $\boldsymbol{\tau}_y \in \mathbb{R}_*^{+n}$ and $\boldsymbol{\tau}_a \in \mathbb{R}_*^{+n}$ represent the time constants of $\mathbf{y}$ and $\mathbf{a}$ neurons.

In addition to receiving external inputs, both $\mathbf{y}$ and $\mathbf{a}$ neurons receive recurrent inputs, represented by the last term in both of the equations. $\mathbf{W}_r \in \mathbb{R}^{n \times n}$ is the recurrent weight matrix that captures lateral connections between the $\mathbf{y}$ neurons. This recurrent input is gated by the $\mathbf{a}$ neurons, via the term $(\mathbf{1} - \mathbf{a}^+)$. Similarly, the *nonnegative* normalization weight matrix, $\mathbf{W} \in \mathbb{R}_*^{n \times n}$, encapsulates the recurrent inputs received by the $\mathbf{a}$ neurons. The differential equations are designed in such a way that when $\mathbf{W}_r = \mathbf{I}$ and $\mathbf{b} = \mathbf{b}_0 = b_0\mathbf{1}$ (i.e., with all elements equal to a constant $b_0$), the principal neurons follow the normalization equation exactly (and approximately when $\mathbf{W}_r \neq \mathbf{I}$) at steady-state,

$$\mathbf{y}_s^+ \equiv \lfloor \mathbf{y}_s \rfloor^2 = \frac{\lfloor \mathbf{z} \rfloor^2}{\boldsymbol{\sigma}^2 + \mathbf{W}\left(\lfloor \mathbf{z} \rfloor^2 + \lfloor -\mathbf{z} \rfloor^2\right)}. \tag{2}$$

$\lfloor \mathbf{z} \rfloor^2$ and $\lfloor -\mathbf{z} \rfloor^2$ represent the contribution of neurons with complementary receptive fields to the normalization pool, and $\lfloor \mathbf{z} \rfloor^2 + \lfloor -\mathbf{z} \rfloor^2 = \mathbf{z}^2$ is the contrast energy of the input. Note that the recurrent gain, $(\mathbf{1} - \mathbf{a}^+)$, is a particular nonlinear function of the output responses/activation designed to achieve DN, while the input gain, $\mathbf{b}^+$, is an input gate that can implement an attention mechanism.

## 4 Stability analysis of high-dimensional model with identity recurrent weights

We consider the stability of the general high-dimensional ORGaNICs (Eq. 1) when the recurrent weight matrix is identity, $\mathbf{W}_r = \mathbf{I}$. We first simplify the dynamical system by noting that $\sqrt{\mathbf{y}^+} - \sqrt{\mathbf{y}^-} = \mathbf{y}$ and $\mathbf{y}^+ + \mathbf{y}^- = \mathbf{y}^2$ yielding the following equations,

$$\begin{aligned}
\boldsymbol{\tau}_y \odot \dot{\mathbf{y}} &= -\sqrt{\lfloor \mathbf{a} \rfloor} \odot \mathbf{y} + \mathbf{b} \odot \mathbf{z} \\
\boldsymbol{\tau}_a \odot \dot{\mathbf{a}} &= -\mathbf{a} + \mathbf{b}_0^2 \odot \boldsymbol{\sigma}^2 + \mathbf{W}\left(\mathbf{y}^2 \odot \lfloor \mathbf{a} \rfloor\right)
\end{aligned} \tag{3}$$

For identity recurrent weights, we have a unique fixed point, given by,

$$\mathbf{y}_s = \frac{\mathbf{b} \odot \mathbf{z}}{\sqrt{\mathbf{b}_0^2 \odot \boldsymbol{\sigma}^2 + \mathbf{W}\left(\mathbf{b}^2 \odot \mathbf{z}^2\right)}}; \quad \mathbf{a}_s = \mathbf{b}_0^2 \odot \boldsymbol{\sigma}^2 + \mathbf{W}\left(\mathbf{b}^2 \odot \mathbf{z}^2\right) \tag{4}$$

Since the normalization weights in the matrix $\mathbf{W}$ are nonnegative, at steady-state we have $\mathbf{a}_s > \mathbf{0}$, so that $\sqrt{\lfloor \mathbf{a}_s \rfloor} = \sqrt{\mathbf{a}_s}$, and the corresponding firing rates at steady-state are,

$$\mathbf{y}_s^\pm = \frac{\lfloor \pm \mathbf{b} \odot \mathbf{z} \rfloor^2}{\mathbf{b}_0^2 \odot \boldsymbol{\sigma}^2 + \mathbf{W}\left(\mathbf{b}^2 \odot \mathbf{z}^2\right)}; \quad \mathbf{a}_s^+ = \sqrt{\mathbf{b}_0^2 \odot \boldsymbol{\sigma}^2 + \mathbf{W}\left(\mathbf{b}^2 \odot \mathbf{z}^2\right)} \tag{5}$$

Note that we recover the normalization equation, Eq. 2, if $\mathbf{b} = \mathbf{b}_0 = b_0\mathbf{1}$. Since the fixed points of $\mathbf{y}$ and $\mathbf{a}$ neurons are known analytically, to prove that this fixed point is *locally asymptotically stable* (i.e., the responses converge asymptotically to the fixed point), we apply the *indirect method of Lyapunov* at this fixed point [64]. This method allows us to analyze the stability of the nonlinear system in the vicinity of the fixed point by studying the corresponding linearized system. The Jacobian matrix $\mathbf{J} \in \mathbb{R}^{2n \times 2n}$ about $(\mathbf{y}_s, \mathbf{a}_s)$, defining the linearized system, is given by,

$$\mathbf{J} = \begin{bmatrix} -\mathbf{D}\left(\frac{\sqrt{\mathbf{a}_s}}{\boldsymbol{\tau}_y}\right) & -\mathbf{D}\left(\frac{\mathbf{y}_s}{2 \odot \sqrt{\mathbf{a}_s} \odot \boldsymbol{\tau}_y}\right) \\ \mathbf{D}\left(\frac{2}{\boldsymbol{\tau}_a}\right)\mathbf{W}\,\mathbf{D}\left(\mathbf{a}_s \odot \mathbf{y}_s\right) & \mathbf{D}\left(\frac{1}{\boldsymbol{\tau}_a}\right)\left(-\mathbf{I} + \mathbf{W}\,\mathbf{D}\left(\mathbf{y}_s^2\right)\right) \end{bmatrix} \tag{6}$$

where $\mathbf{D}(\mathbf{x})$ is a diagonal matrix of appropriate size with the elements of the vector $\mathbf{x}$ on the diagonal. A necessary and sufficient condition for local stability is that the real parts of all eigenvalues of this matrix are negative. We thus proceed by computing the characteristic polynomial for the Jacobian, $p_{\mathbf{J}}(\lambda) \equiv \det(\mathbf{J} - \lambda\mathbf{I})$. The roots of this polynomial, found by setting $p_{\mathbf{J}}(\lambda) = 0$, are the eigenvalues of the system. Consider the block matrix,

$$\mathbf{J} - \lambda\mathbf{I} = \begin{bmatrix} \mathbf{A}_{11} & \mathbf{A}_{12} \\ \mathbf{A}_{21} & \mathbf{A}_{22} \end{bmatrix} = \begin{bmatrix} -\mathbf{D}\left(\frac{\sqrt{\mathbf{a}_s}}{\boldsymbol{\tau}_y}\right) - \lambda\mathbf{I} & -\mathbf{D}\left(\frac{\mathbf{y}_s}{2 \odot \sqrt{\mathbf{a}_s} \odot \boldsymbol{\tau}_y}\right) \\ \mathbf{D}\left(\frac{2}{\boldsymbol{\tau}_a}\right)\mathbf{W}\,\mathbf{D}\left(\mathbf{a}_s \odot \mathbf{y}_s\right) & \mathbf{D}\left(\frac{1}{\boldsymbol{\tau}_a}\right)\left(-\mathbf{I} + \mathbf{W}\,\mathbf{D}\left(\mathbf{y}_s^2\right)\right) - \lambda\mathbf{I} \end{bmatrix} \tag{7}$$

Notice that $\mathbf{A}_{11}$ and $\mathbf{A}_{12}$ are diagonal and therefore they commute, i.e., $\mathbf{A}_{11}\mathbf{A}_{12} = \mathbf{A}_{12}\mathbf{A}_{11}$, so we have that $\det(\mathbf{J} - \lambda\mathbf{I}) = \det(\mathbf{A}_{22}\mathbf{A}_{11} - \mathbf{A}_{21}\mathbf{A}_{12})$ which is a property of the determinant of block matrices with commuting blocks [65]. Therefore, the characteristic polynomial of the linearized system after expansion of the terms and simplification is given by,

$$\det(\mathbf{J} - \lambda\mathbf{I}) = \det\left(\lambda^2\mathbf{I} + \lambda\left[\mathbf{D}\left(\frac{1}{\boldsymbol{\tau}_a}\right) + \mathbf{D}\left(\frac{\sqrt{\mathbf{a}_s}}{\boldsymbol{\tau}_y}\right) - \mathbf{D}\left(\frac{1}{\boldsymbol{\tau}_a}\right)\mathbf{W}\mathbf{D}\left(\mathbf{y}_s^2\right)\right] + \mathbf{D}\left(\frac{\sqrt{\mathbf{a}_s}}{\boldsymbol{\tau}_y \odot \boldsymbol{\tau}_a}\right)\right)$$

(8)

Finding the roots of this polynomial is thus a quadratic eigenvalue problem of the form $\mathcal{L}(\lambda) \equiv \det(\lambda^2\mathbf{I} + \lambda\mathbf{B} + \mathbf{K}) = 0$, which has been studied extensively [66–69]. $\mathcal{L}(\lambda)$ can be interpreted as the characteristic polynomial associated with a system of linear second-order differential equations with constant coefficients of the form $\mathbf{I}\ddot{\mathbf{x}} + \mathbf{B}\dot{\mathbf{x}} + \mathbf{K}\mathbf{x} = \mathbf{0}$. Therefore, proving the stability of our system (i.e., $\mathbf{Re}(\lambda) < 0$ for $\{\lambda : \mathcal{L}(\lambda) = 0\}$), is equivalent to proving the asymptotic stability of $\mathbf{I}\ddot{\mathbf{x}} + \mathbf{B}\dot{\mathbf{x}} + \mathbf{K}\mathbf{x} = \mathbf{0}$.

Tisseur et al. [67] and Kirillov et al. [69] list a set of constraints on the damping matrix, $\mathbf{B}$, and stiffness matrix, $\mathbf{K}$, that yield a stable system, but they are *not* directly applicable to our system. In the context of a high-dimensional mechanical system, our system falls under the category of *gyroscopically stabilized systems with indefinite damping*. Few results are known about the conditions leading to the stability of such systems. By constructing a Lyapunov function, we prove (Appendix B) the following stability theorem that is directly applicable to our system, following an approach similar to Kliem et al. [70].

**Theorem 4.1.** *For a system of linear differential equations with constant coefficients of the form,*

$$\mathbf{I}\ddot{\mathbf{x}} + \mathbf{B}\dot{\mathbf{x}} + \mathbf{K}\mathbf{x} = \mathbf{0}$$

(9)

*where $\mathbf{B} \in \mathbb{R}^{n \times n}$ and $\mathbf{K} \in \mathbb{R}^{n \times n}$ is a positive diagonal matrix (hence $\mathbf{K} \succ 0$), the dynamical system is globally asymptotically stable if $\mathbf{B}$ is Lyapunov diagonally stable.*

Since the stiffness matrix,

$$\mathbf{K} = \mathbf{D}\left(\frac{\sqrt{\mathbf{a}_s}}{\boldsymbol{\tau}_y \odot \boldsymbol{\tau}_a}\right) = \mathbf{D}\left(\frac{\sqrt{\mathbf{b}_0^2 \odot \boldsymbol{\sigma}^2 + \mathbf{W}\left(\mathbf{b}^2 \odot \mathbf{z}^2\right)}}{\boldsymbol{\tau}_y \odot \boldsymbol{\tau}_a}\right)$$

(10)

is a positive diagonal matrix, a sufficient condition for stability of the system is that the damping matrix, $\mathbf{B}$, given by,

$$
\begin{aligned}
\mathbf{B} =& \mathbf{B}_1 + \mathbf{B}_2 - \mathbf{B}_3 \\
=& \mathbf{D}\left(\frac{1}{\boldsymbol{\tau}_a}\right) + \mathbf{D}\left(\frac{\sqrt{\mathbf{a}_s}}{\boldsymbol{\tau}_y}\right) - \mathbf{D}\left(\frac{1}{\boldsymbol{\tau}_a}\right)\mathbf{W}\mathbf{D}\left(\mathbf{y}_s^2\right) \\
=& \mathbf{D}\left(\frac{1}{\boldsymbol{\tau}_a}\right) + \mathbf{D}\left(\frac{\sqrt{\mathbf{b}_0^2 \odot \boldsymbol{\sigma}^2 + \mathbf{W}\left(\mathbf{b}^2 \odot \mathbf{z}^2\right)}}{\boldsymbol{\tau}_y}\right) - \mathbf{D}\left(\frac{1}{\boldsymbol{\tau}_a}\right)\mathbf{W}\mathbf{D}\left(\frac{\mathbf{b}^2 \odot \mathbf{z}^2}{\mathbf{b}_0^2 \odot \boldsymbol{\sigma}^2 + \mathbf{W}\left(\mathbf{b}^2 \odot \mathbf{z}^2\right)}\right)
\end{aligned}
$$

(11)

is *Lyapunov diagonally stable*, i.e., there exists a positive definite diagonal matrix $\mathbf{T}$, such that $\mathbf{T}\mathbf{B} + \mathbf{B}^\top\mathbf{T}$ is positive definite.

Since all of the parameters are positive, and the weights in the matrix $\mathbf{W}$ are nonnegative, we can conclude the following: $\mathbf{B}_1$ and $\mathbf{B}_2$ are positive diagonal matrices and $\mathbf{B}_3$ is a matrix with all positive entries (that may or may not be symmetric). Therefore, $\mathbf{B}$ is a *Z-matrix*, meaning that its off-diagonal entries are nonpositive. Further, a *Z-matrix* is *Lyapunov diagonally stable* if and only if it is a nonsingular *M-matrix*. Intuitively, *M-matrices* are matrices with non-positive off-diagonal elements and "large enough" positive diagonal entries. Berman & Plemmons [71] list 50 equivalent definitions of nonsingular *M-matrices*. We use the one that is best suited for our problem,

**Theorem 4.2.** *(Chapter 6, Theorem 2.3 from [71]) A Z-matrix matrix $\mathbf{B} \in \mathbb{R}^{n \times n}$ is Lyapunov diagonally stable if and only if there exists a convergent regular splitting of the matrix, that is, it has a representation of the form $\mathbf{B} = \mathbf{M} - \mathbf{N}$, where $\mathbf{M}^{-1}$ and $\mathbf{N}$ have all nonnegative entries, and $\mathbf{M}^{-1}\mathbf{N}$ has a spectral radius smaller than 1.*

We now show that, indeed, $\mathbf{B}$ has a *convergent regular splitting* for all combinations of the circuit parameters and for all inputs. We have already shown that $\mathbf{B}$ is a *Z-matrix*, therefore, the first

condition of the theorem is satisfied. Next, we consider the following splitting $\mathbf{B} = \mathbf{M} - \mathbf{N}$ with $\mathbf{M} = \mathbf{B}_1 + \mathbf{B}_2$ and $\mathbf{N} = \mathbf{B}_3$. Since $\mathbf{B}_1$ and $\mathbf{B}_2$ are positive diagonal matrices, $\mathbf{M}^{-1}$ is nonnegative, while $\mathbf{N}$ is also nonnegative because $\mathbf{B}_3$ has all positive entries. Therefore, the only condition left to satisfy is that the spectral radius of $\mathbf{M}^{-1}\mathbf{N}$ is smaller than 1, or that the matrix is convergent.

The matrix $\mathbf{S} = \mathbf{M}^{-1}\mathbf{N} = (\mathbf{B}_1 + \mathbf{B}_2)^{-1}\mathbf{B}_3$ can be written as,

$$\mathbf{S} = \mathbf{D}\left(\frac{1}{1 + (\boldsymbol{\tau}_a/\boldsymbol{\tau}_y) \odot \sqrt{\mathbf{b}_0^2 \odot \boldsymbol{\sigma}^2 + \mathbf{W}\,(\mathbf{b}^2 \odot \mathbf{z}^2)}}\right) \mathbf{W}\mathbf{D}\left(\frac{\mathbf{b}^2 \odot \mathbf{z}^2}{\mathbf{b}_0^2 \odot \boldsymbol{\sigma}^2 + \mathbf{W}\,(\mathbf{b}^2 \odot \mathbf{z}^2)}\right) \quad (12)$$

We prove the following theorem (Appendix D) which directly applies to $\mathbf{S}$,

**Theorem 4.3.** *A matrix $\mathbf{A}$ of the form $\mathbf{A} = \mathbf{D}(\mathbf{t})\,\mathbf{W}\,\mathbf{D}\,(\mathbf{u}/\,(\mathbf{v} + \mathbf{W}\mathbf{u}))$ is convergent (i.e., its spectral radius is less than 1), if $\mathbf{W} \in \mathbb{R}^{n \times n}$ and $\mathbf{t}, \mathbf{u}, \mathbf{v} \in \mathbb{R}^n$ satisfy $0 < t_i < 1$, $u_i \geq 0$, $v_i > 0$ and $w_{ij} \geq 0$ for all $i, j$.*

Defining $\mathbf{t} \to \mathbf{1}/(1 + (\boldsymbol{\tau}_a/\boldsymbol{\tau}_y) \odot \sqrt{\mathbf{b}_0^2 \odot \boldsymbol{\sigma}^2 + \mathbf{W}\,(\mathbf{b}^2 \odot \mathbf{z}^2)})$, $\mathbf{u} \to \mathbf{b}^2 \odot \mathbf{z}^2$ and $\mathbf{v} \to \mathbf{b}_0^2 \odot \boldsymbol{\sigma}^2$, it can be seen that they satisfy the constraints of the theorem, and thus $\mathbf{S}$ is convergent. This implies that $\mathbf{B}$ has a *convergent regular splitting* and, as a result, the linearized dynamical system is unconditionally globally asymptotically stable for all the values of parameters and inputs. Further, the global asymptotic stability of linearization implies the local asymptotic stability of the normalization fixed point for ORGaNICs.

This result holds even when the neurons have different time constants, regardless of their type, as no assumptions were made about the time constants. This finding is significant for machine learning, particularly for designing architectures based on ORGaNICs. It allows neurons/units to integrate information at varying time scales while maintaining a stable circuit that performs normalization dynamically. Moreover, analytical expressions for eigenvalues can be obtained in the following case,

**Theorem 4.4.** *Let $\mathbf{W}_r = \mathbf{I}$, the normalization matrix be given by $\mathbf{W} = \alpha\mathbf{E}$, where $\mathbf{E}$ is the all-ones matrix, and the parameters are scalars, i.e., $\boldsymbol{\tau}_y = \tau_y\mathbf{1}$, $\boldsymbol{\tau}_a = \tau_a\mathbf{1}$, $\mathbf{b}_0 = b_0\mathbf{1}$, and $\boldsymbol{\sigma} = \sigma\mathbf{1}$. Under these conditions, the eigenvalues of the system admit closed form solutions (detailed in Appendix C).*

This result is particularly useful for neuroscience as it elucidates the connection between ORGaNICs parameters and the strength and frequency of oscillatory activity. Since we followed a direct Lyapunov approach to prove Theorem 4.1 as shown in Appendix B, we can derive an *energy* (viz., Lyapunov function) for ORGaNICs as shown in Appendix H.

**Theorem 4.5.** *When $\mathbf{W}_r = \mathbf{I}$, the energy (Lyapunov function) minimized by ORGaNICs in the vicinity of the normalization fixed point, is given by,*

$$V(\mathbf{y}, \mathbf{a}) = \sum_{i=1}^{n} t_i \frac{a_{s_i}}{y_{s_i}^2}\left[\frac{\tau_{y_i}}{\tau_{a_i}}\sqrt{a_{s_i}}\left(y_i - y_{s_i}\right)^2 + \left(\sqrt{a_i}y_i - \sqrt{a_{s_i}}y_{s_i}\right)^2\right]. \quad (13)$$

*Where $t_i$ are the diagonal entries of $\mathbf{T}$ and $y_{s_i}$ $(a_{s_i})$ are the steady-state values of neurons $y_i$ $(a_i)$.*

Specifically, for a two-dimensional model (one $y$ neuron and one $a$ neuron) this expression simplifies to reveal that ORGaNICs behave like a damped harmonic oscillator with *energy*,

$$V(y, a) = \frac{\tau_y}{\tau_a}\sqrt{b_0^2\sigma^2 + wb^2z^2}\left(y - \frac{bz}{\sqrt{b_0^2\sigma^2 + wb^2z^2}}\right)^2 + (\sqrt{a}y - bz)^2 \quad (14)$$

This result demonstrates that ORGaNICs minimize the residual of the instantaneously reconstructed gated input drive ($\sqrt{a}y - bz$), while also ensuring that the principal neuron's response, $y$, achieves DN. The balance between these objectives is governed by the parameters and the external input strength. With fixed parameters, weaker inputs, $z$, cause the model to prioritize input matching over normalization, whereas stronger inputs increasingly engage the normalization objective.

## 5 Stability analysis for arbitrary recurrent weights

Now, we relax the constraint that the recurrent weight matrix must be identity, allowing $\mathbf{W}_r \neq \mathbf{I}$, and see how the stability result changes. This leads to the following set of equations,

$$\begin{aligned} \boldsymbol{\tau}_y \odot \dot{\mathbf{y}} &= -\mathbf{y} + \mathbf{b} \odot \mathbf{z} + \left(\mathbf{1} - \sqrt{\lfloor \mathbf{a} \rfloor}\right) \odot (\mathbf{W}_r\mathbf{y}) \\ \boldsymbol{\tau}_a \odot \dot{\mathbf{a}} &= -\mathbf{a} + \mathbf{b}_0^2 \odot \boldsymbol{\sigma}^2 + \mathbf{W}\,\left(\mathbf{y}^2 \odot \lfloor \mathbf{a} \rfloor\right) \end{aligned} \quad (15)$$

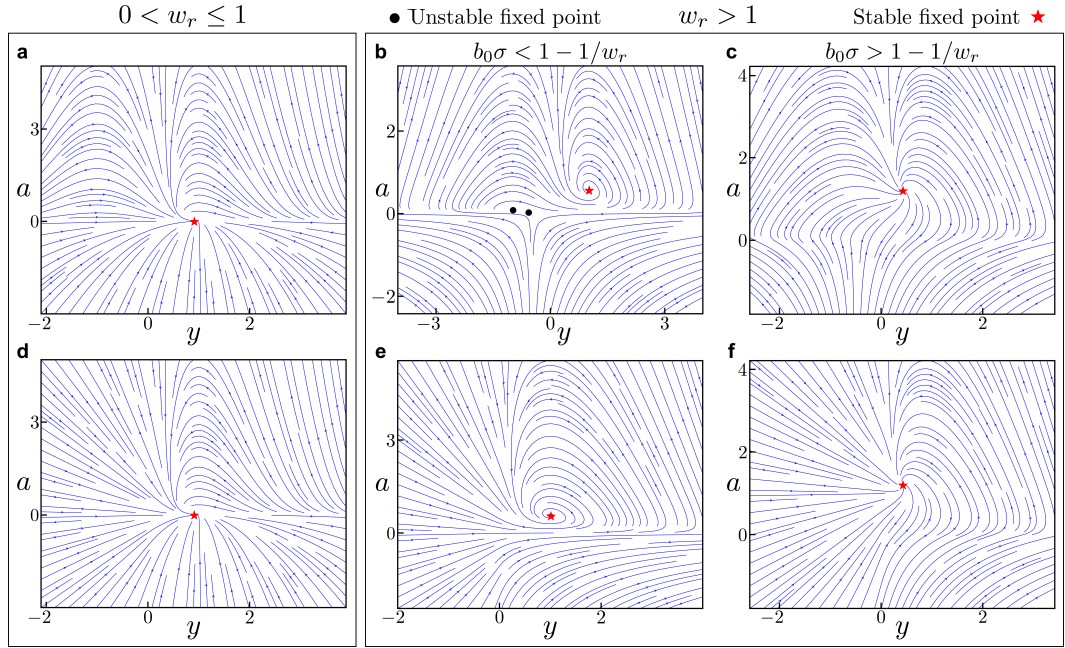

Figure 1: **Phase portraits for 2D ORGaNICs with positive input drive.** We plot the phase portraits of 2D ORGaNICs in the vicinity of the stable fixed points for contractive (**a, d**) and expansive (**b, c, e, f**) recurrence scalar $w_r$. A stable fixed point always exists, regardless of the parameter values. **(a-c)**, The main model (Eq. 16). **(d-f)**, The rectified model (Eq. 102). Red stars and black circles indicate stable and unstable fixed points, respectively. The parameters for all plots are: $b = 0.5$, $\tau_a = 2$ ms, $\tau_y = 2$ ms, $w = 1.0$, and $z = 1.0$. For **(a) & (d)**, the parameters are $w_r = 0.5$, $b_0 = 0.5$, $\sigma = 0.1$; for **(b) & (e)**, $w_r = 2.0$, $b_0 = 0.5$, $\sigma = 0.1$; and for **(c) & (f)**, $w_r = 2.0$, $b_0 = 1.0$, $\sigma = 1.0$.

The linear stability analysis becomes intractable for a general $\mathbf{W}_r$ because we no longer have a closed-form analytical expression for the steady states of $\mathbf{y}$ and $\mathbf{a}$. Additionally, the characteristic polynomial cannot be expressed in a way similar to Eq.8. Nevertheless, for a two-dimensional system,

$$\tau_y \dot{y} = -y + bz + \left(1 - \sqrt{\lfloor a \rfloor}\right) w_r y$$
$$\tau_a \dot{a} = -a + b_0^2 \sigma^2 + w y^2 \lfloor a \rfloor \tag{16}$$

we can prove the following, with a detailed analysis provided in Appendix E.

**Theorem 5.1.** *Given that the recurrence is contracting, i.e., $0 < w_r \leq 1$, when $z > 0$ ($z < 0$) there exists a unique fixed point with $y_s > 0$ ($y_s < 0$) and $a_s > 0$, and it is asymptotically stable.*

**Theorem 5.2.** *Given that the recurrence is expansive, i.e., $w_r > 1$, there are either 1 or 3 fixed points of which at least one is asymptotically stable. When $z > 0$ ($z < 0$) there exists exactly 1 fixed point with $y_s > 0$ ($y_s < 0$) and $a_s > 0$, and it is asymptotically stable. If $b_0\sigma > 1 - 1/w_r$, there are no additional fixed points. If $b_0\sigma < 1 - 1/w_r$, there exist either 0 or 2 additional fixed points with $y_s < 0$ ($y_s > 0$) and $a_s > 0$ whose stability cannot be guaranteed.*

We plot the phase portraits for these different cases in Fig. 1. The key takeaway is that there is always a fixed point $(y_s, a_s)$ with $a_s > 0$ and $y_s$ having the same sign as $z$. This fixed point is asymptotically stable regardless of the value of $w_r$. Based on these results and the proven stability of arbitrary dimensional ORGaNICs when $\mathbf{W}_r = \mathbf{I}$ (as shown in Section 4), we conjecture that

**Conjecture 5.3.** *Consider high-dimensional ORGaNICs with an arbitrary recurrent weight matrix $\mathbf{W}_r$ and no constraints on the remaining parameters. If the norm of the input drive satisfies $||\mathbf{z}|| \leq 1$, and the maximum singular value of $\mathbf{W}_r$ is constrained to be 1, then the system possesses at least one asymptotically stable fixed point.*

This conjecture is supported by empirical evidence showing consistent stability, as ORGaNICs initialized with random parameters and inputs under these constraints have exhibited stability in 100%

**Algorithm 1** Iterative scheme for the fixed point when the maximum singular value of $\mathbf{W}_r$ is 1

---
1: **Input:** ORGaNICs parameters, input $(\mathbf{z})$, tolerance $(\epsilon)$, maximum iterations $(N)$
2: **Output:** Approximation to the fixed point $(\mathbf{y}_s, \mathbf{a}_s)$
3: $\mathbf{a} \leftarrow \boldsymbol{\sigma}^2 \odot \mathbf{b}_0^2 + \mathbf{W}\left(\mathbf{W}_r\left(\mathbf{b} \odot \mathbf{z}\right)\right)^2$        // initial approximation for $a$ neurons
4: $\mathbf{y} \leftarrow \left(\mathbf{W}_r\left(\mathbf{b} \odot \mathbf{z}\right)\right)/\sqrt{\mathbf{a}}$        // initial approximation for $y$ neurons
5: $k \leftarrow 0$
6: **while** $\left\|\mathbf{y} - \mathbf{b} \odot \mathbf{z} - \left(\mathbf{1} - \sqrt{\mathbf{a}}\right) \odot \left(\mathbf{W}_r\mathbf{y}\right)\right\| > \epsilon$ and $k < N$ **do**
7:      $\mathbf{y} \leftarrow \left(\mathbf{I} - \mathbf{W}_r + \mathbf{D}\left(\sqrt{\mathbf{a}}\right)\mathbf{W}_r\right)^{-1}\left(\mathbf{b} \odot \mathbf{z}\right)$        // $y$ neurons update
8:      $\mathbf{a} \leftarrow \mathbf{b}_0^2 \odot \boldsymbol{\sigma}^2 + \mathbf{W}\left(\mathbf{y}^2 * \mathbf{a}\right)$        // $a$ neurons update
9:      $k \leftarrow k + 1$
10: **end while**
11: **return** $(\mathbf{y}, \mathbf{a})$

---

of trials, see Fig. 4. We further speculate that ORGaNICs may be *typically* stable beyond this regime as we find that 100% of trials yield a stable circuit when the constraint on the maximum singular value of $\mathbf{W}_r$ is increased to 2, but it becomes unstable when it is increased to 3.

## 6 Experiments

We provide further empirical evidence in support of Conjecture 5.3 that ORGaNICs is asymptotically stable by showing that stability is preserved when training ORGaNICs using naïve BPTT on two different tasks: 1) static classification of MNIST, 2) sequential classification of pixel-by-pixel MNIST. Because these ML tasks have no relevance for neurobiological or cognitive processes, we relax one aspect of the biological plausibility of ORGaNICs, specifically, allowing arbitrary (learned) nonnegative values for the intrinsic time constants.[1]

### 6.1 Static input classification task

We first show that we can train ORGaNICs on the MNIST handwritten digit dataset [72] presented to the circuit as a static input. This setting corresponds to evolving the responses of the neurons dynamically until they reach a fixed point solution and using the steady-state firing rates of the principal neurons to predict the labels, akin to deep equilibrium models [73]. While the fixed point of the circuit is known when $\mathbf{W}_r = \mathbf{I}$ (given by Eq. 89), we allow $\mathbf{W}_r$ to be learnable and parameterized it to have a maximum singular value of 1. This constraint allows us to find the fixed point responses of all the neurons without simulation, using a fixed point iteration scheme (Algorithm 1) that converges with great accuracy in a few (less than 5) steps, see Fig. 4 & 5. We provide an intuition for why this algorithm works with empirical evidence of fast convergence in Appendix G.

Table 1: Test accuracy on MNIST dataset

| Model | Accuracy |
|---|---|
| SSN (50:50) | 94.9% |
| SSN (80:20) | 95.2% |
| MLP (50) | 98.2% |
| **ORGaNICs** (50:50) | 98.1% |
| **ORGaNICs** (80:80) | 98.2% |
| **ORGaNICs** (two layers) | 98.1% |

We trained ORGaNICs on this task (details provided in Appendix I.1) and compared its performance to SSN [5] trained by dynamics-neutral growth [55]. We found that ORGaNICs perform better than SSN with the same model size, and on par with an MLP (Table 1). We analyzed the eigenvalues of the Jacobian matrix of the trained model and consistently found the largest real part to be negative (Fig. 5), indicating stability. Moreover, we found that stability was maintained during training (Fig. 6).

### 6.2 Time varying input

We trained unconstrained ORGaNICs by naïve BPTT on a classification task of sequential MNIST (sMNIST), proposed by Le et al. [74]. This is a challenging task because it involves long-term dependencies and requires the architecture to maintain and integrate information over long timescales. Briefly, the task involves the presentation of pixels of MNIST images sequentially (one pixel for each

---
[1]Python code for this study is available at https://github.com/martiniani-lab/dynamic-divisive-norm.

Table 2: Test accuracy on sequential pixel-by-pixel MNIST and permuted MNIST

| Model | sMNIST | psMNIST | # units | # params |
|---|---|---|---|---|
| LSTMs [75] | 97.3% | 92.6% | 128 | 68k |
| AntisymmetricRNN [59] | 98.0% | 95.8% | 128 | 10k |
| coRNN [61] | 99.3% | 96.6% | 128 | 34k |
| Lipschitz RNN [60] | 99.4% | 96.3% | 128 | 34k |
| **ORGaNICs** (fixed time constants) | 90.3% | 80.3% | 64 | 26k |
| **ORGaNICs** (fixed time constants) | 94.8% | 84.8% | 128 | 100k |
| **ORGaNICs** | 97.7% | 89.9% | 64 | 26k |
| **ORGaNICs** | 97.8% | 90.7% | 128 | 100k |

timestep) in scanline order, and at the end of the input the model has to predict the digit that was presented. There is a more complicated version of this task, permuted sequential MNIST, in which the pixels of all images are permuted in some random order before being presented sequentially. We train ORGaNICs with different hidden layer sizes (number of $\mathbf{y}$ neurons) on these two tasks by discretizing the rectified ORGaNICs with arbitrary recurrence, Eq. 87, which has all the properties that we have derived for the main model. Since an unstable fixed point is undesirable in such a task, as it may lead to diverging trajectories, we prefer the rectified model (Appendix F) over the main model. We proved that the 2D rectified ORGaNICs (Eq. 102) does not exhibit an unstable fixed point for positive inputs, as it can also be seen in Fig 1. The hidden states of the neurons are initialized with a uniform random distribution (for more details, see Appendix I.2). Additionally, we make the input gains $\mathbf{b}$ and $\mathbf{b}_0$ dynamical with their ODEs given by,

$$
\boldsymbol{\tau}_b \odot \dot{\mathbf{b}} = -\mathbf{b} + f(\mathbf{W}_{bx}\mathbf{x} + \mathbf{W}_{by}\mathbf{y} + \mathbf{W}_{ba}\mathbf{a})
$$
$$
\boldsymbol{\tau}_{b_0} \odot \dot{\mathbf{b}}_0 = -\mathbf{b}_0 + f(\mathbf{W}_{b_0 x}\mathbf{x} + \mathbf{W}_{b_0 y}\mathbf{y} + \mathbf{W}_{b_0 a}\mathbf{a})
$$

(17)

We achieved slightly better performance than LSTMs on sMNIST with a smaller model size and comparable performance on permuted sMNIST, without hyperparameter optimization and without gradient clipping/scaling (Table 2). We found that the trajectories of $\mathbf{y}$ are bounded when it is trained on the sequential task (Fig. 7), indicating stability. We also show that the training of ORGaNICs is stable and does not require gradient clipping when the intrinsic time constants of the neurons are fixed (Table 2).

# 7 Discussion

**Summary:** While extensive research has been aimed at identifying highly expressive RNN architectures that can model complex data, there has been little advancement in developing robust, biologically plausible recurrent neural circuits that are easy to train and perform comparably to their artificial counterparts. Regularization techniques such as batch, group, and layer normalization have been developed and are implemented as ad hoc add-ons making them biologically implausible. In this work, we bridge these gaps by leveraging the recently proposed ORGaNICs model which implements divisive normalization (DN) dynamically in a recurrent circuit. We establish the unconditional stability of an arbitrary dimensional ORGaNICs circuit with an identity recurrent weight matrix ($\mathbf{W}_r$), with all of the other parameters and inputs unconstrained, and provide empirical evidence of stability for ORGaNICs with arbitrary $\mathbf{W}_r$. Since ORGaNICs remain stable for all parameter values and inputs, we do not need to resort to techniques that are restrictive in parameter space, or that require designing unrealistic structures for weight matrices. ORGaNICs' intrinsic stability mitigates the issues of exploding and oscillating gradients, enabling the use of "vanilla" BPTT without the need for gradient clipping, which is instead required when training LSTMs. Moreover, ORGaNICs effectively address the vanishing gradient problem often encountered when training RNNs. This is achieved by processing information across various timescales, resulting in a blend of lossy and non-lossy neurons, while preserving stability. The model's effectiveness in overcoming vanishing gradients is further evidenced by its competitive performance against architectures specifically designed to address this issue, such as LSTMs.

**Dynamic normalization:** Normalization techniques, such as batch and layer normalization, are fundamental in modern ML architectures significantly enhancing the training and performance of

CNNs. However, a principled approach to incorporating normalization into RNNs has remained elusive. While layer normalization is commonly applied to RNNs to stabilize training, it does not influence the underlying circuit dynamics since it is applied a-posteriori to the output activations, leaving the stability of RNNs unaffected. Furthermore, DN has been shown to generalize batch and layer normalization [39], leading to improved performance [39, 41, 42]. ORGaNICs, unlike RNNs with layer normalization, implement DN dynamically within the circuit, marking the first instance of this concept being applied and analyzed in ML. Our work demonstrates that embedding DN within a circuit naturally leads to stability, which is greatly advantageous for trainability. This stability, a consequence of dynamic DN, sets ORGaNICs apart from other RNNs by providing both output normalization and model robustness. As a result, ORGaNICs can be trained using BPTT, achieving performance on par with LSTMs. The key insight is that the dynamic application of DN not only enhances training efficiency but also improves model robustness. This illustrates how the incorporation of neurobiological principles can drive advances in ML.

**Interpretability:** In the proof of stability, we establish a direct connection between ORGaNICs and systems of coupled damped harmonic oscillators, which have long been studied in mechanics and control theory. This analogy not only enables us to derive an interpretable energy function for ORGaNICs (Eq. 13), providing a normative principle of what the circuit aims to accomplish, but also sheds light on the link between normalization and dynamical stability of neural circuits. For a relevant ML task, having an analytical expression for the energy function allows us to quantify the relative contributions of the individual neurons in the trained model, offering more interpretability than other RNN architectures. For instance, Eq. 13 shows that the ratio of time constants ($\tau_y/\tau_a$) for E-I neuron pairs determines how much weight a neuron assigns to divisive normalization relative to aligning its responses with the input drive $\mathbf{z}$. This insight provides a clear functional role for each neuron in the trained model. Moreover, since ORGaNICs are biologically plausible, we can understand how the various components of the dynamical system might be computed within a neural circuit [48], bridging the gap between theoretical models and biological implementation, and offering a means to generate and test hypotheses about neural computation in real biological systems (which we will be reporting elsewhere).

**Limitations:** Although the stability property pertains to a continuous-time system of nonlinear differential equations, typical implementations for tasks with sequential data involve an Euler discretization of these equations for training purposes. This might lead to a stiff dynamical system, potentially causing numerical instabilities and explosive dynamics, highlighting the importance of carefully parameterizing time constants and choosing a small enough time step to maintain stable dynamics. The proof of unconditional stability is only tractable for the two-dimensional circuit and the high-dimensional circuit with $\mathbf{W}_r = \mathbf{I}$. Therefore, we can only conjecture the stability of ORGaNICs for arbitrary $\mathbf{W}_r$, based on these two limiting cases and on empirical evidence. In the current form, the weight matrices of the input gain modulators, $\mathbf{W}_{by}$, $\mathbf{W}_{ba}$, $\mathbf{W}_{b_0y}$, and $\mathbf{W}_{b_0a}$, are each $n \times n$. As a result, the number of parameters grows more rapidly with the hidden state size compared to other RNNs. To mitigate this, we plan to explore using compact and/or convolutional weights to prevent a significant increase in the number of parameters as the hidden state size expands.

**Attention mechanisms in ORGaNICs:** ORGaNICs have a built-in mechanism for attention: modulating the input gain $\mathbf{b}$ (e.g., Eq. 17), coupled with DN. This attention mechanism aligns with experimental data on both increases in the gain of neural responses and improvements in behavioral performance [19, 20, 32, 76–85]. Moreover, this mechanism performs a computation that is analogous to that of an attention head in ML systems (including transformers [2]) as both operate by changing the gain over time. In ORGaNICs, DN replaces the softmax operation typically used in an attention head.

**Future work:** This study has explored only a single layer of ORGaNICs for the sequential tasks. Future work will examine how stacked layers with feedback connections, similar to those in the cortex, perform on benchmarks for sequential modeling and also on cognitive tasks with long-term dependencies. We have thus far shown that ORGaNICs can address the problem of long-term dependencies by learning intrinsic time constants. Future investigations will assess the performance of ORGaNICs for tasks with long-term dependencies by learning to modulate the responses of the $\mathbf{a}$ and $\mathbf{b}$ neurons to control the effective time constant of the recurrent circuit (without changing the intrinsic time constants) [47], i.e., implementing a working memory circuit capable of learning to maintain and manipulate information across various timescales.

## Acknowledgments and Disclosure of Funding

The authors thank anonymous reviewers for their insightful suggestions. The authors also acknowledge valuable discussions with Flaviano Morone, Asit Pal, Mathias Casiulis, and Guanming Zhang. This work was supported by the National Institute of Health under award number R01EY035242. S.M. acknowledges the Simons Center for Computational Physical Chemistry for financial support. This work was supported in part through the NYU IT High Performance Computing resources, services, and staff expertise.

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

# A  Derivation of ORGaNICs

Here, we derive a generalized 2-neuron types (excitatory and inhibitory) ORGaNICs model for a high-dimensional input. The system presented in Eq. 1 is a special case of this generalized model where $p = 2$ and $\mathbf{a}^+ = \sqrt{\lfloor \mathbf{a} \rfloor}$. Assuming $\mathbf{W}$ is the normalization weight matrix, and $\mathbf{z}$ is the input drive, we can write the normalization equations for principal neurons with complementary receptive fields as,

$$\mathbf{y}_s^+ = \frac{\lfloor \mathbf{z} \rfloor^p}{\boldsymbol{\sigma}^p + \mathbf{W}\left(\lfloor \mathbf{z} \rfloor^p + \lfloor -\mathbf{z} \rfloor^p\right)}; \quad \mathbf{y}_s^- = \frac{\lfloor -\mathbf{z} \rfloor^p}{\boldsymbol{\sigma}^p + \mathbf{W}\left(\lfloor \mathbf{z} \rfloor^p + \lfloor -\mathbf{z} \rfloor^p\right)} \tag{18}$$

Note that typically the exponent of the input $p \sim 2$ for cortical neurons. $\lfloor \mathbf{z} \rfloor^p$ and $\lfloor -\mathbf{z} \rfloor^p$ represent the contribution of neurons with complementary receptive fields to the normalization pool. Mathematically, we have, $\lfloor \mathbf{z} \rfloor^p + \lfloor -\mathbf{z} \rfloor^p = |\mathbf{z}|^p$.

Here, we derive, for a general $p$, the dynamical equations that have the fixed point defined by the normalization equation above. First, it is important to distinguish between the membrane potentials and the firing rates of neurons. The coarse (low-pass filtered) membrane potential of a given type of neuron is denoted by the vector, $\mathbf{y}, \mathbf{a}$, with the corresponding firing rates of $\mathbf{y}^+(\mathbf{y}^-), \mathbf{a}^+$. The instantaneous firing rates of the neurons are obtained from the corresponding coarse membrane potentials by applying rectification, denoted by $\lfloor . \rfloor$, and a power law (sub/supra-linear) activation for different types of neurons [86–88]. Therefore, for a set of membrane potentials $\mathbf{x}$ the instantaneous firing rates are $\mathbf{x}^+ = \lfloor \mathbf{x} \rfloor^\alpha$. Specifically for principal neurons, we have, $\mathbf{y}^+ = \lfloor \mathbf{y} \rfloor^p$ and $\mathbf{y}^- = \lfloor -\mathbf{y} \rfloor^p$. Combining the firing of principal neurons with the complementary receptive fields, $\mathbf{y}^+$ and $\mathbf{y}^-$, Eq. 18 can be alternatively written as,

$$|\mathbf{y}_s|^p = \mathbf{y}_s^+ + \mathbf{y}_s^- = \frac{|\mathbf{z}|^p}{\boldsymbol{\sigma}^p + \mathbf{W}\,|\mathbf{z}|^p} \tag{19}$$

Now for the principal neuron $y_j$ receiving an input drive $z_j$, we can rewrite the normalization equation for each neuron as,

$$|y_j^s|^p = \frac{|z_j|^p}{\sigma_j^p + \sum_k W_{jk}|z_k|^p} \tag{20}$$

In the ORGaNICs paradigm [47], the steady-state activity of the principal neurons (single equation for complementary receptive fields) is a weighted sum of input drive and recurrent drive,

$$\tau_{y_j} \frac{\mathrm{d}y_j}{\mathrm{d}t} = -y_j + \underbrace{b_j z_j}_{\text{Weighted input drive}} + \underbrace{\left(1 - a_j^+\right) \sum_k w_{rjk}\left((y_k^+)^{1/p} - (y_k^-)^{1/p}\right)}_{\text{Weighted recurrent drive}} \tag{21}$$

Here $w_r$ are the weights of the recurrent weight matrix $\mathbf{W}_r$ encoding the recurrent/lateral connections between the principal neurons $\mathbf{y}$; $y_k^+$ is the firing rate of the principal neuron $k$, given by $y_k^+ = \lfloor y_k \rfloor^p$, and $y_k^- = \lfloor -y_k \rfloor^p$ is the firing rate of the complementary principal neuron $k$. $b_j$ is the gain to the input drive $z_j$ which simulates attention (realized via gain modulation). $(1 - a_j^+)$ is the gain to the recurrent drive which controls the recurrent amplification.

Now, we find the dynamics of the inhibitory neurons $a_j$ with firing rates $a_j^+$, that yield stable dynamics with the fixed point given by Eq. 18. First, we assume that the recurrent weight matrix $\mathbf{W}_r = \mathbf{I}$. Also, note that the following identity holds: $(y_k^+)^{1/p} - (y_k^-)^{1/p} = \lfloor y_k \rfloor - \lfloor -y_k \rfloor = y_k$. Therefore, Eq.21 can be simplified to,

$$\tau_{y_j} \frac{\mathrm{d}y_j}{\mathrm{d}t} = -y_j + b_j z_j + \left(1 - a_j^+\right) y_j \tag{22}$$

At steady-state, the fixed-points $(y_j^s, a_j^s)$, satisfy the following relationship,

$$a_j^{s+} y_j^s = b_j z_j \tag{23}$$

Taking modulus and raising both sides to power $p$, we get,

$$\left|a_j^{s+} y_j^s\right|^p = |b_j z_j|^p \tag{24}$$

or in vector form,

$$\left|\mathbf{a}_s^+ \odot \mathbf{y}_s\right|^p = |\mathbf{b} \odot \mathbf{z}|^p$$
$$\left(\mathbf{a}_s^+\right)^p \odot |\mathbf{y}_s|^p = \mathbf{b}^p \odot |\mathbf{z}|^p \tag{25}$$

From Eq. 19, we know that

$$|\mathbf{z}|^p = \boldsymbol{\sigma}^p \odot |\mathbf{y}_s|^p + |\mathbf{y}_s|^p \odot \left(\mathbf{W}\, |\mathbf{z}|^p\right) \tag{26}$$

Since we can write the element-wise product between two vectors $\mathbf{x}_1$ and $\mathbf{x}_2$, $\mathbf{x}_1 \odot \mathbf{x}_2 = \mathbf{x}_2 \odot \mathbf{x}_1 = \mathbf{D}(\mathbf{x}_1)\mathbf{x}_2$. Where $\mathbf{D}(\mathbf{x}_1)$ is a diagonal matrix with elements of the vector $\mathbf{x}_1$ on the diagonal. Therefore, using this fact we can rewrite the equation above as,

$$|\mathbf{z}|^p = \boldsymbol{\sigma}^p \odot |\mathbf{y}_s|^p + \mathbf{D}\left(|\mathbf{y}_s|^p\right)\left(\mathbf{W}\, |\mathbf{z}|^p\right)$$
$$|\mathbf{z}|^p = \boldsymbol{\sigma}^p \odot |\mathbf{y}_s|^p + \mathbf{D}\left(|\mathbf{y}_s|^p\right)\mathbf{W}\, |\mathbf{z}|^p$$
$$\left[\mathbf{I} - \mathbf{D}\left(|\mathbf{y}_s|^p\right)\mathbf{W}\right]|\mathbf{z}|^p = \boldsymbol{\sigma}^p \odot |\mathbf{y}_s|^p \tag{27}$$
$$|\mathbf{z}|^p = \left[\mathbf{I} - \mathbf{D}\left(|\mathbf{y}_s|^p\right)\mathbf{W}\right]^{-1}\left(\boldsymbol{\sigma}^p \odot |\mathbf{y}_s|^p\right)$$

Substituting the expression for $|\mathbf{z}|^p$ into Eq. 25, we get,

$$\left(\mathbf{a}_s^+\right)^p \odot |\mathbf{y}_s|^p = \mathbf{b}^p \odot \left(\left[\mathbf{I} - \mathbf{D}\left(|\mathbf{y}_s|^p\right)\mathbf{W}\right]^{-1}\left(\boldsymbol{\sigma}^p \odot |\mathbf{y}_s|^p\right)\right) \tag{28}$$

We simplify this equation as,

$$\left[\mathbf{I} - \mathbf{D}\left(|\mathbf{y}_s|^p\right)\mathbf{W}\right]\left(\frac{\left(\mathbf{a}_s^+\right)^p \odot |\mathbf{y}_s|^p}{\mathbf{b}^p}\right) = \boldsymbol{\sigma}^p \odot |\mathbf{y}_s|^p$$
$$\mathbf{D}\left(1/|\mathbf{y}_s|^p\right)\left[\mathbf{I} - \mathbf{D}\left(|\mathbf{y}_s|^p\right)\mathbf{W}\right]\left(\frac{\left(\mathbf{a}_s^+\right)^p \odot |\mathbf{y}_s|^p}{\mathbf{b}^p}\right) = \mathbf{D}\left(1/|\mathbf{y}_s|^p\right)\left(\boldsymbol{\sigma}^p \odot |\mathbf{y}_s|^p\right) \tag{29}$$
$$\frac{\left(\mathbf{a}_s^+\right)^p}{\mathbf{b}^p} - \mathbf{W}\left(\frac{\left(\mathbf{a}_s^+\right)^p \odot |\mathbf{y}_s|^p}{\mathbf{b}^p}\right) = \boldsymbol{\sigma}^p$$

Now we assume that all of the entries of $\mathbf{b}$ are equal to a constant $b_0$, therefore, we have,

$$\left(\mathbf{a}_s^+\right)^p - \mathbf{W}\left(\left(\mathbf{a}_s^+\right)^p \odot |\mathbf{y}_s|^p\right) = b_0^p \boldsymbol{\sigma}^p$$
$$\left(\mathbf{a}_s^+\right)^p = b_0^p \boldsymbol{\sigma}^p + \mathbf{W}\left(\left(\mathbf{a}_s^+\right)^p \odot |\mathbf{y}_s|^p\right) \tag{30}$$

Element-wise multiplying both sides by $\mathbf{a}_s / \left(\mathbf{a}_s^+\right)^p$ on the left, we get,

$$\mathbf{0} = -\mathbf{a}_s + \frac{\mathbf{a}_s}{\left(\mathbf{a}_s^+\right)^p} \odot \left[b_0^p \boldsymbol{\sigma}^p + \mathbf{W}\left(\left(\mathbf{a}_s^+\right)^p \odot |\mathbf{y}_s|^p\right)\right] \tag{31}$$

This equation is true at the steady-state, but it is also in a form similar to that of $\mathbf{y}$, such that we have a weighted input drive and a weighted recurrent drive, therefore, we propose the following dynamical equation for $\mathbf{a}$,

$$\boldsymbol{\tau}_a \odot \dot{\mathbf{a}} = -\mathbf{a} + \frac{\mathbf{a}}{\left(\mathbf{a}^+\right)^p} \odot \left[b_0^p \boldsymbol{\sigma}^p + \mathbf{W}\left(\left(\mathbf{a}^+\right)^p \odot |\mathbf{y}|^p\right)\right] \tag{32}$$

This equation naturally follows Eq. 31 at steady-state and thus also follows the normalization equation Eq. 19. Note that the equation above is true for any choice of $\mathbf{a}^+$, but cortical neurons have been experimentally observed to have an exponent close to $p = 2$, i.e., $\mathbf{y}^+ = \lfloor\mathbf{y}\rfloor^2$ and $\mathbf{y}^- = \lfloor-\mathbf{y}\rfloor^2$. Therefore, we get a particularly simple form of the equations when $p = 2$ and $\mathbf{a}^+ = \sqrt{\lfloor\mathbf{a}\rfloor}$. In this case, the equation for $\mathbf{a}$ is given by,

$$\boldsymbol{\tau}_a \odot \dot{\mathbf{a}} = -\mathbf{a} + \left[b_0^2 \boldsymbol{\sigma}^2 + \mathbf{W}\left(\left(\mathbf{a}^+\right)^2 \odot \mathbf{y}^2\right)\right] \tag{33}$$

We reintroduce the recurrent weight matrix $\mathbf{W}_r$ and to simulate the effect of attention by gain modulation, we define different gains for $\mathbf{y}$ and $\mathbf{a}$ neurons. Additionally, replacing $\mathbf{y}^2 \to \mathbf{y}^+ + \mathbf{y}^-$ and $\mathbf{y} \to \sqrt{\mathbf{y}^+} - \sqrt{\mathbf{y}^-}$ yields the dynamical system that we analyze,

$$\boldsymbol{\tau}_y \odot \dot{\mathbf{y}} = -\mathbf{y} + \mathbf{b} \odot \mathbf{z} + \left(\mathbf{1} - \mathbf{a}^+\right) \odot \left(\mathbf{W}_r\left(\sqrt{\mathbf{y}^+} - \sqrt{\mathbf{y}^-}\right)\right)$$
$$\boldsymbol{\tau}_a \odot \dot{\mathbf{a}} = -\mathbf{a} + b_0^2 \odot \boldsymbol{\sigma}^2 + \mathbf{W}\left(\left(\mathbf{y}^+ + \mathbf{y}^-\right) \odot \mathbf{a}^{+2}\right) \tag{34}$$

Finally, in the most general form, i.e., any choice of $p$ and nonlinearity for $\mathbf{a}^+$, the dynamical system of ORGaNICs is given by,

$$\boldsymbol{\tau}_y \odot \dot{\mathbf{y}} = -\mathbf{y} + \mathbf{b} \odot \mathbf{z} + \left(1 - \mathbf{a}^+\right) \odot \left(\mathbf{W}_r \left(\left(\mathbf{y}^+\right)^{1/p} - \left(\mathbf{y}^-\right)^{1/p}\right)\right)$$

$$\boldsymbol{\tau}_a \odot \dot{\mathbf{a}} = -\mathbf{a} + \frac{\mathbf{a}}{(\mathbf{a}^+)^p} \odot \left[\mathbf{b}_0^p \odot \boldsymbol{\sigma}^p + \mathbf{W} \left(\left(\mathbf{y}^+ + \mathbf{y}^-\right) \odot \left(\mathbf{a}^+\right)^p\right)\right] \tag{35}$$

# B  Stability theorem

**Theorem B.1.** *For a system of linear differential equations with constant coefficients of the form,*

$$\mathbf{I}\ddot{\mathbf{x}} + \mathbf{B}\dot{\mathbf{x}} + \mathbf{K}\mathbf{x} = \mathbf{0} \tag{36}$$

*where $\mathbf{B} \in \mathbb{R}^{n \times n}$ and $\mathbf{K} \in \mathbb{R}^{n \times n}$ is a positive diagonal matrix (hence $\mathbf{K} \succ 0$), the dynamical system is globally asymptotically stable if $\mathbf{B}$ is Lyapunov diagonally stable.*

*Proof.* The stability of the system is defined by solving the following associated quadratic eigenvalue problem,

$$\mathbb{L}(\lambda) = \det(\lambda^2 \mathbf{I} + \mathbf{B}\lambda + \mathbf{K}) \tag{37}$$

The spectrum of $\mathbb{L}(\lambda)$, i.e., $\{\lambda \in \mathbb{C} : \det(\mathbb{L}(\lambda)) = 0\}$ are also known as the eigenvalues of the system. The system defined by Eq. 36 is globally asymptotically stable if all of the eigenvalues have negative real parts. We take the direct Lyapunov approach to prove the stability of the linear system. We write Eq. 36 in the matrix form as $\dot{\mathbf{z}} = \mathbf{J}\mathbf{z}$, where,

$$\mathbf{z} = \begin{bmatrix} \mathbf{x} \\ \dot{\mathbf{x}} \end{bmatrix} \quad \text{and} \quad \mathbf{J} = \begin{bmatrix} \mathbf{0} & \mathbf{I} \\ -\mathbf{K} & -\mathbf{B} \end{bmatrix} \tag{38}$$

We will first prove the Lyapunov stability ($\mathrm{Re}(\lambda_{\mathbf{J}}) \leq 0$) of this system to find the appropriate block diagonal matrices and then we will prove global asymptotic stability ($\mathrm{Re}(\lambda_{\mathbf{J}}) < 0$). To prove Lyapunov stability (Appendix B.2), we propose a Lyapunov function $V(\mathbf{z}) = \mathbf{z}^\top \mathbf{P}\mathbf{z}$, where $\mathbf{P}$ is a block positive definite matrix defined as follows,

$$\mathbf{P} = \begin{bmatrix} \mathbf{A} & \mathbf{0} \\ \mathbf{0} & \mathbf{T} \end{bmatrix} \tag{39}$$

where $\mathbf{T} \in \mathbb{R}^{n \times n}$ is a positive diagonal matrix such that $\mathbf{TB} + \mathbf{B}^\top \mathbf{T} \succ 0$ (notation for positive definite matrix) and $\mathbf{A} \in \mathbb{R}^{n \times n}$ a flexible symmetric positive definite matrix that we will find using the second Lyapunov criteria. Note that such a matrix $\mathbf{T}$ exits since $\mathbf{B}$ is defined to be Lyapunov diagonally stable. It can be easily seen that $\mathbf{P} \succ 0$ using the first criteria in Section B.1 since $\mathbf{A} \succ 0$ and $\mathbf{T} \succ 0$. Additionally, since $\mathbf{T} \succ 0$, it is invertible.

Now, for Lyapunov stability, we need $\dot{V}(\mathbf{z}) = \mathbf{z}^\top \left(\mathbf{PJ} + \mathbf{J}^\top \mathbf{P}\right) \mathbf{z} \leq 0$. Therefore, we find $\mathbf{A}$ such that $\mathbf{Q} = -\left(\mathbf{PJ} + \mathbf{J}^\top \mathbf{P}\right)$ is positive semi-definite.

$$\begin{aligned} \mathbf{Q} &= -\left(\mathbf{PJ} + \mathbf{J}^\top \mathbf{P}\right) \\ &= -\begin{bmatrix} \mathbf{A} & \mathbf{0} \\ \mathbf{0} & \mathbf{T} \end{bmatrix} \begin{bmatrix} \mathbf{0} & \mathbf{I} \\ -\mathbf{K} & -\mathbf{B} \end{bmatrix} - \begin{bmatrix} \mathbf{0} & -\mathbf{K} \\ \mathbf{I} & -\mathbf{B}^\top \end{bmatrix} \begin{bmatrix} \mathbf{A} & \mathbf{0} \\ \mathbf{0} & \mathbf{T} \end{bmatrix} \\ &= \begin{bmatrix} \mathbf{0} & \mathbf{KT} - \mathbf{A} \\ \mathbf{TK} - \mathbf{A} & \mathbf{TB} + \mathbf{B}^\top \mathbf{T} \end{bmatrix} \end{aligned} \tag{40}$$

We want to define $\mathbf{A} \succ 0$, such that $\mathbf{Q} \succeq 0$. Using the second criteria from Section B.1, we need $\mathbf{TB} + \mathbf{B}^\top \mathbf{T} \succ 0$ and $-(\mathbf{KT} - \mathbf{A})(\mathbf{TB} + \mathbf{B}^\top \mathbf{T})^{-1}(\mathbf{TK} - \mathbf{A}) \succeq 0$. The first condition is satisfied by the definition of $\mathbf{T}$. For the second condition to be satisfied, an obvious candidate for $\mathbf{A}$ is $\mathbf{TK}$. Note that both $\mathbf{T}$ and $\mathbf{K}$ are positive definite and diagonal, therefore they commute ($\mathbf{KT} = \mathbf{TK}$) and $\mathbf{A}$ is symmetric and positive definite. When $\mathbf{A} = \mathbf{TK}$, the LHS of the second condition becomes $\mathbf{0} \succeq 0$. Therefore, the system is Lyapunov stable.

Now, we prove the global asymptotic stability of the system by again using the direct Lyapunov approach. We propose the Lyapunov function of the same form as before, i.e., $V(\mathbf{z}) = \mathbf{z}^\top \mathbf{P}\mathbf{z}$, where $\mathbf{P}$ is a positive definite matrix. But for global asymptotic stability, we need a more stringent condition

on the Lyapunov function, $\dot{V}(\mathbf{z}) = \mathbf{z}^\top (\mathbf{PJ} + \mathbf{J}^\top \mathbf{P}) \mathbf{z} < 0$. Drawing inspiration from the previous exercise, we consider the following form of the matrix $\mathbf{P}$,

$$\mathbf{P} = \begin{bmatrix} \mathbf{TK} & \epsilon \mathbf{I} \\ \epsilon \mathbf{I} & \mathbf{T} \end{bmatrix} \tag{41}$$

Here, $\epsilon > 0$ is a scalar whose magnitude is to be determined based on the Lyapunov criteria for asymptotic stability. First, we need $\mathbf{P} \succ 0$. Applying the first criteria from Section B.1, we want $\epsilon$ to satisfy, $\mathbf{TK} - \epsilon^2 \mathbf{T}^{-1} \succ 0$. Second, for $\dot{V}(\mathbf{z}) < 0$, we want $\mathbf{Q} = -(\mathbf{PJ} + \mathbf{J}^\top \mathbf{P})$ to be positive definite. $\mathbf{Q}$ is given by,

$$\begin{aligned} \mathbf{Q} &= -\begin{bmatrix} \mathbf{TK} & \epsilon \mathbf{I} \\ \epsilon \mathbf{I} & \mathbf{T} \end{bmatrix} \begin{bmatrix} \mathbf{0} & \mathbf{I} \\ -\mathbf{K} & -\mathbf{B} \end{bmatrix} - \begin{bmatrix} \mathbf{0} & -\mathbf{K} \\ \mathbf{I} & -\mathbf{B}^\top \end{bmatrix} \begin{bmatrix} \mathbf{TK} & \epsilon \mathbf{I} \\ \epsilon \mathbf{I} & \mathbf{T} \end{bmatrix} \\ &= \begin{bmatrix} 2\epsilon \mathbf{K} & \epsilon \mathbf{B} \\ \epsilon \mathbf{B}^\top & \mathbf{TB} + \mathbf{B}^\top \mathbf{T} - 2\epsilon \mathbf{I} \end{bmatrix} \end{aligned} \tag{42}$$

Again, we apply the first criteria from Section B.1. $\mathbf{Q} \succ 0$ if and only if $\mathbf{TB} + \mathbf{B}^\top \mathbf{T} - 2\epsilon \mathbf{I} \succ 0$ and $2\epsilon \mathbf{K} - \epsilon^2 \mathbf{B} (\mathbf{TB} + \mathbf{B}^\top \mathbf{T} - 2\epsilon \mathbf{I})^{-1} \mathbf{B}^\top \succ 0$. To simplify notation we replace $\mathbf{TB} + \mathbf{B}^\top \mathbf{T}$ with a positive definite matrix $\mathbf{M}$. Therefore, we have to prove that there exists an $\epsilon > 0$ which satisfies the following conditions,

- $\mathbf{TK} - \epsilon^2 \mathbf{T}^{-1} \succ 0$

- $\mathbf{M} - 2\epsilon \mathbf{I} \succ 0$

- $2\mathbf{K} - \epsilon \mathbf{B} (\mathbf{M} - 2\epsilon \mathbf{I})^{-1} \mathbf{B}^\top \succ 0$

Assuming $t_i$ and $k_i$ to be the diagonal values of the positive diagonal matrices $\mathbf{T}$ and $\mathbf{K}$, we get the following two conditions, $\epsilon < \min(t_i \sqrt{k_i})$ and $\epsilon < \alpha/2$, where $\alpha$ is the smallest eigenvalue of $\mathbf{M}$, or $\alpha = \min(\lambda_\mathbf{M})$.

Now, for the third condition, we will use a number of facts about positive definite matrices which are all listed in [89]. We first consider the following matrix inequality, $\mathbf{M} \succeq \alpha \mathbf{I}$. This notation is equivalent to saying that $\mathbf{M} - \alpha \mathbf{I}$ is positive semi-definite or $\mathbf{M} - \alpha \mathbf{I} \succeq 0$. Therefore, we have,

$$\mathbf{M} - 2\epsilon \mathbf{I} \succeq (\alpha - 2\epsilon) \mathbf{I} \tag{43}$$

Assuming, $\epsilon$ is small enough such that the matrices on LHS and RHS are positive definite, we have,

$$\frac{1}{\alpha - 2\epsilon} \mathbf{I} \succeq (\mathbf{M} - 2\epsilon \mathbf{I})^{-1} \tag{44}$$

Since $\mathbf{B}$ is nonsingular, it is full rank, therefore, we have,

$$\frac{1}{\alpha - 2\epsilon} \mathbf{BB}^\top \succeq \mathbf{B} (\mathbf{M} - 2\epsilon \mathbf{I})^{-1} \mathbf{B}^\top \tag{45}$$

Multiplying both sides by $\epsilon$, we get,

$$\frac{\epsilon}{\alpha - 2\epsilon} \mathbf{BB}^\top \succeq \epsilon \mathbf{B} (\mathbf{M} - 2\epsilon \mathbf{I})^{-1} \mathbf{B}^\top \tag{46}$$

Notice that $\mathbf{BB}^\top$ is a positive definite matrix. Let $\beta$ be the maximum eigenvalue of $\mathbf{BB}^\top$, or $\beta = \max(\lambda_{\mathbf{BB}^\top})$. Therefore, we can add an upper-bound matrix to the inequality as follows,

$$\frac{\epsilon \beta}{\alpha - 2\epsilon} \mathbf{I} \succeq \frac{\epsilon}{\alpha - 2\epsilon} \mathbf{BB}^\top \succeq \epsilon \mathbf{B} (\mathbf{M} - 2\epsilon \mathbf{I})^{-1} \mathbf{B}^\top \tag{47}$$

Now, we find $\epsilon$ such that,

$$2\mathbf{K} \succ \frac{\epsilon \beta}{\alpha - 2\epsilon} \mathbf{I} \tag{48}$$

The range of values for which the above inequality is true is,

$$\epsilon < \min \left( \frac{2 k_i \alpha}{\beta + 4 k_i} \right) \tag{49}$$

If $\epsilon$ satisfies the inequality above, we have,

$$2\mathbf{K} \succ \frac{\epsilon\beta}{\alpha - 2\epsilon}\mathbf{I} \succeq \frac{\epsilon}{\alpha - 2\epsilon}\mathbf{B}\mathbf{B}^\top \succeq \epsilon\mathbf{B}\left(\mathbf{M} - 2\epsilon\mathbf{I}\right)^{-1}\mathbf{B}^\top \tag{50}$$

Therefore, we have, $2\mathbf{K} \succ \epsilon\mathbf{B}\left(\mathbf{M} - 2\epsilon\mathbf{I}\right)^{-1}\mathbf{B}^\top$ and the third condition required for $\epsilon$ is satisfied. This implies that there exists a range of $\epsilon$, given by,

$$0 < \epsilon < \min\left\{\min\left(t_i\sqrt{k_i}\right), \frac{\alpha}{2}, \min\left(\frac{2k_i\alpha}{\beta + 4k_i}\right)\right\} \tag{51}$$

for which

$$\mathbf{P} = \begin{bmatrix} \mathbf{T}\mathbf{K} & \epsilon\mathbf{I} \\ \epsilon\mathbf{I} & \mathbf{T} \end{bmatrix} \tag{52}$$

is a valid Lyapunov function for asymptotic stability. Therefore, the dynamical system is globally asymptotically stable.

$\square$

## B.1 Positive definite block matrices (Schur complement)

For a symmetric block matrix of the form,

$$\mathbf{P} = \begin{bmatrix} \mathbf{A} & \mathbf{B} \\ \mathbf{B}^\top & \mathbf{C} \end{bmatrix} \tag{53}$$

with $\mathbf{A} = \mathbf{A}^\top$ and $\mathbf{C} = \mathbf{C}^\top$. If $\mathbf{C}$ is invertible the following two properties hold, [90],

- $\mathbf{P} \succ 0$ if and only if $\mathbf{C} \succ 0$ and $\mathbf{A} - \mathbf{B}\mathbf{C}^{-1}\mathbf{B}^\top \succ 0$.
- If $\mathbf{C} \succ 0$, then $\mathbf{P} \succeq 0$ if and only if $\mathbf{A} - \mathbf{B}\mathbf{C}^{-1}\mathbf{B}^\top \succeq 0$.

## B.2 Lyapunov stability criteria

Consider a non-linear autonomous dynamical system defined as $\dot{\mathbf{x}} = \mathbf{f}(\mathbf{x})$, with a point of equilibrium at $\mathbf{x} = \mathbf{0}$. Where $\mathbf{x} \in \mathcal{D} \subseteq \mathbb{R}^n$ is the system state vector and $\mathbf{f}(\mathbf{x}) : \mathcal{D} \to \mathbb{R}^n$ is a continuous vector field on $\mathcal{D}$ (contains origin). The dynamical system is called Lyapunov stable if there exists a real scalar function $V(\mathbf{x}) : \mathbb{R}^n \to \mathbb{R}$, also known as the Lyapunov function, such that it satisfies the following conditions,

- $V(\mathbf{x}) = 0$, if and only if $\mathbf{x} = \mathbf{0}$.
- $V(\mathbf{x}) > 0$, if and only if $\mathbf{x} \neq \mathbf{0}$.
- $\dot{V}(\mathbf{x}) \leq 0, \forall \mathbf{x} \neq \mathbf{0}$. Note that for asymptotic stability, we require the strict inequality $\dot{V}(\mathbf{x}) < 0$.

For a linear dynamical system of the form $\dot{\mathbf{x}} = \mathbf{J}\mathbf{x}$, where $\mathbf{J} \in \mathbb{R}^{n\times n}$, with a point of equilibrium at $\mathbf{x} = \mathbf{0}$. Consider a Lyapunov function $V(\mathbf{x})$ of the form $\mathbf{x}^\top\mathbf{P}\mathbf{x}$, such that $\mathbf{P} \succ 0$. By the definition of a positive definite matrix, it satisfies the first two conditions, namely, $V(\mathbf{x}) = \mathbf{x}^\top\mathbf{P}\mathbf{x} = 0$ when $\mathbf{x} = \mathbf{0}$ and $V(\mathbf{x}) = \mathbf{x}^\top\mathbf{P}\mathbf{x} > 0$ when $x \neq \mathbf{0}$. For the third condition, consider $\dot{V}(\mathbf{x})$,

$$\begin{aligned} \dot{V}(\mathbf{x}) &= \frac{\mathrm{d}}{\mathrm{d}t}V(\mathbf{x}) \\ &= \frac{\mathrm{d}}{\mathrm{d}t}\mathbf{x}^\top\mathbf{P}\mathbf{x} \\ &= \mathbf{x}^\top\mathbf{P}\dot{\mathbf{x}} + \dot{\mathbf{x}}^\top\mathbf{P}\mathbf{x} \\ &= \mathbf{x}^\top\left(\mathbf{P}\mathbf{J} + \mathbf{J}^\top\mathbf{P}\right)\mathbf{x} \end{aligned} \tag{54}$$

For stability, We need $\dot{V}(\mathbf{x}) = \mathbf{x}^\top\left(\mathbf{P}\mathbf{J} + \mathbf{J}^\top\mathbf{P}\right)\mathbf{x} \leq 0$. This is satisfied when the matrix $\mathbf{P}\mathbf{J} + \mathbf{J}^\top\mathbf{P} \preceq 0$. In summary, a linear dynamical system $\dot{\mathbf{x}} = \mathbf{J}\mathbf{x}$ is Lyapunov stable if there exists a positive definite matrix $\mathbf{P}$, such that $\mathbf{P}\mathbf{J} + \mathbf{J}^\top\mathbf{P}$ is negative semi-definite.

## C Analytical eigenvalue for the fully normalized circuit

Here we show that when all of the normalization weights in the system are equal, to value $\alpha$, and the various parameters are scalars, i.e., $\boldsymbol{\tau}_y = \tau_y \mathbf{1}$, $\boldsymbol{\tau}_a = \tau_a \mathbf{1}$, $\mathbf{b}_0 = b_0 \mathbf{1}$ and $\boldsymbol{\sigma} = \sigma \mathbf{1}$, we can derive a closed-form analytical expression for all of the eigenvalues. Considering these assumptions, we can break the determinant in Eq. 8 into a diagonal and non-diagonal part as follows,

$$\det(\mathbf{J} - \lambda \mathbf{I}) = \det\left( \lambda^2 \mathbf{I} + \lambda \left[ \frac{\mathbf{I}}{\tau_a} + \frac{\mathbf{D}\left(\sqrt{\mathbf{a}_s}\right)}{\tau_y} \right] + \frac{\mathbf{D}\left(\sqrt{\mathbf{a}_s}\right)}{\tau_y \tau_a} - \lambda \frac{\mathbf{W}\mathbf{D}\left(\mathbf{y}_s{}^2\right)}{\tau_a} \right) \tag{55}$$

Consider the non-diagonal part of the matrix in the determinant, $(\lambda/\tau_a)\mathbf{W}\mathbf{D}\left(\mathbf{y}_s^2\right)$. Since $\mathbf{W}$ is a matrix with all entries equal to a positive constant, $\alpha$, it is rank 1. Therefore, it can be written as the following outer product,

$$\mathbf{W} = \alpha \begin{bmatrix} 1 \\ 1 \\ \vdots \\ 1 \end{bmatrix} \begin{bmatrix} 1 & 1 & \cdots & 1 \end{bmatrix} \tag{56}$$

Therefore, the non-diagonal part of the matrix can be written as,

$$\frac{\lambda}{\tau_a} \mathbf{W}\, \mathbf{D}\left( \frac{\mathbf{b}^2 \odot \mathbf{z}^2}{\sigma^2 b_0^2 \mathbf{1} + \mathbf{W}\left(\mathbf{b}^2 \odot \mathbf{z}^2\right)} \right) = \frac{\lambda \alpha}{\tau_a} \mathbf{u}\mathbf{v}^\top \tag{57}$$

where $\mathbf{u} = [1, 1, ..., 1]^\top$ and $\mathbf{v} = \left(\mathbf{b}^2 \odot \mathbf{z}^2\right) / \left(\sigma^2 b_0^2 \mathbf{1} + \mathbf{W}\left(\mathbf{b}^2 \odot \mathbf{z}^2\right)\right)$. We use the matrix determinant lemma which states that,

$$\det(\mathbf{A} - \gamma \mathbf{u}\mathbf{v}^\top) = (1 - \gamma \mathbf{v}^\top \mathbf{A}^{-1} \mathbf{u}) \det(\mathbf{A}) \tag{58}$$

The matrix $\mathbf{A}$ is given by,

$$\begin{aligned} \mathbf{A} &= \lambda^2 \mathbf{I} + \lambda \left[ \frac{\mathbf{I}}{\tau_a} + \frac{\mathbf{D}\left(\sqrt{\mathbf{a}_s}\right)}{\tau_y} \right] + \frac{\mathbf{D}\left(\sqrt{\mathbf{a}_s}\right)}{\tau_y \tau_a} \\ &= \lambda^2 \mathbf{I} + \lambda \left[ \frac{\mathbf{I}}{\tau_a} + \frac{\mathbf{D}\left(\sqrt{\sigma^2 b_0^2 \mathbf{1} + \mathbf{W}\left(\mathbf{b}^2 \odot \mathbf{z}^2\right)}\right)}{\tau_y} \right] + \frac{\mathbf{D}\left(\sqrt{\sigma^2 b_0^2 \mathbf{1} + \mathbf{W}\left(\mathbf{b}^2 \odot \mathbf{z}^2\right)}\right)}{\tau_y \tau_a} \\ &= \lambda^2 \mathbf{I} + \lambda \left[ \frac{\mathbf{I}}{\tau_a} + \frac{\mathbf{D}\left(\sqrt{\sigma^2 b_0^2 \mathbf{1} + \alpha ||\mathbf{b} \odot \mathbf{z}||^2}\right)}{\tau_y} \right] + \frac{\mathbf{D}\left(\sqrt{\sigma^2 b_0^2 \mathbf{1} + \alpha ||\mathbf{b} \odot \mathbf{z}||^2}\right)}{\tau_y \tau_a} \\ &= \left( \lambda^2 + \lambda \left( \frac{1}{\tau_a} + \frac{\sqrt{\sigma^2 b_0^2 + \alpha ||\mathbf{b} \odot \mathbf{z}||^2}}{\tau_y} \right) + \frac{\sqrt{\sigma^2 b_0^2 + \alpha ||\mathbf{b} \odot \mathbf{z}||^2}}{\tau_y \tau_a} \right) \mathbf{I} \end{aligned} \tag{59}$$

Here, $||x||$, represents the Euclidean norm of $x$. Therefore, $\mathbf{A} = \delta \mathbf{I}$, where $\delta$ is a quadratic scalar polynomial in $\lambda$. Now using Eq. 58, we can write

$$\begin{aligned} \det(\mathbf{J} - \lambda \mathbf{I}) &= \left( 1 - \frac{\lambda \alpha}{\tau_a} \mathbf{v}^\top \left( \frac{1}{\delta}\mathbf{I} \right) \mathbf{u} \right) \delta^n \\ &= \left( \delta - \frac{\lambda}{\tau_a} \frac{\alpha ||\mathbf{b} \odot \mathbf{z}||^2}{\sigma^2 b_0^2 + \alpha ||\mathbf{b} \odot \mathbf{z}||^2} \right) \delta^{n-1} \end{aligned} \tag{60}$$

Solving for the eigenvalues, $\det(\mathbf{J} - \lambda \mathbf{I}) = 0$, we get $2(n-1)$ repeated solutions by equation $\delta^{n-1} = 0$, where each solution is found by solving $\delta = 0$,

$$\lambda^2 + \lambda \left( \frac{1}{\tau_a} + \frac{\sqrt{\sigma^2 b_0^2 + \alpha ||\mathbf{b} \odot \mathbf{z}||^2}}{\tau_y} \right) + \frac{\sqrt{\sigma^2 b_0^2 + \alpha ||\mathbf{b} \odot \mathbf{z}||^2}}{\tau_y \tau_a} = 0 \tag{61}$$

This gives us the following strictly negative eigenvalues,

$$\lambda = -\frac{1}{\tau_a} \quad \& \quad \lambda = -\frac{\sqrt{\sigma^2 b_0^2 + \alpha ||\mathbf{b} \odot \mathbf{z}||^2}}{\tau_y} \tag{62}$$

The potentially complex eigenvalues are given by solving for zeroes of the first part of the factorized determinant,

$$\lambda^2 + \lambda\left(\frac{\sigma^2 b_0^2}{\tau_a(\sigma^2 b_0^2 + \alpha||\mathbf{b}\odot\mathbf{z}||^2)} + \frac{\sqrt{\sigma^2 b_0^2 + \alpha||\mathbf{b}\odot\mathbf{z}||^2}}{\tau_y}\right) + \frac{\sqrt{\sigma^2 b_0^2 + \alpha||\mathbf{b}\odot\mathbf{z}||^2}}{\tau_y\tau_a} = 0 \quad (63)$$

The eigenvalues found by solving this equation have negative real parts (as expected) for all choices of parameters since the coefficient of $\lambda$ and the constant term of the quadratic equation are positive for all choices of parameters and inputs. Therefore, this solution does admit complex roots (thereby oscillations) for certain choices of the parameters which can be found by solving this quadratic equation.

## D   Convergent theorem

**Theorem D.1.** *A matrix $\mathbf{A}$ of the form,*

$$\mathbf{A} = \mathbf{D}(\mathbf{t})\,\mathbf{W}\,\mathbf{D}\left(\frac{\mathbf{u}}{\mathbf{v} + \mathbf{Wu}}\right) \quad (64)$$

*is convergent, i.e., its spectral radius is less than one, if $\mathbf{W} \in \mathbb{R}^{n\times n}$ and $\mathbf{t}, \mathbf{u}, \mathbf{v} \in \mathbb{R}^n$ with the additional constraints $0 < t_i < 1$, $u_i \geq 0$, $v_i > 0$ and $w_{ij} \geq 0$ for all $i, j$.*

*Proof.* Assuming the constraints mentioned in the theorem, we first notice that,

$$\mathbf{W}\,\mathbf{D}\left(\frac{\mathbf{u}}{\mathbf{v} + \mathbf{Wu}}\right) \quad (65)$$

is a nonnegative matrix. Further, because $\mathbf{D}(\mathbf{t})$ is a positive diagonal matrix with all entries less than 1, we notice that the following is true element-wise,

$$\mathbf{D}(\mathbf{t})\,\mathbf{W}\,\mathbf{D}\left(\frac{\mathbf{u}}{\mathbf{v} + \mathbf{Wu}}\right) < \mathbf{W}\,\mathbf{D}\left(\frac{\mathbf{u}}{\mathbf{v} + \mathbf{Wu}}\right) \quad (66)$$

We denote the spectral radius of $\mathbf{A}$, $\max\{|\lambda| : \lambda = \sigma(\mathbf{A})\}$, by $\rho(\mathbf{A})$, where $\sigma(\mathbf{A})$ is the spectrum of $\mathbf{A}$ and make use of the following inequality for the spectral radius of nonnegative matrices.,

**Theorem D.2.** *(Theorem 8.1.18 from [91]) Let $\mathbf{X}$ and $\mathbf{Y}$ be nonnegative matrices with spectral radius $\rho(\mathbf{X})$ and $\rho(\mathbf{Y})$, respectively. If $\mathbf{X} \leq \mathbf{Y}$ ($x_{ij} \leq y_{ij}, \forall\, i, j$), then $\rho(\mathbf{X}) \leq \rho(\mathbf{Y})$.*

Therefore, we have,

$$\rho(\mathbf{A}) \leq \rho\left(\mathbf{W}\,\mathbf{D}\left(\frac{\mathbf{u}}{\mathbf{v} + \mathbf{Wu}}\right)\right) \quad (67)$$

Now the elements of the matrix are given by,

$$\mathbf{W}\,\mathbf{D}\left(\frac{\mathbf{u}}{\mathbf{v} + \mathbf{Wu}}\right) = \begin{bmatrix} w_{11} & w_{12} & \cdots \\ w_{21} & w_{22} & \cdots \\ \vdots & \vdots & \ddots \end{bmatrix} \begin{bmatrix} \frac{u_1}{v_1 + \sum_j w_{1j}u_j} & 0 & \cdots \\ 0 & \frac{u_2}{v_2 + \sum_j w_{2j}u_j} & \cdots \\ \vdots & \vdots & \ddots \end{bmatrix} \quad (68)$$

Let $s_{ij}$ be the $i, j$ element of this matrix. Upon multiplication of the matrices above, we find that,

$$s_{ij} = \frac{w_{ij}u_j}{v_j + \sum_k w_{jk}u_k} \quad (69)$$

Now we use the following theorem that provides an upper bound for the spectral radius based on the entries of the matrix.

**Theorem D.3.** *(Theorem 8.1.26 from [91]) Let $\mathbf{S}$ be a nonnegative matrix. Then for any positive vector $\mathbf{x} \in \mathbb{R}^n$ with entries $x_j$, we have,*

$$\rho(\mathbf{S}) \leq \max_{1\leq i\leq n} \frac{1}{x_i} \sum_{j=1}^n s_{ij}x_j \quad (70)$$

We pick $x_j = v_j + \sum_k w_{jk} u_k$, which is a positive vector, and apply the theorem to the matrix. Substituting $s_{ij}$ and $x_j$, we get the following bound for the spectral radius,

$$\rho\left(\mathbf{W}\,\mathbf{D}\left(\frac{\mathbf{u}}{\mathbf{v}+\mathbf{Wu}}\right)\right) \leq \max_{1\leq i\leq n} \frac{1}{v_i+\sum_k w_{ik}u_k} \sum_{j=1}^{n} \frac{w_{ij}u_j}{v_j+\sum_k w_{jk}u_k}\left(v_j+\sum_k w_{jk}u_k\right) \quad (71)$$

Upon simplification, we get,

$$\rho\left(\mathbf{W}\,\mathbf{D}\left(\frac{\mathbf{u}}{\mathbf{v}+\mathbf{Wu}}\right)\right) \leq \max_{1\leq i\leq n} \frac{\sum_j w_{ij}u_j}{v_i+\sum_k w_{ik}u_k} = \max_{1\leq i\leq n} \frac{\sum_k w_{ik}u_k}{v_i+\sum_k w_{ik}u_k} < 1 \quad (72)$$

The inequality is true, because, for all $i$, the denominator is larger than the numerator because of the extra positive term $v_i$. Therefore,

$$\rho\left(\mathbf{A}\right) \leq \rho\left(\mathbf{W}\,\mathbf{D}\left(\frac{\mathbf{u}}{\mathbf{v}+\mathbf{Wu}}\right)\right) < 1 \quad (73)$$

and $\mathbf{A}$ is a convergent matrix. $\qquad\qquad\square$

## E   Linear stability analysis of the two-dimensional model

Here we consider the dynamical stability of the 2D model containing one neuron each of $y$ and $a$ when the recurrent scalar, $w_r$ can take any positive value. The dynamical system to be analyzed is given by,

$$\begin{aligned}
\tau_y \dot{y} &= -y + bz + \left(1 - \sqrt{\lfloor a \rfloor}\right) w_r y \\
\tau_a \dot{a} &= -a + b_0^2 \sigma^2 + w y^2 \lfloor a \rfloor
\end{aligned} \quad (74)$$

with a positive real constraint on the following set of parameters, $\tau_y, \tau_a, b, b_0, \sigma, w$. We first notice the symmetry in the dynamical system about $y = 0$. Replacing, $\hat{y} \rightarrow -y$, we get the mirrored dynamical system,

$$\begin{aligned}
\tau_{\hat{y}} \dot{\hat{y}} &= -\hat{y} + b(-z) + \left(1 - \sqrt{\lfloor a \rfloor}\right) w_r \hat{y} \\
\tau_a \dot{a} &= -a + b_0^2 \sigma^2 + w \hat{y}^2 \lfloor a \rfloor
\end{aligned} \quad (75)$$

These equations are the same as Eq. 74, up to a sign change in $z$. Therefore, we can derive analogous conditions for stability for $z < 0$ once the conditions for $z > 0$ are known.

The steady-state of Eq. 74, $(y_s, a_s)$ satisfies the following equations,

$$y_s = bz + \left(1 - \sqrt{\lfloor a_s \rfloor}\right) w_r y_s \quad (76)$$

$$a_s = b_0^2 \sigma^2 + w y_s^2 \lfloor a_s \rfloor \quad (77)$$

Since the RHS of the second equation is always positive, if a root exists, we have $a_s > 0$. Therefore, we can remove the rectification around $a_s$ and look for positive solutions for $a_s$. Upon rearranging the terms, we get,

$$(1 - w_r + w_r \sqrt{a_s})\, y_s = bz \quad (78)$$

$$(1 - w y_s^2)\, a_s = b_0^2 \sigma^2 \quad (79)$$

Substituting $y_s$ from Eq. 78 in Eq. 79, we get the following quartic equation in $m = \sqrt{a_s}$,

$$\begin{aligned}
w_r{}^2 m^4 + 2(1-w_r)w_r m^3 + \left((1-w_r)^2 - wb^2z^2 - b_0^2\sigma^2 w_r{}^2\right) m^2 \\
- 2(1-w_r)w_r b_0^2\sigma^2 m - (1-w_r)^2 b_0^2\sigma^2 = 0
\end{aligned} \quad (80)$$

For a valid fixed point $(y_s, a_s)$, i.e., $y_s \in \mathbb{R}$ and $a_s \in \mathbb{R}_*^+$, a necessary condition is that $a_s$ must be positive and real, which in turn implies $m$ must be positive and real.

**Theorem E.1.** *A fixed point, $(y_s, a_s)$, is valid if and only if it satisfies $\sqrt{a_s} > 0$.*

*Proof.* Due to Lemma E.2, we have: a fixed point, $(y_s, a_s)$, is valid if and only if it satisfies $m > b_0\sigma$. Further due to Lemma E.3, we have that the conditions $m > 0$ and $m > b_0\sigma$ are equivalent. Combining these two we get the statement of the theorem. $\qquad\square$

**Lemma E.2.** *A fixed point, $(y_s, a_s)$, is valid if and only if it satisfies $\sqrt{a_s} > b_0\sigma$.*

*Proof.* $\implies$ Given, $m = \sqrt{a_s} > b_0\sigma$, we have $a_s > b_0^2\sigma^2$, therefore, $a_s \in \mathbb{R}_*^+$; and $y_s = \sqrt{m^2 - b_0^2\sigma^2}/(m\sqrt{w})$, therefore, $y_s \in \mathbb{R}$. $\impliedby$ Given $a_s \in \mathbb{R}_*^+$ and $y_s \in \mathbb{R}$, implies $m \in \mathbb{R}_*^+$. Now, Eq. 79 posits that, $m = b_0\sigma/\sqrt{1 - wy_s^2}$, therefore, $m > b_0\sigma$. $\qquad\square$

**Lemma E.3.** *For a positive real root, $m = \sqrt{a_s}$, to the quartic equation (Eq. 80), the condition $m > 0$ is equivalent to the condition $m > b_0\sigma$. Further, no fixed point satisfies $0 < m < b_0\sigma$.*

*Proof.* $\implies$ Given $m > b_0\sigma$, we have $m > 0$ because $b_0 > 0$ and $\sigma > 0$. $\impliedby$ Since $m$ is a root of the quartic, it also satisfies Eq. 79. Therefore, we have $m = b_0\sigma/\sqrt{1 - wy_s^2}$. Since we are given $m > 0$, this implies that $0 < 1 - wy_s^2 < 1$, or $m > b_0\sigma$. Therefore, $m > 0$ is equivalent to $m > b_0\sigma$. This also implies that there exists no fixed point that satisfies $0 < m < b_0\sigma$. $\qquad\square$

Further, the Jacobian matrix at the fixed point of the dynamical system, in terms of the parameters and the fixed point is given by,

$$\mathbf{J} = \begin{bmatrix} \frac{w_r - 1 - w_r\sqrt{a_s}}{\tau_y} & -\frac{w_r y_s}{2\sqrt{a_s}\tau_y} \\ \frac{2wa_s y_s}{\tau_a} & \frac{-1 + wy_s^2}{\tau_a} \end{bmatrix} \tag{81}$$

From the linear stability theory, we know that a fixed point $(y_s, a_s)$ is asymptotically stable when the real part of the eigenvalues of $\mathbf{J}$ are less than 0, i.e., $\mathrm{Re}(\lambda_{\mathbf{J}}) < 0$. For a 2D system, this is equivalent to the conditions: $\mathrm{Tr}(\mathbf{J}) < 0$ and $\det(\mathbf{J}) > 0$. The trace of the Jacobian matrix is given by,

$$\begin{aligned} \mathrm{Tr}(\mathbf{J}) &= \frac{w_r - 1 - w_r\sqrt{a_s}}{\tau_y} + \frac{-1 + wy_s^2}{\tau_a} \\ &= -\left( \frac{1 - w_r + w_r\sqrt{a_s}}{\tau_y} + \frac{1 - wy_s^2}{\tau_a} \right) \\ &= -\left( \frac{1 - w_r + w_r\sqrt{a_s}}{\tau_y} + \frac{b_0^2\sigma^2}{a_s\tau_a} \right) \end{aligned} \tag{82}$$

The determinant of the Jacobian matrix is given by,

$$\begin{aligned} \det(\mathbf{J}) &= \left( \frac{w_r - 1 - w_r\sqrt{a_s}}{\tau_y} \right)\left( \frac{-1 + wy_s^2}{\tau_a} \right) + \left( \frac{w_r y_s}{2\sqrt{a_s}\tau_y} \right)\left( \frac{2wa_s y_s}{\tau_a} \right) \\ &= \frac{(1 - w_r)(1 - wy_s^2)}{\tau_a\tau_y} + \frac{w_r\sqrt{a_s}}{\tau_a\tau_y} \\ &= \frac{1}{\tau_y\tau_a}\left( \frac{(1 - w_r)b_0^2\sigma^2}{a_s} + w_r\sqrt{a_s} \right) \end{aligned} \tag{83}$$

Now we consider the different conditions on $w_r$ and the input drive $z$ for stability and state the various cases as theorems along with their proofs,

### E.1 Contractive constraint on recurrence ($0 < w_r \leq 1$)

- $z > 0$ : *There exists a unique fixed point with $y_s > 0$ and $a_s > 0$ and it is asymptotically stable.*

  *Proof.* The existence and uniqueness of the fixed point, along with its stability properties, are established by Lemma E.4. Additionally, the positivity of $a_s$ (i.e., $a_s > 0$) ensures that the expression $1 - w_r + w_r\sqrt{a_s} > 0$. Consequently, given that $z > 0$, it follows from Eq. 78 that $y_s > 0$. $\qquad\square$

- $z < 0$ : *There exists a unique fixed point with $y_s < 0$ and $a_s > 0$ and it is asymptotically stable.*

  *Proof.* Since this condition becomes equivalent to $z > 0$, up to a sign change in $y$ (Eq. 75), it is straightforward to see why this is true. □

**Lemma E.4.** *Given $0 < w_r \leq 1$ and $z \in \mathbb{R}$, there exists a unique fixed point and it is asymptotically stable.*

*Proof.* We observe that given $0 < w_r \leq 1$, if there exists a valid fixed point, which satisfies $a_s > 0$, $\text{Tr}(\mathbf{J})$ in Eq. 82 is less than 0 (since $1 - w_r + w_r\sqrt{a_s} > 0$ for any $\sqrt{a_s} > 0$), and $\det(\mathbf{J})$ in Eq. 83 is greater than 0 for all combinations of parameters and $z \in \mathbb{R}$. Therefore, we need to find the constraints on the parameters that allow for at least one fixed point. The fixed point, $m = \sqrt{a_s}$, satisfies the quartic polynomial in Eq 80. Due to Theorem E.1, for a valid fixed point, we need to find positive real roots that satisfy $m > 0$.

The sequence of signs of the coefficients of the polynomial when $0 \leq w_r < 1$ is given by $(+, +, \pm, -, -)$. We use *Descartes' Rule of Signs*, which states the following: The number of positive real roots of a polynomial $p(x)$ is either equal to the number of sign changes (omitting the zero coefficients) between consecutive non-zero coefficients of $p(x)$, or it is less than this by a multiple of 2. Since there is exactly one sign change from left to right in the sequence, regardless of the sign of the coefficient of $m^2$, we know that the equation above has exactly one real positive root for $m$. This further implies that $a_s$ has exactly one positive root and the corresponding fixed point $(y_s, a_s)$, is locally dynamically stable for all of the combinations of parameters and $z \in \mathbb{R}$.

Note that the result also holds for $w_r = 1$. There is exactly one positive fixed point $(y_s, a_s)$ and it is given by, i.e., $y_s = bz/\sqrt{b_0^2\sigma^2 + wb^2z^2}$ and $a_s = b_0^2\sigma^2 + wb^2z^2$. □

### E.2 Expansive constraint on recurrence ($w_r > 1$)

Now we consider the case when $w_r > 1$. We first find an alternative form of the determinant of the Jacobian (Eq.83) that is more amenable to the case, $w_r > 1$,

$$
\begin{aligned}
\det(\mathbf{J}) &= \frac{1}{\tau_y\tau_a}\left((1 - w_r)(1 - wy_s^2) + w_r\sqrt{a_s}\right) \\
&= \frac{1}{\tau_y\tau_a}\left(1 - w_r + w_r\sqrt{a_s} + (w_r - 1)wy_s^2\right)
\end{aligned}
\tag{84}
$$

Rewriting the trace,

$$
\text{Tr}(\mathbf{J}) = -\left(\frac{1 - w_r + w_r\sqrt{a_s}}{\tau_y} + \frac{b_0^2\sigma^2}{a_s\tau_a}\right)
\tag{85}
$$

The properties of stability can be summarized in the following two cases,

- $z > 0$ : *There exists exactly one fixed point with $y_s > 0$ and $a_s > 0$ and it is asymptotically stable. Further, if $b_0\sigma > 1 - 1/w_r$, there exist no additional fixed points. But if $b_0\sigma < 1 - 1/w_r$, then there exist either two or no fixed points with $y_s < 0$ and $a_s > 0$ and they may or may not be stable.*

  *Proof.* Since the fixed point satisfies Eq. 78, there are two possibilities, either $1 - w_r + w_r\sqrt{a_s} > 0$ or $1 - w_r + w_r\sqrt{a_s} < 0$. We consider them separately,
  
  ▶ $1 - w_r + w_r\sqrt{a_s} > 0$ : If $z > 0$, we have $y_s > 0$ because $y_s$ satisfies Eq. 78. Further due to Lemma E.5, this fixed point is unique and asymptotically stable for all combinations of parameters.
  
  ▶ $1 - w_r + w_r\sqrt{a_s} < 0$ : If $z > 0$, we have $y_s < 0$ because $y_s$ satisfies Eq. 78. Further, if we are given $b_0\sigma > 1 - 1/w_r$, no root exists. We prove this by contradiction, assuming a root exists that satisfies $\sqrt{a_s} < 1 - 1/w_r$, since $b_0\sigma > 1 - 1/w_r$, this root satisfies $0 < \sqrt{a_s} < b_0\sigma$, but due to Lemma E.3 no such root exists, therefore we have a contradiction.
  Now if we have $b_0\sigma < 1 - 1/w_r$, the root for $m = \sqrt{a_s}$ must satisfy $b_0\sigma < m < 1 - 1/w_r$. Since either one or three roots satisfy $m > 0$ (equivalently $m > b_0\sigma$) in

Eq. 80 and exactly one root satisfies $m > 1 - 1/w_r$ (Lemma E.5), we conclude that the number of roots for $m$ in the interval $(b_0\sigma, 1 - 1/w_r)$ are either two or none.

$\square$

- $z < 0$ : *There exists exactly one fixed point with $y_s < 0$ and $a_s > 0$ and it is asymptotically stable. Further, if $b_0\sigma > 1 - 1/w_r$, there exist no additional fixed points. But if $b_0\sigma < 1 - 1/w_r$, then there exist either two or no fixed points with $y_s > 0$ and $a_s > 0$ and they may or may not be stable.*

  *Proof.* Since this condition becomes equivalent to $z > 0$, up to a sign change in $y$ (Eq. 75), it is straightforward to see why this is true. $\square$

**Lemma E.5.** *Given $w_r > 1$, there exists a unique fixed point, $(y_s, a_s)$, that satisfies $1 - w_r + w_r\sqrt{a_s} > 0$ and it is asymptotically stable.*

*Proof.* The fixed point, $m = \sqrt{a_s}$, satisfies the quartic polynomial in Eq 80. Due to Theorem E.1, for a valid fixed point, we need to find positive real roots that satisfy $m > 0$.

The sequence of signs of the coefficients of the polynomial is given by $(+, -, \pm, +, -)$. Regardless of the sign of the coefficient of $m^2$, using *Descartes' Rule of Signs*, we find that since there are 3 sign changes, there are either 1 or 3 positive real roots for $m$. Since, we are given $w_r > 1$ and $1 - w_r + w_r\sqrt{a_s} > 0$, these roots are valid only if they satisfy $1 - w_r + w_r m > 0$, or, $m > 1 - 1/w_r$. We make a transformation $m \to \hat{m} + (1 - 1/w_r)$ which gives us,

$$w_r{}^2\hat{m}^4 - 2(1 - w_r)w_r\hat{m}^3 + \left((1 - w_r)^2 - wb^2z^2 - b_0^2\sigma^2 w_r{}^2\right)\hat{m}^2 - 2wb^2z^2(1 - 1/w_r)\hat{m} - wb^2z^2(1 - 1/w_r)^2 = 0 \tag{86}$$

The number of valid roots is given by the number of positive real roots of this polynomial. The signs of the coefficients are given by, $(+, +, \pm, -, -)$. There is exactly one sign change, hence there is a unique fixed point that satisfies $1 - w_r + w_r\sqrt{a_s} > 0$.

Now we prove that this fixed point is asymptotically stable. It can be easily seen that $\text{Tr}(\mathbf{J}) < 0$ in Eq. 85 and $\det(\mathbf{J}) > 0$ in Eq. 84 when $1 - w_r + w_r\sqrt{a_s} > 0$ and $w_r > 1$. Therefore, this unique fixed point is asymptotically stable. $\square$

# F  Analysis of the Rectified model

Here, we present the stability results for the model with only the positive part of the complementary receptive fields present. The model is given by the following dynamical equations,

$$\begin{aligned}
\boldsymbol{\tau}_y \odot \dot{\mathbf{y}} &= -\mathbf{y} + \mathbf{b} \odot \mathbf{z} + \left(\mathbf{1} - \mathbf{a}^+\right) \odot \lfloor \mathbf{W}_r \mathbf{y} \rfloor \\
\boldsymbol{\tau}_a \odot \dot{\mathbf{a}} &= -\mathbf{a} + \mathbf{b}_0^2 \odot \boldsymbol{\sigma}^2 + \mathbf{W}\left(\mathbf{y}^+ \odot \mathbf{a}^{+2}\right)
\end{aligned} \tag{87}$$

We will follow the same procedure for stability analysis as we did for the main model and state the key steps in the various derivations for stability.

## F.1  Stability of the high-dimensional system

We analyze the stability of the system when $\mathbf{W}_r = \mathbf{I}$. Upon substituting $\mathbf{y}^+ \to \lfloor \mathbf{y} \rfloor^2$ and $\mathbf{a}^+ \to \sqrt{\lfloor \mathbf{a} \rfloor}$, we get the following dynamical system,

$$\begin{aligned}
\boldsymbol{\tau}_y \odot \dot{\mathbf{y}} &= -\mathbf{y} + \mathbf{b} \odot \mathbf{z} + \left(\mathbf{1} - \sqrt{\lfloor \mathbf{a} \rfloor}\right) \odot \lfloor \mathbf{y} \rfloor \\
\boldsymbol{\tau}_a \odot \dot{\mathbf{a}} &= -\mathbf{a} + \mathbf{b}_0^2 \odot \boldsymbol{\sigma}^2 + \mathbf{W}\left(\lfloor \mathbf{y} \rfloor^2 \odot \lfloor \mathbf{a} \rfloor\right)
\end{aligned} \tag{88}$$

The fixed point is given by,

$$\begin{aligned}
\mathbf{y}_s &= \frac{\lfloor \mathbf{b} \odot \mathbf{z} \rfloor}{\sqrt{\mathbf{b}_0^2 \odot \boldsymbol{\sigma}^2 + \mathbf{W}\lfloor \mathbf{b} \odot \mathbf{z} \rfloor^2}} - \lfloor -\mathbf{b} \odot \mathbf{z} \rfloor \\
\mathbf{a}_s &= \mathbf{b}_0^2 \odot \boldsymbol{\sigma}^2 + \mathbf{W}\lfloor \mathbf{b} \odot \mathbf{z} \rfloor^2
\end{aligned} \tag{89}$$

Also, $\lfloor \mathbf{y}_s \rfloor$ is given by,

$$\lfloor \mathbf{y}_s \rfloor = \frac{\lfloor \mathbf{b} \odot \mathbf{z} \rfloor}{\sqrt{\mathbf{b}_0^2 \odot \boldsymbol{\sigma}^2 + \mathbf{W} \lfloor \mathbf{b} \odot \mathbf{z} \rfloor^2}} \tag{90}$$

The Jacobian matrix, $\mathbf{J}$, about this fixed point is given by,

$$\mathbf{J} = \begin{bmatrix} -\mathbf{D}\left(\frac{\sqrt{\mathbf{a}_s} \odot \mathbf{1}_{\mathbf{z} \geq 0} + \mathbf{1}_{\mathbf{z} < 0}}{\boldsymbol{\tau}_y}\right) & -\mathbf{D}\left(\frac{\lfloor \mathbf{y}_s \rfloor}{2 \odot \sqrt{\mathbf{a}_s} \odot \boldsymbol{\tau}_y}\right) \\ \mathbf{D}\left(\frac{2}{\boldsymbol{\tau}_a}\right) \mathbf{W} \mathbf{D}\left(\mathbf{a}_s \odot \lfloor \mathbf{y}_s \rfloor\right) & \mathbf{D}\left(\frac{1}{\boldsymbol{\tau}_a}\right)\left(-\mathbf{I} + \mathbf{W} \mathbf{D}\left(\lfloor \mathbf{y}_s \rfloor^2\right)\right) \end{bmatrix} \tag{91}$$

Here $\mathbf{1}_{\mathbf{z} \geq 0}$ is an indicator function that returns a vector the size of $\mathbf{z}$ with 1 at locations where $z_i \geq 0$ and 0 elsewhere. Similarly, $\mathbf{1}_{\mathbf{z} < 0}$ returns a vector the size of $\mathbf{z}$ with 1 at locations where $z_i < 0$ and 0 elsewhere. Further $\mathbf{J} - \lambda \mathbf{I}$ is given by,

$$\begin{aligned} \mathbf{J} &= \begin{bmatrix} \mathbf{A}_{11} & \mathbf{A}_{12} \\ \mathbf{A}_{21} & \mathbf{A}_{22} \end{bmatrix} \\ &= \begin{bmatrix} -\mathbf{D}\left(\frac{\sqrt{\mathbf{a}_s} \odot \mathbf{1}_{\mathbf{z} \geq 0} + \mathbf{1}_{\mathbf{z} < 0}}{\boldsymbol{\tau}_y}\right) - \lambda \mathbf{I} & -\mathbf{D}\left(\frac{\lfloor \mathbf{y}_s \rfloor}{2 \odot \sqrt{\mathbf{a}_s} \odot \boldsymbol{\tau}_y}\right) \\ \mathbf{D}\left(\frac{2}{\boldsymbol{\tau}_a}\right) \mathbf{W} \mathbf{D}\left(\mathbf{a}_s \odot \lfloor \mathbf{y}_s \rfloor\right) & \mathbf{D}\left(\frac{1}{\boldsymbol{\tau}_a}\right)\left(-\mathbf{I} + \mathbf{W} \mathbf{D}\left(\lfloor \mathbf{y}_s \rfloor^2\right)\right) - \lambda \mathbf{I} \end{bmatrix} \end{aligned} \tag{92}$$

Since $\mathbf{A}_{11}$ and $\mathbf{A}_{12}$ commute, we can write, $\det(\mathbf{J} - \lambda \mathbf{I}) = \det(\mathbf{A}_{22}\mathbf{A}_{11} - \mathbf{A}_{21}\mathbf{A}_{12})$. Upon simplification, we get,

$$\begin{aligned} \det(\mathbf{J} - \lambda \mathbf{I}) = \det\Bigg(&\lambda^2 \mathbf{I} + \lambda \left[\mathbf{D}\left(\frac{1}{\boldsymbol{\tau}_a}\right) + \mathbf{D}\left(\frac{\sqrt{\mathbf{a}_s} \odot \mathbf{1}_{\mathbf{z} \geq 0} + \mathbf{1}_{\mathbf{z} < 0}}{\boldsymbol{\tau}_y}\right) \\ &- \mathbf{D}\left(\frac{1}{\boldsymbol{\tau}_a}\right) \mathbf{W} \mathbf{D}\left(\lfloor \mathbf{y}_s \rfloor^2\right)\right] + \mathbf{D}\left(\frac{\sqrt{\mathbf{a}_s} \odot \mathbf{1}_{\mathbf{z} \geq 0} + \mathbf{1}_{\mathbf{z} < 0}}{\boldsymbol{\tau}_y \odot \boldsymbol{\tau}_a}\right)\Bigg) \end{aligned} \tag{93}$$

Note that, here we used the fact that $\mathbf{y}_s$ has the same sign as $\mathbf{z}$, as seen from Eq. 89. The dynamical system is stable if all eigenvalues of this characteristic polynomial have negative real parts. Just like the main model, we map this to a quadratic eigenvalue problem of the form $\mathcal{L}(\lambda) = \det(\lambda^2 \mathbf{I} + \lambda \mathbf{B} + \mathbf{K}) = 0$. Now, we see if the conditions of Theorem 4.1 are met. The stiffness matrix is given by,

$$\mathbf{K} = \mathbf{D}\left(\frac{\sqrt{\mathbf{a}_s} \odot \mathbf{1}_{\mathbf{z} \geq 0} + \mathbf{1}_{\mathbf{z} < 0}}{\boldsymbol{\tau}_y \odot \boldsymbol{\tau}_a}\right) = \mathbf{D}\left(\frac{\sqrt{\mathbf{b}_0^2 \odot \boldsymbol{\sigma}^2 + \mathbf{W} \lfloor \mathbf{b} \odot \mathbf{z} \rfloor^2} \odot \mathbf{1}_{\mathbf{z} \geq 0} + \mathbf{1}_{\mathbf{z} < 0}}{\boldsymbol{\tau}_y \odot \boldsymbol{\tau}_a}\right) \tag{94}$$

Clearly, $\mathbf{K}$ is a positive diagonal matrix for all choices of parameters and input. Now we check if the $\mathbf{B}$ is a Lyapunov diagonally stable matrix, which is implied when $\mathbf{B}$ admits a *regular convergent splitting*, i.e., it has a representation of the form $\mathbf{B} = \mathbf{M} - \mathbf{N}$, where $\mathbf{M}^{-1}$ and $\mathbf{N}$ have all nonnegative entries and $\mathbf{M}^{-1}\mathbf{N}$ has a spectral radius smaller than 1.

$$\mathbf{B} = \underbrace{\mathbf{D}\left(\frac{1}{\boldsymbol{\tau}_a}\right) + \mathbf{D}\left(\frac{\sqrt{\mathbf{a}_s} \odot \mathbf{1}_{\mathbf{z} \geq 0} + \mathbf{1}_{\mathbf{z} < 0}}{\boldsymbol{\tau}_y}\right)}_{\mathbf{M}} - \underbrace{\mathbf{D}\left(\frac{1}{\boldsymbol{\tau}_a}\right) \mathbf{W} \mathbf{D}\left(\lfloor \mathbf{y}_s \rfloor^2\right)}_{\mathbf{N}} \tag{95}$$

Since $\mathbf{M}$ is a positive diagonal matrix, all the entries of $\mathbf{M}^{-1}$ are nonnegative. Also, since all the matrices involved in the definition of $\mathbf{N}$ are nonnegative, therefore their product, $\mathbf{N}$, is nonnegative. We are only left to prove that $\mathbf{M}^{-1}\mathbf{N} = \mathbf{S}$ has a spectral radius smaller than 1. Consider the matrix $\mathbf{S}$,

$$\mathbf{S} = \mathbf{D}\left(\frac{1}{1 + (\boldsymbol{\tau}_a/\boldsymbol{\tau}_y) \odot (\sqrt{\mathbf{a}_s} \odot \mathbf{1}_{\mathbf{z} \geq 0} + \mathbf{1}_{\mathbf{z} < 0})}\right) \mathbf{W} \mathbf{D}\left(\frac{\lfloor \mathbf{b} \odot \mathbf{z} \rfloor^2}{\mathbf{b}_0^2 \odot \boldsymbol{\sigma}^2 + \mathbf{W} \lfloor \mathbf{b} \odot \mathbf{z} \rfloor^2}\right) \tag{96}$$

Theorem D puts a bound of 1 on the spectral radius of this matrix. Define $\mathbf{t} \to 1/\left(1 + (\boldsymbol{\tau}_a/\boldsymbol{\tau}_y) \odot (\sqrt{\mathbf{a}_s} \odot \mathbf{1}_{\mathbf{z} \geq 0} + \mathbf{1}_{\mathbf{z} < 0})\right)$, $\mathbf{u} \to \lfloor \mathbf{b} \odot \mathbf{z} \rfloor^2$ and $\mathbf{v} \to \mathbf{b}_0^2 \odot \boldsymbol{\sigma}^2$, we notice that they follow the constraints of the theorem, therefore, $\mathbf{S}$ is convergent. This implies that $\mathbf{B}$ has a *convergent regular splitting*, therefore, the linearized dynamical system is unconditionally globally asymptotically stable (and nonlinear dynamical system is locally asymptotically stable) across all the values of parameters and inputs.

## F.2 Analytical eigenvalue for fully normalized circuit

Following a procedure similar to that in Appendix C, when all of the normalization weights in the system are equal, to value $\alpha$, and the various parameters are scalars, i.e., $\boldsymbol{\tau}_y = \tau_y \mathbf{1}$, $\boldsymbol{\tau}_a = \tau_a \mathbf{1}$, $\mathbf{b}_0 = b_0 \mathbf{1}$ and $\boldsymbol{\sigma} = \sigma \mathbf{1}$, we can write the analytical expressions for eigenvalues. The characteristic polynomial is given by,

$$\det(\mathbf{J} - \lambda \mathbf{I}) = \left( 1 - \frac{\lambda \alpha}{\tau_a} \mathbf{v}^\top \mathbf{D} \left( \frac{1}{\delta_1 \mathbf{1}_{\mathbf{z} \geq 0} + \delta_2 \mathbf{1}_{\mathbf{z} < 0}} \right) \mathbf{u} \right) \delta_1^{n_1} \delta_2^{n_2} \tag{97}$$

where, $\mathbf{u} = [1, 1, ..., 1]^\top$, $\mathbf{v} = \lfloor \mathbf{b} \odot \mathbf{z} \rfloor^2 / \left( \sigma^2 b_0^2 \mathbf{1} + \mathbf{W} \lfloor \mathbf{b} \odot \mathbf{z} \rfloor^2 \right)$; $n_1$ and $n_2$ are the number of nonnegative and negative values, respectively, in the input drive $\mathbf{z}$; $\delta_1$ and $\delta_2$ are given by,

$$\delta_1 = \lambda^2 + \lambda \left( \frac{1}{\tau_a} + \frac{\sqrt{\sigma^2 b_0^2 + \alpha || \lfloor \mathbf{b} \odot \mathbf{z} \rfloor ||^2}}{\tau_y} \right) + \frac{\sqrt{\sigma^2 b_0^2 + \alpha || \lfloor \mathbf{b} \odot \mathbf{z} \rfloor ||^2}}{\tau_y \tau_a}$$

$$\delta_2 = \lambda^2 + \lambda \left( \frac{1}{\tau_a} + \frac{1}{\tau_y} \right) + \frac{1}{\tau_y \tau_a} \tag{98}$$

Simplification of Eq. 97 gives us,

$$\det(\mathbf{J} - \lambda \mathbf{I}) = \left( \delta_1 - \frac{\lambda}{\tau_a} \frac{\alpha || \lfloor \mathbf{b} \odot \mathbf{z} \rfloor ||^2}{\sigma^2 b_0^2 + \alpha || \lfloor \mathbf{b} \odot \mathbf{z} \rfloor ||^2} \right) \delta_1^{n_1 - 1} \delta_2^{n_2} \tag{99}$$

Since the characteristic polynomial is a product of quadratic polynomials, we can solve them analytically. The strictly negative eigenvalues are given by,

$$\lambda = -\frac{1}{\tau_a}; \quad \lambda = -\frac{1}{\tau_y} \quad \& \quad \lambda = -\frac{\sqrt{\sigma^2 b_0^2 + \alpha || \lfloor \mathbf{b} \odot \mathbf{z} \rfloor ||^2}}{\tau_y} \tag{100}$$

The potentially complex eigenvalues are given by the solution to the following quadratic equation,

$$\lambda^2 + \lambda \left( \frac{\sigma^2 b_0^2}{\tau_a (\sigma^2 b_0^2 + \alpha || \lfloor \mathbf{b} \odot \mathbf{z} \rfloor ||^2)} + \frac{\sqrt{\sigma^2 b_0^2 + \alpha || \lfloor \mathbf{b} \odot \mathbf{z} \rfloor ||^2}}{\tau_y} \right) + \frac{\sqrt{\sigma^2 b_0^2 + \alpha || \lfloor \mathbf{b} \odot \mathbf{z} \rfloor ||^2}}{\tau_y \tau_a} = 0 \tag{101}$$

## F.3 Linear stability analysis of the two-dimensional model

The dynamical system to consider is,

$$\tau_y \dot{y} = -y + bz + \left( 1 - \sqrt{\lfloor a \rfloor} \right) \lfloor w_r y \rfloor$$

$$\tau_a \dot{a} = -a + b_0^2 \sigma^2 + w \lfloor y \rfloor^2 \lfloor a \rfloor \tag{102}$$

Since a valid fixed point must have $a_s > 0$, the fixed point $(y_s, a_s)$ satisfies,

$$(1 - w_r + w_r \sqrt{a_s}) \lfloor y_s \rfloor - \lfloor -y_s \rfloor = bz \tag{103}$$

$$(1 - w \lfloor y_s \rfloor^2) a_s = b_0^2 \sigma^2 \tag{104}$$

We divide this into two cases,

- $y_s > 0$ : The fixed point is given by the equations,

$$(1 - w_r + w_r \sqrt{a_s}) y_s = bz \tag{105}$$

$$(1 - w y_s^2) a_s = b_0^2 \sigma^2 \tag{106}$$

A fixed point, $(y_s, a_s)$, is valid only if it satisfies $y_s \in \mathbb{R}_*^+$ and $a_s \in \mathbb{R}_*^+$. The Jacobian matrix is given by,

$$\mathbf{J} = \begin{bmatrix} \frac{w_r - 1 - w_r \sqrt{a_s}}{\tau_y} & -\frac{w_r y_s}{2\sqrt{a_s} \tau_y} \\ \frac{2 w a_s y_s}{\tau_a} & \frac{-1 + w y_s^2}{\tau_a} \end{bmatrix} \tag{107}$$

Note that this is equivalent to the main model, with the additional constraint of $y_s > 0$, whose stability analysis is presented in Appendix E.

- $y_s < 0$ : Eq. 103 & 104 yield us a unique fixed point $y_s = bz$ and $a_s = b_0^2\sigma^2$. Since $y_s < 0$ and $y_s = bz$, this is only possible when $z < 0$. The Jacobian matrix about $(y_s, a_s)$ is given by,

$$\mathbf{J} = \begin{bmatrix} -\frac{1}{\tau_y} & 0 \\ 0 & -\frac{1}{\tau_a} \end{bmatrix} \tag{108}$$

Since the eigenvalues of $\mathbf{J}$ are $\lambda_{\mathbf{J}} = -1/\tau_y, -1/\tau_a$ are real and both negative, this fixed point is always stable.

Combining the cases above and results already established in Appendix E, we characterize the stability of the system as follows,

### F.3.1 Contractive constraint on recurrence ($0 < w_r \le 1$)

- $z > 0$ : *There exists a unique fixed point with $y_s > 0$ and $a_s > 0$ and it is asymptotically stable.*

- $z < 0$ : *There exists a unique fixed point with $y_s < 0$ and $a_s > 0$, given by, $(bz, b_0^2\sigma^2)$, and it is asymptotically stable.*

### F.3.2 Expansive constraint on recurrence ($w_r > 1$)

- $z > 0$ : *There exists a unique fixed point with $y_s > 0$ and $a_s > 0$ and it is asymptotically stable.*

- $z < 0$ : *There exists exactly one fixed point with $y_s < 0$ and $a_s > 0$ given by, $(bz, b_0^2\sigma^2)$, and it is asymptotically stable. Further, if $b_0\sigma > 1 - 1/w_r$, there exist no additional fixed points. But if $b_0\sigma < 1 - 1/w_r$, then there exist either two or no fixed points with $y_s > 0$ and $a_s > 0$ and they may or may not be stable.*

## G Iterative algorithm

In this section, we present an iterative approach to finding the fixed point for ORGaNICs with an arbitrary recurrent weight matrix. We show that this algorithm converges in a few steps (2-10) with great accuracy. We consider the system given by Eq. 15,

$$\boldsymbol{\tau}_y \odot \dot{\mathbf{y}} = -\mathbf{y} + \mathbf{b} \odot \mathbf{z} + \left(\mathbf{1} - \sqrt{\lfloor \mathbf{a} \rfloor}\right) \odot (\mathbf{W}_r \mathbf{y})$$
$$\boldsymbol{\tau}_a \odot \dot{\mathbf{a}} = -\mathbf{a} + \mathbf{b}_0^2 \odot \boldsymbol{\sigma}^2 + \mathbf{W} \left(\mathbf{y}^2 \odot \lfloor \mathbf{a} \rfloor\right) \tag{109}$$

The fixed point of this system ($\mathbf{y}_s$ and $\mathbf{a}_s$) is found by solving the following simultaneous equations,

$$\mathbf{y}_s = \mathbf{b} \odot \mathbf{z} + (\mathbf{1} - \sqrt{\mathbf{a}_s}) \odot (\mathbf{W}_r \mathbf{y}_s) \tag{110}$$

$$\mathbf{a}_s = \mathbf{b}_0^2 \odot \boldsymbol{\sigma}^2 + \mathbf{W} \left(\mathbf{y}_s^2 \odot \mathbf{a}_s\right) \tag{111}$$

These equations do not admit a closed-form analytical solution when $\mathbf{W}_r \ne \mathbf{I}$. We first find a good approximation for the initialization of $\mathbf{y}_s$ and $\mathbf{a}_s$ and then define the iterative algorithm. The equation for $\mathbf{y}_s$ can be written in terms of $\mathbf{a}_s$ as,

$$\mathbf{y}_s = (\mathbf{I} + (\mathbf{D}(\sqrt{\mathbf{a}_s}) - \mathbf{I})\mathbf{W}_r)^{-1} (\mathbf{b} \odot \mathbf{z}) \tag{112}$$

Now applying the Woodbury matrix identity, which states that

$$(\mathbf{A} + \mathbf{U}\mathbf{C}\mathbf{V})^{-1} = \mathbf{A}^{-1} - \mathbf{A}^{-1}\mathbf{U}\left(\mathbf{C}^{-1} + \mathbf{V}\mathbf{A}^{-1}\mathbf{U}\right)^{-1}\mathbf{V}\mathbf{A}^{-1}, \tag{113}$$

to the inverse in Eq. 112 with $\mathbf{A} = \mathbf{I}$, $\mathbf{U} = \mathbf{I}$, $\mathbf{C} = \mathbf{D}(\sqrt{\mathbf{a}_s}) - \mathbf{I}$ and $\mathbf{V} = \mathbf{W}_r$, we get,

$$\mathbf{y}_s = \left(\mathbf{I} - \left((\mathbf{D}(\sqrt{\mathbf{a}_s}) - \mathbf{I})^{-1} + \mathbf{W}_r\right)^{-1}\mathbf{W}_r\right)(\mathbf{b} \odot \mathbf{z}) \tag{114}$$

We approximate the above equation by assuming that $\mathbf{W}_r$ is a symmetric matrix with the eigende-composition given by $\mathbf{Q}\mathbf{\Lambda}\mathbf{Q}^\top$, with $\mathbf{Q}^\top\mathbf{Q} = \mathbf{Q}\mathbf{Q}^\top = \mathbf{I}$ and $\mathbf{\Lambda} = \mathbf{D}(\boldsymbol{\lambda})$ is a diagonal matrix with

the eigenvalues as its diagonal entries. This gives us the following approximation,

$$
\begin{aligned}
\mathbf{y}_s &\approx \left(\mathbf{I} - \left(\mathbf{Q}\left(\left(\mathbf{D}\left(\sqrt{\mathbf{a}_s}\right) - \mathbf{I}\right)^{-1} + \mathbf{\Lambda}\right)\mathbf{Q}^\top\right)^{-1}\mathbf{Q}\mathbf{\Lambda}\mathbf{Q}^\top\right)(\mathbf{b}\odot\mathbf{z}) \\
&= \left(\mathbf{I} - \mathbf{Q}\mathbf{D}\left(\boldsymbol{\lambda} + \frac{1}{\sqrt{\mathbf{a}_s} - 1}\right)^{-1}\mathbf{Q}^\top\mathbf{Q}\mathbf{\Lambda}\mathbf{Q}^\top\right)(\mathbf{b}\odot\mathbf{z}) \\
&= \left(\mathbf{I} - \mathbf{Q}\mathbf{D}\left(\frac{\boldsymbol{\lambda}*\sqrt{\mathbf{a}_s} - \boldsymbol{\lambda}}{1 - \boldsymbol{\lambda} + \boldsymbol{\lambda}*\sqrt{\mathbf{a}_s}}\right)\mathbf{Q}^\top\right)(\mathbf{b}\odot\mathbf{z}) \\
&= \left(\mathbf{I} - \mathbf{Q}\left(\mathbf{I} - \mathbf{D}\left(\frac{1}{1 - \boldsymbol{\lambda} + \boldsymbol{\lambda}*\sqrt{\mathbf{a}_s}}\right)\right)\mathbf{Q}^\top\right)(\mathbf{b}\odot\mathbf{z}) \\
&= \mathbf{Q}\mathbf{D}\left(\frac{1}{1 - \boldsymbol{\lambda} + \boldsymbol{\lambda}*\sqrt{\mathbf{a}_s}}\right)\mathbf{Q}^\top(\mathbf{b}\odot\mathbf{z}) \\
&= \mathbf{Q}\mathbf{D}\left(\frac{1}{\boldsymbol{\lambda} - \boldsymbol{\lambda}^2 + \boldsymbol{\lambda}^2*\sqrt{\mathbf{a}_s}}\right)\mathbf{\Lambda}\mathbf{Q}^\top(\mathbf{b}\odot\mathbf{z})
\end{aligned}
\tag{115}
$$

We approximate the eigenvalues, $\boldsymbol{\lambda}$, by the maximum eigenvalue of the $\mathbf{W}_r$ and assume the entries of $\sqrt{\mathbf{a}_s}$ are identical. This gives us the following initial guess for $\mathbf{y}_s$.

$$
\begin{aligned}
\mathbf{y}_s^0 &= \mathbf{D}\left(\frac{1}{\lambda_m - \lambda_m^2 + \lambda_m^2\sqrt{\mathbf{a}_s^0}}\right)\mathbf{Q}\mathbf{\Lambda}\mathbf{Q}^\top(\mathbf{b}\odot\mathbf{z}) \\
&= \frac{\mathbf{W}_r(\mathbf{b}\odot\mathbf{z})}{\lambda_m - \lambda_m^2 + \lambda_m^2\sqrt{\mathbf{a}_s^0}}
\end{aligned}
\tag{116}
$$

For the initial guess of $\mathbf{a}_s^0$, we use Eq. 111 and plug in the following on the RHS $\mathbf{a}_s \to \mathbf{b}_0^2\odot\boldsymbol{\sigma}^2$ and the corresponding $\mathbf{y}_s$ found by using Eq. 116. This gives us the following,

$$
\mathbf{a}_s^0 = \boldsymbol{\sigma}^2\odot\mathbf{b}_0^2 + \mathbf{W}\left(\left(\frac{\mathbf{W}_r(\mathbf{b}\odot\mathbf{z})}{\lambda_m - \lambda_m^2 + \lambda_m^2\sqrt{\mathbf{b}_0^2\odot\boldsymbol{\sigma}^2}}\right)^2\odot\left(\mathbf{b}_0^2\odot\boldsymbol{\sigma}^2\right)\right)
\tag{117}
$$

Next we update the $\mathbf{y}_s$ and $\mathbf{a}_s$ by performing the following iterations derived using Eq. 110 & 111. For instance, $(\mathbf{y}_s^1, \mathbf{a}_s^1)$ are given by,

$$
\begin{aligned}
\mathbf{y}_s^1 &= \left(\mathbf{I} - \mathbf{W}_r + \mathbf{D}\left(\sqrt{\mathbf{a}_s^0}\right)\mathbf{W}_r\right)^{-1}(\mathbf{b}\odot\mathbf{z}) \\
\mathbf{a}_s^1 &= \mathbf{b}_0^2\odot\boldsymbol{\sigma}^2 + \mathbf{W}\left(\left(\mathbf{y}_s^1\right)^2*\mathbf{a}_s^0\right)
\end{aligned}
\tag{118}
$$

This procedure is summarized in Algorithm 2. Substituting $\lambda_m = 1$ in Eq. 116 & 117 yields simpler initial conditions and gives us Algorithm 1. Even though we had assumed that $\mathbf{W}_r$ should be symmetric, in practice we find that this algorithm leads to fast convergence even for non-symmetric matrices. The fast convergence is owed to the fact that we have a good initial approximation of the solution. We also find that this iteration scheme works only for recurrent weight matrices with a maximum singular value of 1.

## H Energy of ORGaNICs

Here, we find the *energy* (Lyapunov function) that is minimized by the dynamics of the ORGaNICs in the vicinity of the normalization fixed point. We consider the dynamical system with $\mathbf{W}_r = \mathbf{I}$, which is given by Eq. 3. Upon linearizing about the fixed point we get the following linear dynamical system,

$$
\begin{bmatrix}\dot{\mathbf{y}} \\ \dot{\mathbf{a}}\end{bmatrix} = \begin{bmatrix}-\mathbf{D}\left(\frac{\sqrt{\mathbf{a}_s}}{\boldsymbol{\tau}_y}\right) & -\mathbf{D}\left(\frac{\mathbf{y}_s}{2\odot\sqrt{\mathbf{a}_s}\odot\boldsymbol{\tau}_y}\right) \\ \mathbf{D}\left(\frac{2}{\boldsymbol{\tau}_a}\right)\mathbf{W}\,\mathbf{D}\left(\mathbf{a}_s\odot\mathbf{y}_s\right) & \mathbf{D}\left(\frac{1}{\boldsymbol{\tau}_a}\right)\left(-\mathbf{I} + \mathbf{W}\,\mathbf{D}\left(\mathbf{y}_s^2\right)\right)\end{bmatrix}\begin{bmatrix}\mathbf{y} - \mathbf{y}_s \\ \mathbf{a} - \mathbf{a}_s\end{bmatrix}
\tag{119}
$$

---

**Algorithm 2** Iterative scheme for finding the fixed point

---

1: **Input:** ORGaNICs parameters and input $(\mathbf{z})$, Tolerance $\epsilon$, maximum iterations $N$

2: **Output:** Approximation to the fixed point $(\mathbf{y}_s, \mathbf{a}_s)$

3: $\mathbf{a} \leftarrow \boldsymbol{\sigma}^2 \odot \mathbf{b}_0^2 + \mathbf{W} \left( \left( \frac{\mathbf{W}_r(\mathbf{b} \odot \mathbf{z})}{\lambda_m - \lambda_m^2 + \lambda_m^2 \sqrt{\mathbf{b}_0^2 \odot \boldsymbol{\sigma}^2}} \right)^2 \odot \left( \mathbf{b}_0^2 \odot \boldsymbol{\sigma}^2 \right) \right)$  // initial approximation for $\mathbf{a}$

4: $\mathbf{y} \leftarrow \frac{\mathbf{W}_r(\mathbf{b} \odot \mathbf{z})}{\lambda_m - \lambda_m^2 + \lambda_m^2 \sqrt{\mathbf{a}}}$  // initial approximation for $\mathbf{y}$

5: $k \leftarrow 0$

6: **while** $\| \mathbf{y} - \mathbf{b} \odot \mathbf{z} - \left( \mathbf{1} - \sqrt{\mathbf{a}} \right) \odot (\mathbf{W}_r \mathbf{y}) \| > \epsilon$ and $k < N$ **do**

7:      $\mathbf{y} \leftarrow \left( \mathbf{I} - \mathbf{W}_r + \mathbf{D}\left(\sqrt{\mathbf{a}}\right)\mathbf{W}_r \right)^{-1} (\mathbf{b} \odot \mathbf{z})$  // $\mathbf{y}$ update

8:      $\mathbf{a} \leftarrow \mathbf{b}_0^2 \odot \boldsymbol{\sigma}^2 + \mathbf{W} \left( \mathbf{y}^2 * \mathbf{a} \right)$  // $\mathbf{a}$ update

9:      $k \leftarrow k + 1$

10: **end while**

11: **return** $(\mathbf{y}, \mathbf{a})$

---

This system is dynamically equivalent (admits the same eigenvalues) to a system of coupled harmonic oscillators with the following equations,

$$\ddot{\mathbf{x}} + \left[ \mathbf{D}\left(\frac{1}{\boldsymbol{\tau}_a}\right) + \mathbf{D}\left(\frac{\sqrt{\mathbf{a}_s}}{\boldsymbol{\tau}_y}\right) - \mathbf{D}\left(\frac{1}{\boldsymbol{\tau}_a}\right) \mathbf{W} \mathbf{D}\left(\mathbf{y}_s^2\right) \right] \dot{\mathbf{x}} + \mathbf{D}\left(\frac{\sqrt{\mathbf{a}_s}}{\boldsymbol{\tau}_y \odot \boldsymbol{\tau}_a}\right) \mathbf{x} = \mathbf{0}. \qquad (120)$$

We can rewrite the linear system in terms of the position, $\mathbf{x}$, and the velocity, $\mathbf{v}$,

$$\begin{bmatrix} \dot{\mathbf{x}} \\ \dot{\mathbf{v}} \end{bmatrix} = \begin{bmatrix} \mathbf{0} & \mathbf{I} \\ -\mathbf{D}\left(\frac{\sqrt{\mathbf{a}_s}}{\boldsymbol{\tau}_y \odot \boldsymbol{\tau}_a}\right) & -\left[ \mathbf{D}\left(\frac{1}{\boldsymbol{\tau}_a}\right) + \mathbf{D}\left(\frac{\sqrt{\mathbf{a}_s}}{\boldsymbol{\tau}_y}\right) - \mathbf{D}\left(\frac{1}{\boldsymbol{\tau}_a}\right) \mathbf{W} \mathbf{D}\left(\mathbf{y}_s^2\right) \right] \end{bmatrix} \begin{bmatrix} \mathbf{x} \\ \mathbf{v} \end{bmatrix} \qquad (121)$$

Since this system is of the form $\mathbf{I}\ddot{\mathbf{x}} + \mathbf{B}\dot{\mathbf{x}} + \mathbf{K}\mathbf{x} = \mathbf{0}$, Eq. 36, the *energy* of this dynamical system is given by $V(\mathbf{z}) = \mathbf{z}^\top \mathbf{P} \mathbf{z}$, or,

$$V(\mathbf{x}, \mathbf{v}) = \begin{bmatrix} \mathbf{x}^\top & \mathbf{v}^\top \end{bmatrix} \begin{bmatrix} \mathbf{TK} & \epsilon \mathbf{I} \\ \epsilon \mathbf{I} & \mathbf{T} \end{bmatrix} \begin{bmatrix} \mathbf{x} \\ \mathbf{v} \end{bmatrix} = \mathbf{x}^\top (\mathbf{TK}) \mathbf{x} + \mathbf{v}^\top \mathbf{T} \mathbf{v} + 2\epsilon \mathbf{x}^\top \mathbf{v} \qquad (122)$$

Here, $\mathbf{T}$ is any positive diagonal matrix such that $\mathbf{TB} + \mathbf{B}^\top \mathbf{T} \succ 0$ and $\mathbf{K}$ is also a positive diagonal matrix given by $\mathbf{D}\left(\sqrt{\mathbf{a}_s}/(\boldsymbol{\tau}_y \odot \boldsymbol{\tau}_a)\right)$. Now, for a valid Lyapunov function, we can take $\epsilon$ to be arbitrarily small, Eq. 51. Therefore, the *energy* minimized by the dynamical system is given by $V(\mathbf{x}, \mathbf{v}) = \mathbf{x}^\top (\mathbf{TK}) \mathbf{x} + \mathbf{v}^\top \mathbf{T} \mathbf{v}$. This is a high-dimensional version of the energy of a damped harmonic oscillator. For a single oscillator, we have $V(x, v) = t(kx^2 + v^2)$ which is proportional to the total energy (kinetic + potential) of the oscillator.

We now express this *energy* in terms of the variables relevant to ORGaNICs, i.e., we find $V(\mathbf{y}, \mathbf{a})$. First, we denote the Jacobian matrices in RHS of Eq. 119 & 121 by $\mathbf{A}$ & $\mathbf{B}$, respectively. We note the simple fact that $\mathbf{A}$ & $\mathbf{B}$ are related by a similarity transformation (a change of basis). This means that there exists an invertible matrix $\mathbf{U}$, such that $\mathbf{A} = \mathbf{U}^{-1} \mathbf{B} \mathbf{U}$ and the corresponding transform is given by $[\mathbf{x} \ \mathbf{v}]^\top = \mathbf{U}[\mathbf{y} - \mathbf{y}_s \ \mathbf{a} - \mathbf{a}_s]^\top$. Assuming that $\mathbf{U}$ is invertible, we can write this equation as $\mathbf{U}\mathbf{A} = \mathbf{B}\mathbf{U}$. To solve this, we consider a block matrix representation of $\mathbf{U}$ and find the following solution,

$$\mathbf{U} = \begin{bmatrix} \mathbf{D}\left(\frac{\sqrt{\mathbf{a}_s} \odot \boldsymbol{\tau}_y}{\mathbf{y}_s}\right) & \mathbf{0} \\ -\mathbf{D}\left(\frac{\mathbf{a}_s}{\mathbf{y}_s}\right) & -\frac{1}{2}\mathbf{I} \end{bmatrix} \qquad (123)$$

This change of basis gives us the transformation,

$$\begin{bmatrix} \mathbf{x} \\ \mathbf{v} \end{bmatrix} = \begin{bmatrix} \mathbf{D}\left(\frac{\sqrt{\mathbf{a}_s} \odot \boldsymbol{\tau}_y}{\mathbf{y}_s}\right) & \mathbf{0} \\ -\mathbf{D}\left(\frac{\mathbf{a}_s}{\mathbf{y}_s}\right) & -\frac{1}{2}\mathbf{I} \end{bmatrix} \begin{bmatrix} \mathbf{y} - \mathbf{y}_s \\ \mathbf{a} - \mathbf{a}_s \end{bmatrix} \qquad (124)$$

or,

$$\mathbf{x} = \frac{\sqrt{\mathbf{a}_s} \odot \boldsymbol{\tau}_y}{\mathbf{y}_s} \odot (\mathbf{y} - \mathbf{y}_s)$$

$$\mathbf{v} = -\frac{\mathbf{a}_s}{\mathbf{y}_s} \odot (\mathbf{y} - \mathbf{y}_s) - \frac{(\mathbf{a} - \mathbf{a}_s)}{2} \qquad (125)$$

Substituting these expressions into $V(\mathbf{x}, \mathbf{v}) = \mathbf{x}^\top (\mathbf{T}\mathbf{K})\,\mathbf{x} + \mathbf{v}^\top \mathbf{T}\mathbf{v}$, and assuming the diagonal entries of the matrix $\mathbf{T}$ to be $t_i$ and substituting the diagonal entries of $\mathbf{K}$, $k_i \to \sqrt{a_{si}}/(\tau_{y_i}\tau_{ai})$, we get the *energy* in terms of $\mathbf{y}$ and $\mathbf{a}$,

$$V(\mathbf{y}, \mathbf{a}) = \sum_{i=1}^{n} t_i \left[ \frac{\tau_{y_i}}{\tau_{ai}} \frac{a_{si}^{3/2}}{y_{si}^2} (y_i - y_{si})^2 + \frac{a_{si}}{y_{si}^2} \left( \sqrt{a_{si}}\,(y_i - y_{si}) + \frac{y_{si}}{2\sqrt{a_{si}}}(a_i - a_{si}) \right)^2 \right]$$

(126)

We notice that Taylor expanding the term $\sqrt{a_i}y_i$ about $\sqrt{a_{si}}y_{si}$ and ignoring the second order terms, we get,

$$\sqrt{a_i}y_i \approx \sqrt{a_{si}}y_{si} + \sqrt{a_{si}}\,(y_i - y_{si}) + \frac{y_{si}}{2\sqrt{a_{si}}}(a_i - a_{si})$$

(127)

Therefore the *energy* function, $V(\mathbf{y}, \mathbf{a})$, is given by,

$$V(\mathbf{y}, \mathbf{a}) = \sum_{i=1}^{n} t_i \frac{a_{si}}{y_{si}^2} \left[ \frac{\tau_{y_i}}{\tau_{ai}} \sqrt{a_{si}}\,(y_i - y_{si})^2 + (\sqrt{a_i}y_i - \sqrt{a_{si}}y_{si})^2 \right]$$

(128)

Notice that $\sqrt{a_{si}}y_{si} = b_i z_i$. This gives us the following expression for the *energy* function,

$$V(\mathbf{y}, \mathbf{a}) = \sum_{i=1}^{n} t_i \frac{a_{si}}{y_{si}^2} \left[ \frac{\tau_{y_i}}{\tau_{ai}} \sqrt{a_{si}}\,(y_i - y_{si})^2 + (\sqrt{a_i}y_i - b_i z_i)^2 \right]$$

(129)

Further, for an ORGaNICs model containing one $y$ and one $a$ neuron, after removing the proportionality constants, the *energy* function is given by,

$$V(y, a) = \frac{\tau_y}{\tau_a} \sqrt{a_s}\,(y - y_s)^2 + (\sqrt{a}y - bz)^2$$

(130)

After plugging in the steady-state values, we get,

$$V(y, a) = \frac{\tau_y}{\tau_a} \sqrt{b_0^2 \sigma^2 + w b^2 z^2} \left( y - \frac{bz}{\sqrt{b_0^2 \sigma^2 + w b^2 z^2}} \right)^2 + (\sqrt{a}y - bz)^2$$

(131)

For this system, it is easy to verify that this is a valid Lyapunov function and is minimized by the dynamics of the circuit. We now demonstrate that it has the properties of a Lyapunov function. First, $V(y_s, a_s) = 0$ and $V(y_s, a_s) > 0 \;\forall\; y \neq y_s \;\&\; a \neq a_s$, this can be easily seen from Eq. 130. Second, we need to show that, $\dot{V}(y, a) < 0 \;\forall\; y \neq y_s \;\&\; a \neq a_s$. Using Eq. 119 & 131 $\dot{V}(y, a)$, we can write the total time derivative of the *energy* to be,

$$\frac{dV(y, a)}{dt} = \frac{\partial V}{\partial y}\frac{dy}{dt} + \frac{\partial V}{\partial a}\frac{da}{dt}$$

$$= -\frac{(2ya_s - 3a_s y_s + ay_s)^2 \left( \sqrt{a_s}\tau_a - wy_s^2\tau_y + \tau_y \right)}{2a_s \tau_a \tau_y}$$

(132)

Since $a_s > 0$ and,

$$\left( \sqrt{a_s}\tau_a - wy_s^2\tau_y + \tau_y \right) = \left( \sqrt{a_s}\tau_a + \tau_y \left( \frac{b_0^2 \sigma^2}{b_0^2 \sigma^2 + w b^2 z^2} \right) \right) > 0,$$

(133)

for all the choices of parameters and $\forall\; y \neq y_s, a \neq a_s$, we have $\dot{V}(y, a) < 0$, therefore, it is a valid Lyapunov function and can be interpreted as the *energy* that decreases with time via the dynamics of ORGaNICs.

# I  Training details

The code (written in PyTorch [92]) to produce all the results can be found at https://github.com/martiniani-lab/dynamic-divisive-norm. For both the static input and the sequential input, we train ORGaNICs on the MNIST handwritten digit dataset [72], and to the best of our knowledge, it does not pose any privacy concern and has been used widely by the ML community freely. The simulations were performed on an HPC cluster. All of the models were trained on a single A100 (80GB) GPU. We use Adam optimizer [93] with default parameters for minimizing the loss function.

## I.1 Static MNIST input

We performed a random split of 57,000 training samples and 3,000 validation samples and picked the model with the largest validation accuracy for testing. To make a direct comparison to SSN [55], we use the same architecture structure as theirs. First, we train an autoencoder (Table 3) to reduce the dimensionality of MNIST images to 40 by using a mean-squared loss function. Then, we use this 40-dimensional vector as input to ORGaNICs and train it using the cross-entropy loss function. We additionally make the input gain $\mathbf{b}$ dependent on the input $\mathbf{x}$, $\mathbf{b} = f(\mathbf{W}_{bx}\mathbf{x})$, where $f$ is sigmoid. A layer of ORGaNICs is given by Eq. 15,

$$\boldsymbol{\tau}_y \odot \dot{\mathbf{y}} = -\mathbf{y} + f(\mathbf{W}_{bx}\mathbf{x}) \odot (\mathbf{W}_{zx}\mathbf{x}) + \left(\mathbf{1} - \sqrt{\lfloor \mathbf{a} \rfloor}\right) \odot (\mathbf{W}_r \mathbf{y})$$

$$\boldsymbol{\tau}_a \odot \dot{\mathbf{a}} = -\mathbf{a} + \mathbf{b}_0^2 \odot \boldsymbol{\sigma}^2 + \mathbf{W}\left(\mathbf{y}^2 \odot \lfloor \mathbf{a} \rfloor\right)$$

(134)

The "output" of a layer is the steady-state firing rate of the neuron with the positive receptive field, i.e., $\mathbf{y}_s^+ = \lfloor \mathbf{y}_s \rfloor^2$. We parameterize $\mathbf{W}_r$ to have a maximum singular value of 1 and instead of simulating the dynamical system to find the fixed point, we use the iterative Algorithm 1 with a maximum number of steps = 10. More details about the parameters are given in Table 4; kaiming uniform initialization is used from [94]. Additional hyperparameters are given in Table 7. We train ORGaNICs in a single-layer setting with the number of $\mathbf{y}$ neurons encoding the input, $N_1 = 50, 80$. We also train two-layer ORGaNICs (Table 5) with $N_1 = 120$ and $N_2 = 60$ neurons in each layer. The model is trained using backpropagation and takes approximately 10 min to fully train.

Table 3: Autoencoder architecture

| Layer | Shape | Nonlinearity |
|---|---|---|
| Input $\rightarrow$ encoder (layer-1) | $784 \times 360$ | ReLU |
| encoder (layer-1) $\rightarrow$ encoder (layer-2) | $360 \times 120$ | ReLU |
| encoder (layer-2) $\rightarrow$ embedding | $120 \times 40$ | sigmoid |
| embedding $\rightarrow$ decoder (layer-1) | $40 \times 120$ | ReLU |
| decoder (layer-1) $\rightarrow$ decoder (layer-2) | $120 \times 360$ | ReLU |
| decoder (layer-2) $\rightarrow$ output | $360 \times 784$ | sigmoid |

Table 4: ORGaNICs parametrization for static MNIST classification

| Parameter | Shape | Learned | Initialization |
|---|---|---|---|
| $\mathbf{W}_{zx}$ | $N \times M$ | yes | kaiming uniform |
| $\mathbf{W}_{bx}$ | $N \times M$ | yes | kaiming uniform |
| $\mathbf{W}_r$ | $N \times N$ | yes | identity |
| $\mathbf{W}$ | $N \times N$ | yes | ones |
| $\mathbf{b}_0$ | $N$ | yes | random normal |
| $\boldsymbol{\sigma}$ | $N$ | no | ones |

Table 5: ORGaNICs architecture for static MNIST classification

| Layer | Shape | Nonlinearity |
|---|---|---|
| Input $\rightarrow$ ORGaNICs (layer-1) | $40 \times N_1$ | None |
| ORGaNICs (layer-1) $\rightarrow$ ORGaNICs (layer-2) | $N_1 \times N_2$ | None |
| ORGaNICs (layer-2) $\rightarrow$ fully-connected | $N_2 \times 10$ | None |

## I.2 Permuted and Unpermuted sequential MNIST

We performed a random split of 57,000 training samples and 3,000 validation samples and picked the model with the largest validation accuracy for testing. The unpermuted sequential MNIST task is defined as follows: for a given $28 \times 28$ image, we flatten it to get a one-dimensional, 784 timestep-long input. Then these pixels are presented as an input ($\mathbf{x}_i$, one pixel at each time-step $i$) to the Euler discretized rectified ORGaNICs model (Eq. 87) with rectified input drive, given by the following

equations,

$$\mathbf{y}_{i+1} = \mathbf{y}_i + \frac{\Delta t}{\boldsymbol{\tau}_y} \odot \left(-\mathbf{y}_i + \mathbf{b}_i \odot \lfloor \mathbf{W}_{zx}\mathbf{x}_i \rfloor + \left(\mathbf{1} - \mathbf{a}_i^+\right) \odot \lfloor \mathbf{W}_r\mathbf{y}_i \rfloor\right)$$

$$\mathbf{a}_{i+1} = \mathbf{a}_i + \frac{\Delta t}{\boldsymbol{\tau}_a} \odot \left(-\mathbf{a}_i + \mathbf{b}_{0,i}^2 \odot \boldsymbol{\sigma}^2 + \mathbf{W}\left(\mathbf{y}_i^+ \odot \mathbf{a}_i^{+2}\right)\right)$$

$$\mathbf{b}_{i+1} = \mathbf{b}_i + \frac{\Delta t}{\boldsymbol{\tau}_b} \odot \left(-\mathbf{b}_i + f(\mathbf{W}_{bx}\mathbf{x}_i + \mathbf{W}_{by}\mathbf{y}_i + \mathbf{W}_{ba}\mathbf{a}_i)\right)$$

$$\mathbf{b}_{0,i+1} = \mathbf{b}_{0,i} + \frac{\Delta t}{\boldsymbol{\tau}_{b_0}} \odot \left(-\mathbf{b}_{0,i} + f(\mathbf{W}_{b_0 x}\mathbf{x}_i + \mathbf{W}_{b_0 y}\mathbf{y}_i + \mathbf{W}_{b_0 a}\mathbf{a}_i)\right)$$

(135)

When we are done presenting the pixels we use the last hidden state, i.e., $\mathbf{y}_{784}$, to make the predictions. To make this more challenging we also train ORGaNICs on permuted sMNIST where we first permute the pixels of all the images in some random order and the rest of the task is the same. Instead of parametrizing $\boldsymbol{\tau}$, we parametrize $\Delta t/\boldsymbol{\tau}_y = 0.05 * f(\mathbf{p}_y)$, $\Delta t/\boldsymbol{\tau}_a = 0.01 * f(\mathbf{p}_a)$ and $\Delta t/\boldsymbol{\tau}_b = 0.1 * f(\mathbf{p}_b)$ and $\Delta t/\boldsymbol{\tau}_{b_0} = 0.1 * f(\mathbf{p}_{b_0})$, so we can control the dimensionless relative time constants. In practice, we find it is better to make the $\mathbf{a}$ neurons sluggish compared to $\mathbf{y}$. This is based on the intuition given by the two-dimensional phase portrait for different relative time constants Fig. 3. All the parameters (including $\mathbf{W}_r$) are unconstrained for this task with initialization specified in Table 6. Since ORGaNICs are stable, we did not need to use gradient clipping for training, which is commonly used for LSTMs. Additionally, we train the model using a StepLR learning rate scheduler with parameters given in Table 7. The model is trained using backpropagation through time (BPTT) and takes approximately 30 hours to fully train.

Table 6: ORGaNICs parametrization for sequential MNIST classification

| Parameter | Shape | Learned | Initialization |
|---|---|---|---|
| $\mathbf{W}_{zx}$ | $N \times 1$ | yes | kaiming uniform |
| $\mathbf{W}_{bx}$ | $N \times 1$ | yes | kaiming uniform |
| $\mathbf{W}_{by}$ | $N \times N$ | yes | kaiming uniform |
| $\mathbf{W}_{ba}$ | $N \times N$ | yes | kaiming uniform |
| $\mathbf{W}_{b_0 x}$ | $N \times 1$ | yes | kaiming uniform |
| $\mathbf{W}_{b_0 y}$ | $N \times N$ | yes | kaiming uniform |
| $\mathbf{W}_{b_0 a}$ | $N \times N$ | yes | kaiming uniform |
| $\mathbf{W}_r$ | $N \times N$ | yes | identity |
| $\mathbf{W}$ | $N \times N$ | yes | ones |
| $\boldsymbol{\sigma}$ | $N$ | no | ones |

Table 7: Hyperparameters

| Hyperparameter | Static MNIST | Sequential MNIST |
|---|---|---|
| Batch size | 256 | 256 |
| Initial Learning rate | 0.001 | 0.01 |
| Weight decay | $10^{-5}$ | $10^{-5}$ |
| Step size (StepLR) | None | 30 epochs |
| Gamma (StepLR) | None | 0.8 |

## J  Supplementary figures

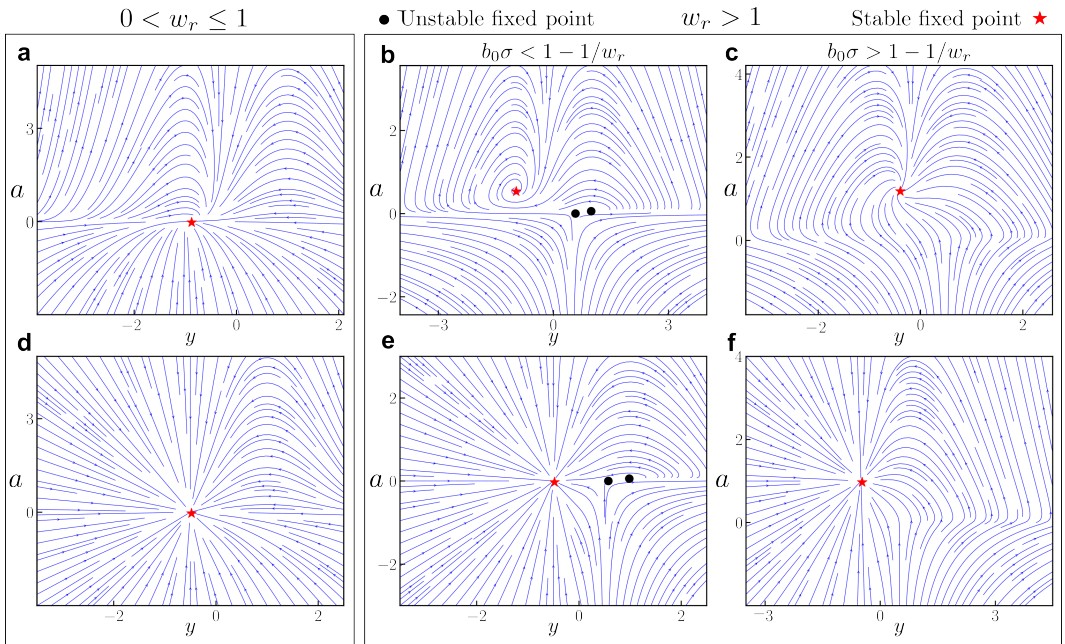

Figure 2: **Phase portraits for 2D ORGaNICs with negative input drive.** We plot the phase portraits of 2D ORGaNICs in the vicinity of the stable fixed point for contractive (**a, d**) and expansive (**b, c, e, f**) recurrence scalar $w_r$. A stable fixed point always exists, regardless of the parameter values. **(a-c)**, The main model (Eq. 16). **(d-f)**, The rectified model (Eq. 102). Red stars and black circles indicate stable and unstable fixed points, respectively. The parameters for all plots are: $b = 0.5$, $\tau_a = 2\,\text{ms}$, $\tau_y = 2\,\text{ms}$, $w = 1.0$, and $z = -1.0$. For **(a) & (d)**, the parameters are $w_r = 0.5$, $b_0 = 0.5$, $\sigma = 0.1$; for **(b) & (e)**, $w_r = 2.0$, $b_0 = 0.5$, $\sigma = 0.1$; and for **(c) & (f)**, $w_r = 2.0$, $b_0 = 1.0$, $\sigma = 1.0$.

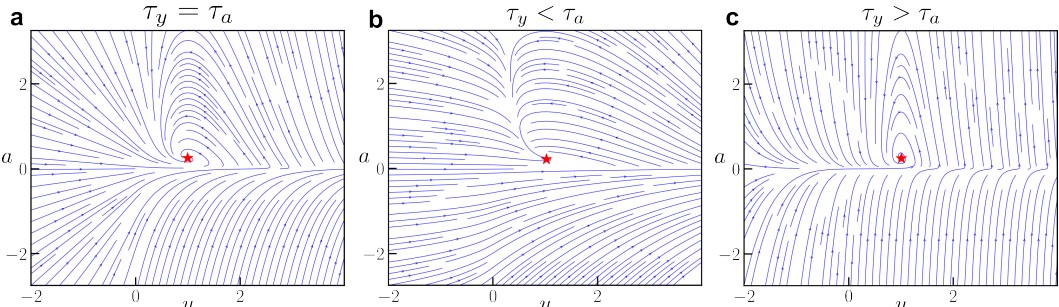

Figure 3: **Phase portraits for 2D rectified ORGaNICs for different time constants.** Red stars indicate stable fixed points. The parameters for all plots are: $w_r = 1.0$, $b_0 = 0.5$, $b = 0.5$, $\sigma = 0.1$, $w = 1.0$, and $z = 1.0$. For **(a)**, the time constants are $\tau_a = 2\,\text{ms}$, $\tau_y = 2\,\text{ms}$; for **(b)**, $\tau_a = 10\,\text{ms}$, $\tau_y = 2\,\text{ms}$; for **(c)**, $\tau_a = 2\,\text{ms}$, $\tau_y = 10\,\text{ms}$.

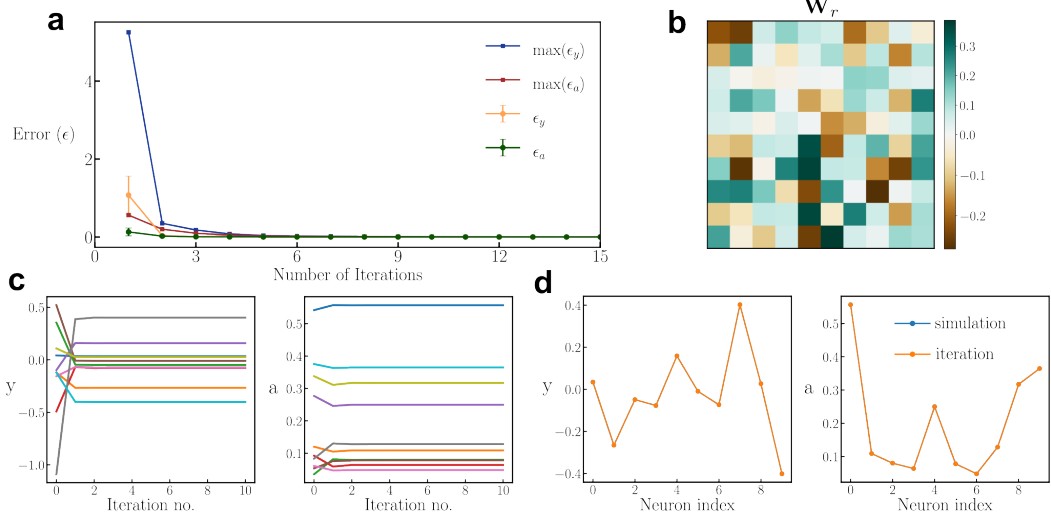

Figure 4: **Fast convergence of the iterative algorithm.** The results are for 20-dimensional ORGaN-ICs (10 **y** and 10 **a** neurons) with random parameters and inputs with the additional constraint of the maximum singular value of $\mathbf{W}_r$ equal to 1 and $||\mathbf{z}|| < 1$. **(a)**, Mean (with error bars representing 1-sigma S.D.) and maximum errors ($\epsilon$) as a function of number of iterations. $\epsilon$ is calculated as the norm of the difference between the true solution (found by simulation starting with random initialization) and the iteration solution. **(b)**, An example of a randomly sampled $\mathbf{W}_r$. **(c)**, Steady-state approximation as a function of iteration number. Different lines represent different neurons. **(d)**, Overlap between the iteration solution (after 15 iterations) and the true solution.

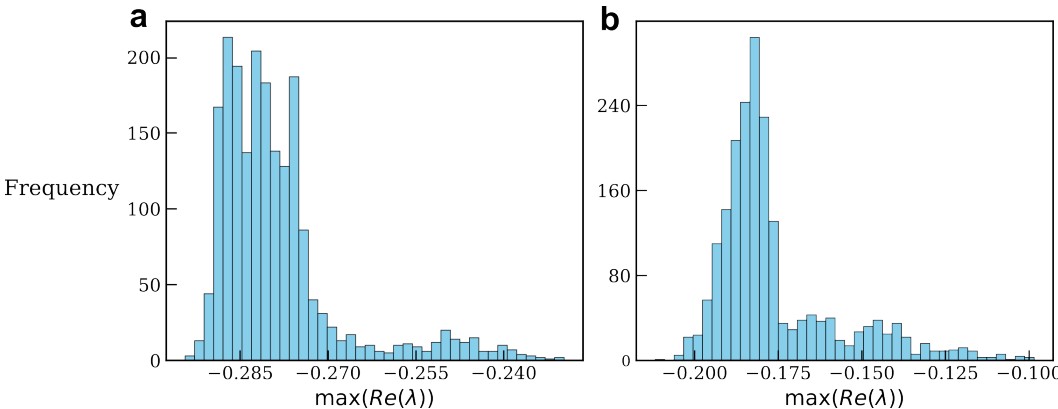

Figure 5: **Histogram for the eigenvalue with the largest real part**. We train two-layer ORGaNICs ($\tau_a = \tau_y = 2\,\text{ms}$) with a static MNIST input where $\mathbf{W}_r$ is constrained to have a maximum singular value of 1. We plot the histogram of eigenvalues of the Jacobian matrix with the largest real part, for inputs from the test set. We find that all the eigenvalues of the Jacobian have negative real parts, implying asymptotic stability. **(a)**, histogram for the first layer. **(b)**, histogram for the second layer. Note that since this is implemented in a feedforward manner, this is a cascading system with no feedback, hence we can perform the stability analysis of the two layers independently.

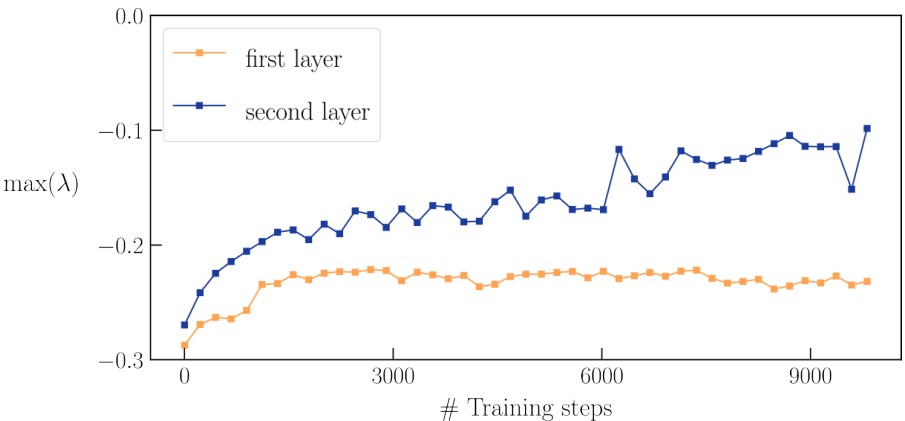

Figure 6: **Eigenvalue with the largest real part while training on static input (MNIST) classification task.** This plot shows the largest real part of eigenvalues across all test samples as training progresses. The fact that the largest real part consistently remains below zero indicates that the system maintains stability throughout training.

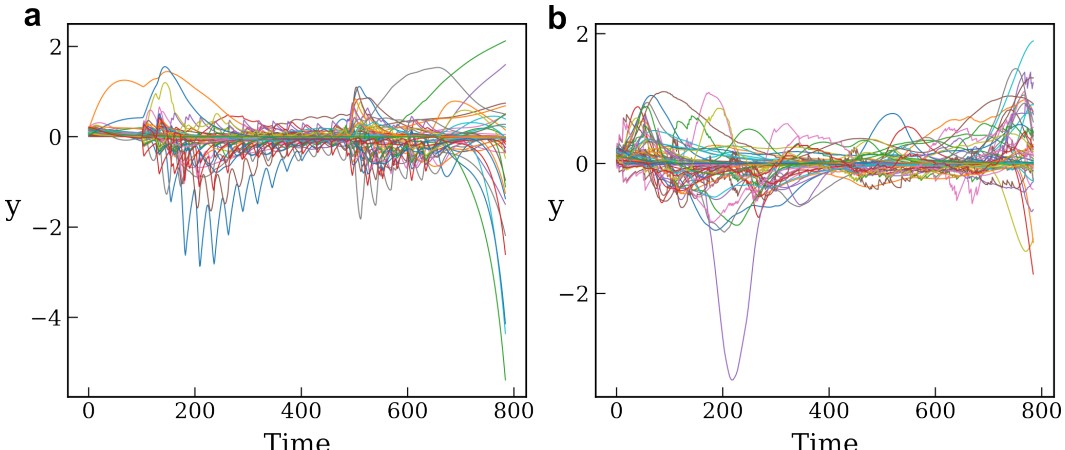

Figure 7: **Trajectories of the hidden states (y).** This plot shows the dynamics of the hidden state as the input is being presented sequentially. We train ORGaNICs (128 units) as an RNN on **(a)**, unpermuted sequential MNIST and **(b)**, permuted sequential MNIST. The inputs are picked randomly from the test set. The hidden state trajectory remains bounded, indicating stability.

