# OpenReview forum: "Unconditional stability of a recurrent neural circuit implementing divisive normalization"
_NeurIPS.cc/2024/Conference — NeurIPS 2024 poster_

### Official Review · Reviewer_ma5B · 2024-07-09

**Soundness:** 4
**Presentation:** 3
**Contribution:** 3
**Rating:** 7
**Confidence:** 3

**Summary:**

Stability is a critical notion in the understanding of dynamical systems as well as for learning them. This paper studies the stability of a biologically plausible recurrent cortical model, ORGaNICs, that implements divisive normalization. More precisely:
- it demonstrates the local stability of two specific subclasses of the model.
- it shows that the model performs competitively on standard machine learning tasks.

**Strengths:**

Understanding how properties of biological neural networks lead to better learning properties is an important area of research and this paper nicely contributes to it by highlighting the potential importance of divisive normalization in stabilizing neural dynamics and facilitating training. The paper is overall well-written and nicely connected to both theoretical and empirical neuroscience results. The theoretical stability analysis is non-trivial and provides important insights into the model. The empirical results are solid.

**Weaknesses:**

The main weakness of the paper is that it might not be fully suited to the NeurIPS community in its current form:
- the introduction of the dynamics lacks some intuition on what the dynamics achieves. Equation 2 partially does so, but is likely not sufficient to fully convey the intuition. If possible, connecting to existing RNN architectures would help. Introducing the Lypanuov energy earlier could also help.
- many notions used in Section 4 might deserve a better introduction as NeurIPS is not a dynamical systems conference. This holds for "Lyapunov diagonally stable", "Z-matrix", "M-matrix" and, to some extent, to "stability" and "indirect method of Lyapunov". To make their results more accessible, the authors may want to provide intuitive definitions of these terms and why they are important.

Additionally, the connections made to the machine learning literature are sometimes imprecise:
- in L43, it is written that stability helps mitigate the vanishing gradient problem, which is inaccurate (e.g. a system with very fast time constants is stable but suffers from vanishing gradients). On top of that, stable recurrent networks having slow time constants can still suffer from some form of exploding gradients [Zucchet and Orvieto, 2024](https://arxiv.org/abs/2405.21064).
- in L61, the authors write that divisive normalization generalizes batch and layer normalization. While there are definitely some links, this is not true: divisive normalization affects the recurrent dynamics whereas both batch and layer normalizations are usually applied after the recurrent layer. In particular, those normalization schemes do not affect the stability of the system.
- in L349, divisive normalization is directly compared to the form of attention used in Transformers. Transformers normalize over the time axis, which is highly implausible and different from the type of attention mentioned here. This point deserves to be made more precise.

Finally, there are some additional links to existing ML literature that would nicely fit within the paper:
- the kind of neural network studied in Section 6.1 is known as [Deep equilibrium models, DEQ](https://arxiv.org/abs/1909.01377). One of the main difficulties that come with training such models is to [keep the dynamics stable](https://arxiv.org/abs/2106.14342) throughout learning. The studied architecture may be less prone to these behaviors: adding the DEQ baseline to the experiments and monitoring the stability of the dynamics during training would be interesting additions to Section 6.1.
- in Section 6.2, the baseline architectures are rather "old". The state-of-the-art networks on tasks requiring modeling long-range dependencies are state-space models (e.g. [S4](https://arxiv.org/abs/2111.00396) or [LRU](https://arxiv.org/abs/2303.06349)). Additionally, sequential models are known to be extremely sensitive to initializations on tasks such as sMNIST, see [Amos et al. 2023](https://arxiv.org/abs/2310.02980). It is therefore important to describe the initialization scheme of the recurrence in the main text, as it is likely a main driver of the performance.

I hope these remarks can help the authors to improve their paper.

**Questions:**

- The model considered has the same number of E and I neurons, which is, to the best of my knowledge, far from being the case in the brain. Is the system still stable when the number of I neurons is much smaller than the number of E neurons?

**Limitations:**

Limitations are properly addressed.

---

> ### Author Rebuttal · Authors · 2024-08-07
>
> **Weaknesses**:
>
>
> - “the introduction of the dynamics … why they are important”. **Answer:** We thank the reviewer for the insightful feedback. We will add more intuition about the dynamics in the model description and compare ORGaNICs to LSTM, an RNN architecture similar to ORGaNICs. We will also provide intuitive definitions of the terms used in the stability proof.
>
>
> - “in L43, it is written that stability … Zucchet and Orvieto, 2024”. **Answer:** We appreciate the referee's observation. While stability indeed addresses the issue of exploding gradients, excessive stability—where the real parts of the eigenvalues of the Jacobians are significantly smaller than 0—can lead to a lossy system prone to vanishing gradients. ORGaNICs effectively mitigates both issues: Exploding gradients: through its inherent stability. Vanishing gradients: by processing information across various timescales (and via the long effective time constant when the membrane time constants are fixed) while maintaining stability, resulting in a blend of lossy and non-lossy neurons. The efficacy of ORGaNICs in mitigating vanishing gradients is demonstrated by its competitive performance against architectures specifically designed to address this issue, such as LSTMs. We will provide a more detailed explanation of this in our revised manuscript. Moreover, we posit that the built-in normalization feature of ORGaNICs may alleviate the "curse of memory" described by Zucchet and Orvieto (2024), as normalization is proposed as a potential solution to this problem.
>
>
> - “in L61, … stability of the system”. **Answer:** Divisive normalization (DN) was introduced as a model of steady-state response of neurons, functioning as a static nonlinearity similar to batch and layer normalization. However, in the brain, there are no static nonlinearities; it is proposed that DN is achieved via a recurrent circuit. ORGaNICs is such a recurrent circuit designed so that the responses of neurons at steady state follow the DN equation, with stability being an emergent property of the circuit. The DN equation indeed generalizes batch and layer normalization, as shown in the work by Ren, M., Liao, R., Urtasun, R., Sinz, F. H., & Zemel, R. S. (2016).
>
>
> - “in L349, … made more precise”. **Answer:** The key similarity is that both operate via changing the input gain, which also models a wide range of results from neurophysiological and psychophysical experiments. In particular, it has been shown that normalization explains experimental measurements of temporal attention (Denison, Carrasco, Heeger, Nature Human Behavior, 2021), i.e., over the time axis, analogous to transformers.
>
> - “the kind of neural network … additions to Section 6.1”. **Answer:** We thank the reviewer for pointing this out. We checked if the models dynamics are stable throughout training and found that this is indeed true. Please see Fig.2 of the attached pdf.
>
> - “in Section 6.2, … of the performance”. **Answer:** We are aware that state-space models perform better at these tasks and we will mention this (along with citations) in the paper, but our goal is to compare ORGaNICs with other RNNs designed with the property of less-dissipation and stability to mitigate the problem of vanishing and exploding gradients. We thank the reviewer for pointing out the sensitivity of initialization on sequential modeling tasks. We have added that detail (uniform random initialization) in the revised manuscript.
>
> **Questions:**
>
>
> - “The model … E neurons?” **Answer:** Experimental data suggest that the ratio of E and I neurons is 4:1. Our results are also applicable for a different E/I ratio than 1.

---

> > ### Comment · Reviewer_ma5B · 2024-08-08
> >
> > I acknowledge the rebuttal and keep my score as it is.

---

### Official Review · Reviewer_8Tz2 · 2024-07-12

**Soundness:** 3
**Presentation:** 4
**Contribution:** 1
**Rating:** 5
**Confidence:** 4

**Summary:**

This work is based on ORGaNICs, a particular type of RNN architecture that implements divisive normalization (from neuroscience). The authors explore whether ORGaNICs are stable enough to be meaningfully (and stably) trained by gradient descent. Due to their stability, which the authors have proven theoretically for specific cases and empirically in general, the authors claim that ORGaNICs can learn long-term dependencies and solve vanishing and exploding gradients found in other neurodynamical models.

**Strengths:**

Trained RNNs in neuroscience have always had a standard vanilla architecture or have minimal modifications. This work introduces the possibility of training a more complex model motivated by biology while having no stability issues.

**Weaknesses:**

Subjectively, I believe that the scope of this work is too limited for the audience at NeurIPS, but I will not consider this in my final score.

The general theme of my objective issues with this work is the slight overselling of this work:

1. Fundamentally, ORGaNIC is an architecture. Song et al. [2016] implemented Dale's law by manipulating the recurrent weight matrix, which is a **method** applicable to any architecture. Soo et al. [2024] trained vanilla RNNs on long-term dependencies using skip connections through time, which once again is a **method** applicable to any architecture. So the claim that these "**models** do not accurately reflect cortical dynamics" in lines 105-106 do not make sense to me. To illustrate my point, can the authors comment on whether there are any issues if Soo et al. [2024] is applied to ORGaNICs (beyond the fact that it is not needed)?

2. This work seems to be strongly driving the point that divisive normalization is a key criteria for biological plausibility. There are a variety of biological properties that the brain exhibits. For example, there are 22 different properties of V1 identified in Marques et al. [2020], of which specific versions of DN make up some of them. It is not convincing to select a model based on one particular effect found in V1.

Marques et al. [2020]: https://www.biorxiv.org/content/10.1101/2021.03.01.433495v2

3. The authors claim that ORGaNICs is stable and therefore solves exploding and vanishing gradients, which does not make sense. The way the authors used the word "stable" here is in the language of dynamical systems. "Stability" in the context of exploding and vanishing gradients has a different meaning. The fact that the model can learn sequential MNIST (gradient stability) does not mean that it is due to its (dynamical) stability. A completely (dynamically) stable architecture can still have vanishing gradients. Lines 115-116 suggest to me that the authors are somewhat aware of this but not completely.

**Questions:**

ORGaNIC begins with the word "oscillatory". Can the authors comment on whether the network must always be oscillatory? I see that the analysis looks into stable and unstable fixed points, which means that the answer is no, which makes the name confusing to me.

Likewise, can the authors elaborate on the "gate" in ORGaNIC and how it is the "biophysically plausible extension" (line 86) of gating in machine learning? Specifically, why it is biophysically plausible, and why it is an extension.

**Limitations:**

In line with my issue of overselling this work, there really is not any honest discussion of genuine limitations in the discussion section, which the authors claim they have done so in the checklist. Discretization problems (lines 350-354) and using single-layers (line 355) are not specific to this work at all. I would encourage the authors to talk about genuine limitations. For example, I suspect but cannot prove, that there would be some increased computational complexity compared to vanilla RNNs, since having multiple differential equation for a single model means that more things need to be done in a single time step.

This work, focusing on a biologically plausible model, shows the model performing machine learning experiments (in contrast, in Heeger and Mackey [2019] the model is performing neuroscience experiments).

Also, while this model implements divisive normalization, as I mentioned above there are many biological phenomena that the brain exhibits. It would be unreasonable to ask the authors to state what ORGaNICs cannot do, but at least I would like to ask the authors if ORGaNICs can do everything that SSNs (which they cited) can do?

These are the limitations that I believe readers would like to see.

---

> ### Author Rebuttal · Authors · 2024-08-07
>
> **Weaknesses:**
>
>
> - “Fundamentally, ORGaNIC … one particular effect found in V1”.  **Answer:** DN is observed in numerous cortical areas beyond V1 and across different species (Carandini & Heeger, 2012). It explains a wide range of experimental phenomena in various neural systems and cognitive processes (citations in lines 49-53). DN can also be shown to be the appropriate nonlinearity for removing dependencies in sensory inputs, resulting in a set of neural responses that are statistically independent (Simoncelli and Olshausen, 2001). Consequently, DN is not just an effect observed in V1, but a computational principle (Carandini & Heeger, 2012) that can be derived from a statistical optimization criterion. It is thus unsurprising that DN has been shown to dramatically improve the performance of ML models in several applications (e.g., Balleé, Laparra, Simoncelli, 2015; Balleé, Laparra, Simoncelli, 2017). Moreover, batch and layer normalization (of which DN is a generalization, see Ren, M., Liao, R., Urtasun, R., Sinz, F. H., & Zemel, R. S. (2016)) are critically important architectural motifs in modern deep learning architectures. There is thus overwhelming evidence that DN is an essential feature of both natural and artificial neural systems. In this paper we demonstrate that DN can be integrated into the dynamics of an RNN, leading to a provably stable circuit for which an interpretable normative principle (Lyapunov function) can be derived. This work does not just propose an architecture, but provides 1) unprecedented insight into the impact of normalization on the dynamics of RNNs, through an interpretable normative principle; 2) establishes an important precedent for how the incorporation of neurobiological principles can drive advances in ML.
>
>
> - “The authors claim … aware of this but not completely”. **Answer:** In RNN architectures derived from discretized ODEs, such as ORGaNICs, the stability of the dynamical system is intrinsically linked to the vanishing and exploding gradients (VG and EG) problem. Mathematical analysis and details can be found in, Haber and Ruthotto (2017); Chang, Chen, Haber, and Chi (2019); and Erichson et al. (2020). ORGaNICs effectively addresses both problems: EG: Mitigated through the architecture's inherent stability. VG: Addressed by processing information across various timescales (and via the long effective time constant when the membrane time constants are fixed), resulting in a blend of lossy and non-lossy neurons. The effectiveness of ORGaNICs in tackling VG is evidenced by its competitive performance against architectures specifically designed to address this issue, such as LSTMs.
>
>
> **Questions:**
>
>
> - “ORGaNIC begins … name confusing to me”. **Answer:** ORGaNICs is oscillatory in the sense that it can be mapped to a damped harmonic “oscillator” as we show in the paper. This means that for the right parameters and inputs, stochastically driven ORGaNICs exhibit a peak in the power spectrum, which is consistent with the  LFP (Local Field Potential) oscillatory activity observed in neural recordings, even though there are no sustained oscillations.
>
> - “Likewise, can the authors elaborate … it is an extension”. **Answer:** Gating in ORgaNICs is performed by modulating the input gain ($\mathbf{b}$) and the recurrent gain ($\mathbf{1} - \mathbf{a}^+$). Possible mechanisms of how the responses of these neurons may be computed by loops through higher visual cortical areas and/or thalamocortical loops are discussed in the Discussion/Mechanisms of Heeger, D. J., & Zemlianova, K. O. (2020) and its SI Appendix. The gate in ORGaNICs is an extension of the gates in LSTMs/GRUs because the recurrent gain/gate in ORGaNICs (unlike LSTMs/GRUs) is a particular nonlinear function of the output responses/activation, designed to achieve normalization.
>
>
>
> **Limitations:**
>
>
> - “In line with my issue … in a single time step”. **Answer:** The computational cost of ORGaNICs is similar to LSTMs since like LSTMs we have three extra sets of variables ($\mathbf{a}, \mathbf{b}, \mathbf{b}_0$) which are not directly used for prediction. Therefore, ORGaNICs is more computationally expensive than vanilla RNNs. We thank the reviewer for pointing this fact out. We have included this fact in the discussion. However, we do not consider this a limitation, as the increased computational cost results in improved stability, trainability, and interpretability.
>
> - “This work, … neuroscience experiments)”. **Answer:** Since training biologically plausible neurodynamical models using backpropagation is a challenging task (Soo, W., & Lengyel, M. (2022)), we demonstrate the trainability of ORGaNICs by naive backpropagation and backpropagation through time, without gradient clipping/scaling, on well studied ML tasks. Training ORGaNICs on neuroscience experiments will be done in future work.
>
>
> - “Also, while this model implements … SSNs (which they cited) can do?” **Answer:** ORGaNICs has been shown to model a wide range of neural phenomena, including sustained activity, sequential activity, motor preparation, and motor control (Heeger, D. J., & Mackey, W. E. (2019)), as well as simulate the dynamics of V1 activity (Heeger, D. J., & Zemlianova, K. O. (2020); S. Rawat, D.J. Heeger, and S. Martiniani. Cosyne Abstracts 2024) and the emergence of communication subspaces (S. Rawat, D.J. Heeger, and S. Martiniani. Cosyne Abstracts 2023). To address the reviewer’s question, ORGaNICs can do everything that an LSTM can do (while being stable, easily trainable, and interpretable). While a direct comparison of ORGaNICs and SSN across many tasks has not been done, on the one task we consider (static MNIST, following Soo, W., & Lengyel, M. (2022)) ORGaNICs outperforms SSN on the first try (i.e., without hyperparameter optimization). Consistent with our claims, ORGaNICs also performs comparably to LSTMs on sequential tasks such as sequential  (and permuted) MNIST.

---

> > ### Comment · Reviewer_8Tz2 · 2024-08-08
> > **Early response**
> >
> > I thank the authors for the thorough response. I will reply again in a few days after carefully reading them. But I need to post this early response first to give the authors a chance to reply.
> >
> > In my original review, I mentioned the point about the lack of any honest discussion of limitations, and then suggested three possible points for the authors to discuss as a limitation. Perhaps the authors felt the need to defend them as if they were criticisms, which is why I am writing this post to clarify: those points are not criticisms and I am truly encouraging them to give limitations of their work just like in any complete piece of research.
> >
> > Basically I am saying that in the "limitations" part of the response, I still do not see any limitations.

---

> > > ### Author Response · Authors · 2024-08-11
> > > **Discussion of limitations**
> > >
> > > In addition to the general limitations (discretization problem and using single-layers), we also discussed other limitations throughout the text. For clarity, we will summarize them again in the Discussion in a new version of the manuscript. These are,
> > >
> > > - **Limitation #1:** We pointed out very clearly that we cannot rigorously prove stability for the case of a general recurrent weight matrix. So our most general results rely on empirical evidence obtained through extensive numerical tests presented in the original manuscript, as well as additional results presented in response to the reviewers (see Fig.1 of the attached pdf for a demonstration of stability from random initial conditions).
> > >
> > > - **Limitation #2:** We are not yet taking advantage of the modulators to control the effective time constant and instead are learning the intrinsic time constants. This potential limitation was pointed out by reviewer NGE1 who asked for further validation of the model with fixed time constants. We welcomed the suggestion and performed further numerical tests in which we found that ORGaNICs achieves good performance even when we fix the intrinsic time constants (as shown in Table 1 of the attached pdf).
> > >
> > > - **Limitation #3:** In the current work, the weight matrices $ \mathbf{W_{by}}$, $ \mathbf{W_{ba}}$, $ \mathbf{W_{b_0 y}}$ and $\mathbf{W_{b_0 a}}$ are $n \times n$. This leads to the number of parameters increasing faster than other RNNs with hidden state size, as is evident from Table 2. This was pointed out by reviewer 2CUH. We will stress this limitation further in the updated version of the work. In future work, we plan to test the performance with compact and/or convolutional weights so that the # of parameters does not increase markedly with the hidden state size.
> > >
> > > To the extent that we tested the model for the purpose of demonstrating stability, and trainability by SGD without gradient clipping/scaling in a standard ML task, these are the only limitations that we encountered. We will add a section to the Discussion to summarize them clearly.
> > >
> > > Based on our current understanding and results, we have no evidence (theoretical or experimental) of additional limitations. In future work, we will address the challenges noted above, and benchmark the model extensively on additional ML tasks. If the reviewer has specific questions that we have not thought of, we would be happy to investigate them and perform additional numerical tests as we did in response to other reviewers.

---

> ### Comment · Reviewer_8Tz2 · 2024-08-12
> **Good response**
>
> I thank the authors for the follow up response. I am glad that authors finally understand the position I am coming from, so I will continue with additional comments.
>
> My main point, since the very first review I posted here, is that the tone of the work is a little overblown, citing the following examples:
> 1. criticizing method papers that are not specific to any architecture to promote your own model (point 1 of my weaknesses that remains unaddressed)
> 2. absolutely judging every model based on whether it can or cannot do DN (point 2 of my weaknesses that the authors doubled down on in the reply, highlighting the importance of DN)
>
> which leads me to feel like the paper is not coming from a genuine standpoint of trying to contribute to academia, but rather trying to sell itself. Also, it is appalling that the authors can claim "we have no evidence (theoretical or experimental) of additional limitation" when V1 has so many interesting phenomena (see bolded below of biological traits that ORGaNIC does not have). Again, from my purely sincere point of view for this work to read like a good paper, might I suggest a more contributing narrative that fosters collaboration and improvement as a field instead of relying on criticism:
>
> - the vanilla RNN architecture was adapted to be biologically realistic by [Song et al. 2016] by incoporating Dale's Law. [Soo et al. 2024] developed a method for such RNNs to learn long-term dependencies. ORGaNIC is a model that is already built on biological principles, and can learn long-term dependencies intrinsically, therefore not needing any of those results.
> (those papers provided methods, not models to be compared with ORGaNIC)
> - DN is a phenomenon found in the brain, and ORGaNIC is able to express this effect. The biological visual system also expresses other phenomena, such as **surround suppression, adaptation, attention-based mechanisms, foveal and peripheral vision, retinotopic mapping, columnar structure, binocular and monocular effects, unique processing of color and plasticity**. Other models have been built to study those traits, but here we focus on DN as it is one of the more interesting effects that is believed to stem purely from dynamical effects.
>
> (Edit: To be clear, these are examples of how to write a collaborative narrative. I am in no way forcing the authors to include anything in the paper.)
>
> Once again, I am glad that the authors are now giving genuine limitations, and I hope that they can continue to accept my constructive criticism on the points above. I will reply again right before the deadline to make my final decision.

---

> > ### Comment · Reviewer_8Tz2 · 2024-08-13
> > **Thank you**
> >
> > As the discussion period is coming to an end, I realize that this back and forth might have required too much time and effort from the authors, so I will post my concluding remarks now, giving the authors full benefit of the doubt in outstanding issues.
> >
> > Overall, the main rebuttal has addressed many doubts that I have, and I will raise my score by 1. This work represents progress for previously handcrafted models in neuroscience to be trained and actually perform tasks, which is a meaningful contribution to neuroscience.

---

> > > ### Author Response · Authors · 2024-08-13
> > > **Updating statements and summary of key results**
> > >
> > > As we noted in our original response, the model’s limitations had already been stated in the original manuscript. We will summarize them in the Discussion.
> > >
> > > The paper investigates a neurobiologically plausible RNN model that achieves divisive normalization (DN) through recurrent excitation. This model has been shown to recapitulate a broad range of neurophysiological observations, but the application of this theory to neuroscience is not the focus of the current paper. The core contributions can be summarized at a lay level as follows:
> > > - **Unconditional Stability:** ORGaNICs is shown to be unconditionally stable under mild constraints (this is highly nontrivial). We develop mathematical machinery to prove this in a couple of limiting cases, by expressing ORGaNICs as a mechanical (specifically, gyroscopically stabilized) system – to do this we even need to prove new results on mechanical systems.
> > > - **Implications for ML:** This stability allows ORGaNICs to be trained using naïve BPTT without gradient clipping/scaling, performing comparably to LSTMs. The point is not simply that ORGaNICs is at least as good as LSTMs, but that imposing DN dynamically makes training more efficient and robust.  Moreover, ORGaNICs trained by naïve BPTT fares well when compared to SSN trained by a DNG (the comparison is to SSN, not to DNG).
> > > - **Dynamic Normalization:** Unlike standard RNNs, where normalization is imposed a-posteriori to the output layer, ORGaNICs implements normalization dynamically: this is the first time this idea is tested in ML and analyzed theoretically. We believe that this is an important contribution. In contrast layer/batch normalization does not affect the stability of the system.
> > > - **Theoretical Insights:** The connection to mechanical systems enables deriving a Lyapunov function, offering a normative principle for understanding the model's dynamics.
> > >
> > > Much of the reviewer’s concerns are focused on the language used in the Related Work section which we will rephrase as follows (adapting the reviewer’s first suggestion):
> > >
> > > (Song et al., 2016) incorporated Dale’s law into the vanilla RNN architecture, which was successfully trained across a variety of cognitive tasks. Building on this, (Soo et al., 2024) developed a technique for such RNNs to learn long-term dependencies by using skip connections through time. ORGaNICs is a model that is already built on biological principles, and can learn long-term dependencies intrinsically, therefore it does not require the method used by (Soo et al., 2024).
> > >
> > > Regarding the remaining concerns. The reviewer raises concerns about modeling the full range of V1 phenomena, but V1 is not the focus of this paper. There is no mention of V1 or the visual system anywhere (except when discussing the history of DN). Many of the phenomena mentioned by the reviewer have been or can be incorporated into ORGaNICs (see paragraph below). Application of the ORGaNICs theory to model V1 and several other neural systems will be done in future work. Regardless, we welcome the reviewer’s suggestion and we will add the following text:
> > >
> > > DN has been proposed as a canonical neural computation (Carandini & Heeger, 2012) and is linked to many well-documented physiological (Brouwer & Heeger, 2011; Cavanaughet et al., 2002) and psychophysical (Xing & Heeger, 2000; Petrov et al., 2005) phenomena. DN models diverse neural processes: adaptation (Wainwright et al., 2002; Westrick et al., 2016), attention (Reynolds & Heeger, 2009)), automatic gain control (Heeger, Simoncelli & Movshon, 1996), decorrelation and statistical whitening (Lyu, & Simoncelli, 2009). Since ORGaNICs’ response follows the DN equation at steady-state, it already incorporates this wide variety of neural phenomena. ORGaNICs have been shown to capture some of the dynamics of neural activity (Heeger & Zemlianova 2020; Rawat et. al., Cosyne 2024). Additional phenomena not explained by DN (Duong et. al., NeurIPS 2024) can in principle be integrated into the model. Regardless, more work needs to be done, of course, to explain the full range of neurophysiological phenomena. In this paper, however, we focus on the effects of DN on the dynamical stability of ORGaNICs.
> > >
> > > When we state that “we have no evidence (theoretical or experimental) of additional limitations” we are referring specifically to ORGaNICs as an instance of RNNs for ML (because that is the focus of this paper), not to limitations of the theory as a neurobiological circuit model (which we intend to publish separately). We have no evidence that there is something that ORGaNICs couldn’t do when compared to an LSTM.
> > >
> > > Finally, we assure that the paper has been written with the most genuine intentions, purely driven by curiosity. In fact, the core contributions are conceptual and technical (in the form of theorems), rather than ad-hoc benchmarks. We are grateful for the reviewer’s constructive comments which we believe will help clarify the key contributions of the paper.

---

> > > > ### Comment · Reviewer_8Tz2 · 2024-08-14
> > > > **Thank you again**
> > > >
> > > > I thank the authors for all the time and effort put into addressing my remarks, and I appreciate them for their diligence.

---

### Official Review · Reviewer_FwjG · 2024-07-16

**Soundness:** 3
**Presentation:** 2
**Contribution:** 2
**Rating:** 5
**Confidence:** 2

**Summary:**

This paper studies the stability properties of a model of cortical circuits which was introduced in 2019 (and I wasn't yet aware of): the ORGaNIC model by Heeger & Mackey. This LSTM-like model uses a simple set of differential equations that unify several phenomena observed in cortex, including normalization (Carandini & Heeger, 2012), and in some sense map LSTMs onto plausible cortical circuitry. This paper presents two main theoretical results: (i) that the model is locally stable around its unique fixed point when the recurrent matrix is the identity, and that (ii) existence and uniqueness of a stable fixed point can be shown for a general 2D model (really just one "principal neuron", and a gate variable) under certain conditions on the parameters. The paper wraps up with a few training experiments on (regular/sequential/permuted) MNIST classification, showing that ORGaNICs perform well.

**Strengths:**

This paper is very strong on a technical level. The maths are sound as far as I could tell from going through the paper carefully (mostly main text; took only a cursory look at the appendices) but not re-deriving the equations myself. The proof of local stability in ORGaNICs when $W_r = I$ is elegant and actually rather sophisticated; perhaps some of the most useful collaterals of this proof are (i) the explicit derivation of the eigenvalues of the Jacobian when additionally all normalization weights are equal to the same (positive) value, and (ii) the energy function of Eq 13 that provides further insight into the dynamics of ORGaNICs. Both of these add to the interpretability of this model class. The iterative fixed-point-finding algorithm (Algorithm 1) seems novel and useful, too.

**Weaknesses:**

Overall this is a highly esoteric paper for which I have difficulty assessing potential impact (hence my high uncertainty rating). The theoretical results are impressively detailed but seem fairly limited in scope (how are the W_r = I and 2D cases relevant for either ML or neuroscience?). I am also not super convinced by the utility of the broader conjecture on stability at the end of section 5. The experiments are fairly limited in scope and breadth, too (e.g. no mention of hyperparameter tuning for the SSN comparison).

The abstract claims that it is “thanks to its intrinsic stability property” that ORGaNICs perform comparably to LSTMs; while this seems like a sensible hypothesis I don't think the paper really shows that, yet I believe the paper would be much stronger if that was shown to be true. In this respect, I wonder in what way the comparison to SSNs (e.g. Soo & Lengyel) helps make this case; is it because SSNs also perform a form of normalization at steady-state, much like ORGaNICs, but have otherwise no stability guarantees? Is it even true that SSN training is brittle / often fails because the network loses stability, thus giving rise to a stronger-than-usual tradeoff between training stability and learning rate magnitude? How could the author rule out that there might be another fundamental difference between these 2 models that has little to do with stability and yet underlies the performance difference? After all, no formal stability guarantees exist for standard LSTMs as far as I am aware, and that doesn't prevent them from training very well on most tasks (was gradient clipping even necessary for the LSTMs in those particular experiments?). In summary, I suppose the paper lacks a couple of convincing ablations to make the point that stability guarantees increase training performance.

On clarity of exposition: this is a pretty hard paper to follow, it's very dense and doesn't really offer the hierarchy of exposition that a reader would want to see -- the theoretical results are presented in a very flat way, with a permanent back and forth between details and bigger picture that I found hard to navigate.

**Questions:**

- The abstract (and main text) says you trained your models using "simple backpropagation through time" but appendix I says you trained all models using Adam -- in principle there is no contradiction between these two claims, as backprop is also involved in Adam, but I wonder why you wrote "simple backpropagation" (which could easily be misunderstood as "simple gradient descent", i.e. not Adam). What's non-simple backpropagation?

- The formatting of references is annoying :D please use parentheses to clearly separate references from the main text.

- typo on l104: "dynamics-neural growth" → "dynamics-neutral growth"

**Limitations:**

Some technical limitations are stated in the Discussion section.

*EDIT*: following the rebuttal, I am raising my score to a 5.

---

> ### Author Rebuttal · Authors · 2024-08-07
>
> **Weaknesses:**
>
> - “Overall this is a … uncertainty rating)”. **Answer:**
> We argue that the potential impact is high. LSTMs and GRUs have had a huge impact on ML/AI applications even though they are not always stable and hence require ad hoc techniques for training (e.g., gradient clipping/scaling). Furthermore, although normalization has been shown to improve the training and performance of CNNs, there has been no principled way of adding normalization to RNNs (putting a normalization layer between RNN layers is not the same as normalizing the activations within each RNN layer). ORGaNICs overcomes these limitations, with comparable capabilities to LSTMs and GRUs, with built-in recurrent normalization (motivated by neurobiology) that, as we show, is sufficient to guarantee dynamical stability. Moreover, we note also that normalization is an essential architectural motif in modern ML, and that divisive normalization has been shown to generalize batch and layer normalization, as shown in the work by Ren, M., et. al., (2016). As such, the ability to integrate normalization in the dynamics of an RNN, and to analytically demonstrate the impact of normalization through the derivation of an interpretable normative principle (Lyapunov function) is far from esoteric and very much needed in the theory of deep learning. Finally, our work establishes an important precedent for how the incorporation of neurobiological principles can drive advances in ML.
>
> - “The theoretical … end of section 5”. **Answer:** Considering the cases of $\mathbf{W}_r = \mathbf{I}$ and the two-dimensional model, we explore two different limits of arbitrary $\mathbf{W}_r$. The first limit involves relaxing constraints on all parameters in a high-dimensional model while assuming $\mathbf{W}_r = \mathbf{I}$. The second limit involves relaxing constraints on all parameters, including $w_r$, for a 2D model. These two limits provide a foundation for understanding the empirical results regarding stability in models with arbitrary recurrence, which is intractable with the approach used in the paper. There is also a rich history of studying neural mean field theories (Wilson, H. R., & Cowan, J. D. (1972); Kraynyukova, N., & Tchumatchenko, T. (2018)) which yield two-dimensional E-I circuits used for modeling the cortex. The conjecture is crucially important because we want to learn an arbitrary recurrent weight matrix during ML and neuroscience tasks. We have empirical evidence in support of this conjecture (see Fig.4,5 and Fig.1,2 of the attached pdf). We also use this conjecture in the paper itself (for the static input task).
>
> - “The experiments … SSN comparison)”. **Answer:** We thank the reviewer for the suggestions regarding additional tasks. To satisfy the reviewer, in a subsequent version of the paper, we will include ORGaNICs’ performance on the parity task for comparison. However, we stress that the primary focus of this paper is to demonstrate the stability, trainability, and interpretability of ORGaNICs compared to other RNNs. There was no hyperparameter tuning for any of the tasks – ORGaNICs simply outperforms SSNs on the first attempt at the task we considered.
>
> - “The abstract claims… to be true”. **Answer:** By that statement, we mean to convey that ORGaNICs mitigates the problem of exploding gradients: through its inherent stability; and vanishing gradients: by processing information across various timescales while maintaining stability, resulting in a blend of lossy and non-lossy neurons. This leads to a trainable RNN circuit, with a performance comparable to LSTMs without the need for specialized techniques like gradient clipping/scaling. We have revised this sentence for better clarity.
>
> - “In this respect, ...  training performance”. **Answer:** Like SSN, ORGaNICs is a neurodynamical model originally designed to explain cortical activity. The comparison with SSN is simply meant to demonstrate that just like SSN, ORGaNICs can be successfully trained on a static MNIST task. So we are not trying to make the point that ORGaNICs performs better due to its stability guarantees, but rather that ORGaNICs can be trained by naive BPTT and perform comparably to SSN trained by the non-trivial method of dynamics-neutral growth (DNG). The result of the comparison is that ORGaNICs performs slightly better (despite no hyperparameter tuning) while being much easier to train than SSN. If we were to speculate on the performance gap between ORGaNICs and SSN, our best guess would be that DNG constrains the range of possible models (viz. parameters) thus reducing the expressivity of SSN. It is beyond the scope of this paper to identify why SSN trained by DNG does not do as well, but what is certain is that it is much harder to train SSN than ORGaNICs. The main takeaway is that ORGaNICs can be trained naively because it is stable, and it performs as well or better than alternative neurodynamical models (e.g., SSN) trained by sophisticated techniques.
> Regarding the LSTMs, they were trained using gradient clipping in Arjovsky, M., Shah, A., & Bengio, Y. (2016).
>
> - “On clarity of exposition: … navigate”. **Answer:** We thank the reviewer for the suggestion. We will revise the introduction of the paper to provide a clearer roadmap of the main results of the paper and how they are organized.
>
> **Questions:**
>
> - “The abstract … backpropagation?” **Answer:**   We mean to convey the fact that we train ORGaNICs by naive backpropagation (not simple gradient descent), meaning that 1) we do not use gradient clipping/scaling that is typically adopted during BPTT when training RNNs such as LSTMs; 2) we do not need to resort to specialized techniques such as the sophisticated DNG method used to train SNN, an alternative neurodynamical model. We have resolved this ambiguity by revising the statement.
>
> - “The formatting … text”. **Answer:** We thank the reviewer for the suggestion, we have updated the references using the suggested format.

---

> > ### Comment · Reviewer_FwjG · 2024-08-12
> >
> > Thank you for your rebuttal. I can see the potential impact of this work a bit more clearly now, and will raise my score to a 5, also reflecting a comparative rating taking into account the strength of the field across all my review assignments. Regarding clarity of exposition, I would encourage the authors to do more than just “revis[ing] the introduction of the paper to provide a clearer roadmap of the main results of the paper” -- my concern is that the main text itself is hard going, with the authors often going off on a tangent, and including a plethora of very technical details where the reader would instead like to be given a clear synthesis. Also:
> >
> > > There was no hyperparameter tuning for any of the tasks – ORGaNICs simply outperforms SSNs on the first attempt at the task we considered.
> >
> > Surely the comparison to SSN is meaningless unless you (at least) hyper-tune the SSN results; otherwise, how do we know if you weren't just lucky here? In general though, I think it's a good idea to hyper-optimize even one's own method, as this saves a lot of time downstream to people who would like to compare to your method. For example, if your answer to my concern above is that you, in fact, did not run your own SSN simulations but directly imported the (hyper-tuned) results from a previous SSN paper, then you are relying on SSN authors having done their job, but you are not doing yours.

---

> > > ### Author Response · Authors · 2024-08-13
> > >
> > > We thank the reviewer for their feedback. Regarding the exposition, we will enhance the paper by providing clearer, more intuitive explanations of terms like “Lyapunov diagonally stable,” “Z-matrix,” “M-matrix,” and “indirect method of Lyapunov,” as suggested by reviewer ma5B. However, keeping the technical details is crucial. One of the core innovations of the paper is the technical approach that we developed to prove the stability of the recurrent circuit model (ORGaNICs), connecting the linear stability of a neural circuit model to the dynamic analysis of mechanical systems. As such, we are not merely going off on a tangent, but achieving a rigorous proof of stability for a neural circuit model by linking two disparate fields. This has important implications for the derivation of a Lyapunov function and thus a normative principle that makes ORGaNICs interpretable. We need to be technical enough to make this link precise and for the description of the approach to be correct and reproducible. We note that there are many mathematical papers published in NeurIPS that are richer in theorems and technical details/jargon than this work. We tried to strike a balance to reach a broader audience, so we welcome the reviewer’s comment and will clarify the writing further.
> > >
> > > Regarding the comparison to SSN, the results by Song 2022 were not hyper-tuned either. We used the same learning rate, optimizer configuration, and equal number of layers and number of units in each layer to make the comparison fair. We will state this clearly in the manuscript. Moreover, we will conduct hyperparameter tuning to facilitate future comparisons and ensure our results are robust.
> > >
> > > Nevertheless, we would like to underscore once again that the numerical comparison of ORGaNICs with SSN or LSTMs are not the main point of the paper. Specifically, the core contributions of the paper can be summarized at a lay level as follows:
> > >
> > > - We take an existing neurobiologically plausible RNN model designed to achieve divisive normalization (DN) exactly via recurrent excitation. This model has been shown to recapitulate a broad range of neurophysiological observations, not by design but as a result of imposing a specific circuit function (i.e., DN). Given the resemblance to LSTMs, we hypothesize that this model should also do well on ML tasks and ask what is the impact of imposing DN on the dynamics and trainability of an RNN.
> > >
> > > - We discover empirically that ORGaNICs is unconditionally stable under very mild constraints (this is highly nontrivial). We thus develop mathematical machinery to prove this unconditional stability in a couple of limiting cases (d=2 for all parameter values, and $\mathbf{W}_r = \mathbf{I}$ for generic d and all other parameter values) by expressing ORGaNICs as a mechanical (specifically, gyroscopically stabilized) system – to do this we even need to prove new results on mechanical systems.
> > >
> > > - But what are the implications of this unconditional stability, if any? Stability is crucial to ensure the trainability of RNNs that typically require gradient clipping/scaling. We discover that by virtue of its stability, ORGaNICs can be trained on a standard ML task (sMNIST and psMNIST) by naïve BPTT, without gradient clipping/scaling. In fact, naïve BPTT  works so well that on a first attempt, it does about as well as LSTMs (trained with gradient clipping/scaling and hyper-tuned) – it is not luck, the result is robust with respect to re-initialization of the training. The point then is not simply that ORGaNICs is at least as good as LSTMs, but that imposing DN dynamically makes training effortless. Note that unlike standard RNNs, where normalization is imposed a-posteriori to the output layer, ORGaNICs implements normalization dynamically: this is the first time this idea is tested in ML and analyzed theoretically (as pointed out enthusiastically by reviewer ma5B). Moreover, ORGaNICs trained by naïve BPTT fares well when compared to SSN trained by a specialized technique. The reason we choose SSN is that this model is also neurobiologically inspired and it has also been shown to approximate normalization in certain parameter ranges.
> > >
> > > - Finally, the connection to mechanical systems allows us to derive a Lyapunov function that provides a normative principle for the dynamics of the models (i.e., a means of interpreting why and how the model tends to the stable fixed point).
> > >
> > > We will better summarize the paper’s key contributions in the paper’s introduction and discussion.
> > >
> > > We kindly ask that the reviewer assess our paper on its own merit, and not “reflecting a comparative rating taking into account the strength of the field across all my review assignments”. This request is in line with the NeurIPS FAQ that states “Q: Can I accept or reject all the papers in my stack?” “A: Please accept and reject papers based on their own merits. You do not have to match the conference acceptance rate.”

---

### Official Review · Reviewer_2CUH · 2024-07-18

**Soundness:** 3
**Presentation:** 3
**Contribution:** 3
**Rating:** 6
**Confidence:** 4

**Summary:**

The paper discusses the development and analysis of "Oscillatory Recurrent Gated Neural Integrator Circuits" (ORGaNICs), a biologically plausible model of recurrent neural networks that implements divisive normalization. The authors prove the unconditional local stability of ORGaNICs with an identity recurrent weight matrix using the indirect method of Lyapunov function. They also demonstrate empirical stability for higher-dimensional circuits. The model's performance is evaluated on static and dynamic classification tasks, showing comparable results to LSTMs without specialized training strategies, thanks to its intrinsic stability properties.

**Strengths:**

1. Biological Plausibility: ORGaNICs are designed with a structure and dynamics that are more aligned with biological neural circuits than traditional artificial neural networks.
2. Biological divisive normalization: Implemented divisive normalization using a more bioplausible network.
3. Trainability: ORGaNICs are a biophysically plausible extensions of LSTM and can be directly trained by BPTT.

**Weaknesses:**

1. Generalization: The paper does not extensively discuss how well the results might generalize to more complex or different types of tasks beyond the tested benchmarks.
2. Performance on Benchmarks: The proposed model performance still has a gap compared to machine learning models.
3. Scalability: parameter size scales too fast with model size and there is no obvious way for impovement.

**Questions:**

1. Can this model provide more interpretability in relevant machine learning tasks?
2. Besides normalization and stability, can this E-I model offer more insights into neuroscience compared to machine learning models?
3. Many existing machine learning models have already implemented various forms of divisive like normalization such as layer normalization. What are advantages or innovations of the divisive normalization implemented in this model.
4. In comparison to standard machine learning models, biophysically plausible models typically offer superior interpretability. However, the paper offers limited discussion on this topic.

**Limitations:**

1. In the sequence modeling benchmarks, only 2 tasks are evaluated. It is recommended to include a broader range of sequence modeling tasks to allow for more comprehensive comparisons, such as those where SSM excels in long sequence modeling or where LSTM performs well in formal language tasks like the parity task.

---

> ### Author Rebuttal · Authors · 2024-08-07
>
> **Weaknesses:**
>
> - “Generalization: The paper ... the tested benchmarks”. **Answer:** We expect ORGaNICs to perform well on other ML benchmarks, especially those concerning sequential data. This will be done in a future study, therefore we refrain from making any specific claims about performance on other benchmarks. In this paper, we sought to demonstrate the stability, trainability, and interpretability of ORGaNICs compared to other RNNs, and present proofs and evidence supporting these claims. Specifically, we were able to rigorously prove for the two limiting cases that ORGaNICs is absolutely stable and demonstrated empirically that this stability holds broadly. We achieved good performance on ML tasks using naive BPTT without resorting to gradient clipping/scaling or other techniques required for alternative neurobiological models like SSN, and without hyperparameter tuning. We derived a Lyapunov function for our model offering an interpretable optimization principle explaining the dynamics of ORGaNICs.
>
> - “Performance on Benchmarks: … machine learning models. **Answer:** There was no hyperparameter tuning nor an attempt to get SOTA performance in this paper. The top-performing ML models are designed with properties of low-dissipation and stability to solve the problem of vanishing and exploding gradients. In contrast, ORGaNICs is a biologically plausible circuit designed to implement divisive normalization (DN), a computational principle experimentally found in a wide range of cortical areas (i.e., different brain regions) across different species, to simulate cortical activity. Unlike ML models, stability in ORGaNICs is not engineered but it emerges from its neurobiological plausible design. Despite not being optimized for the tasks considered, ORGaNICs performs competitively with other RNN models. To our knowledge, no other neurodynamical biologically plausible model achieves this.
>
> - “Scalability: parameter size scales … way for impovement”. **Answer:** In future work, we will design the gains $\mathbf{b}$ and $\mathbf{b_0}$ to be dependent locally on $\mathbf{y}$ and $\mathbf{a}$, This will significantly reduce the number of parameters compared to the current model where the matrices $\mathbf{W_{by}}$, $ \mathbf{W_{ba}}$, $ \mathbf{W_{b_0 y}}$ and $ \mathbf{W_{b_0 a}}$ are $n \times n$.
>
> **Questions:**
>
> - “Can this model provide … machine learning models?” **Answer:** We draw a direct connection between ORGaNICs and systems of coupled damped harmonic oscillators, which have been studied in mechanics and control theory for decades. This connection allows us to derive an interpretable energy function for a high-dimensional ORGaNICs circuit (Eq. 128), providing a normative principle of what the circuit aims to accomplish (see Eq. 13 and the subsequent paragraph). For a relevant ML task, having analytical expressions for the energy function (which is minimized by the dynamics of ORGaNICs) allows us to quantify the relative contributions of the individual neurons in the trained model, offering more interpretability than other RNN architectures. For instance, Eq. 128 reveals that the ratio $\tau_y / \tau_a$ of a neuron pair ('y' and its corresponding 'a') indicates the "weight" a neuron assigns to normalization relative to aligning its responses to the input. This provides a clear functional role for neurons in the trained model. Furthermore, since ORGaNICs is biologically plausible, we understand how the different terms of the dynamical system may be computed biologically within a neural circuit (Heeger & Mackey, 2019). This bridges the gap between theoretical models and biological implementation, providing a framework to test hypotheses about neural computation in real biological systems.
>
>
> - “Many existing … implemented in this model”. **Answer:** Divisive normalization (DN) was introduced as a model of the steady-state response of neurons, functioning as a static nonlinearity similar to batch and layer normalization. The DN equation has also been shown to generalize batch and layer normalization, as detailed in the work by Ren, M., Liao, R., Urtasun, R., Sinz, F. H., & Zemel, R. S. (2016). However, in the brain, there are no static nonlinearities; it has thus been proposed that DN is achieved via a recurrent circuit. ORGaNICs is such a recurrent circuit designed so that the responses of neurons at steady state follow the DN equation. While batch and layer normalization are ad hoc implementations that do not affect the dynamics (they are applied to the output layer), ORGaNICs implements DN dynamically. Additionally, whereas batch and layer normalization do not inherently affect the stability of the model (because they do not influence the dynamics), our paper demonstrates that a model implementing DN naturally exhibits stability, which is greatly advantageous for trainability. This stability, derived from the dynamic implementation of DN, sets ORGaNICs apart by providing both output normalization and model robustness.
>
> - “In comparison… limited discussion on this topic”. **Answer:** We thank the reviewer for this suggestion. We will add a paragraph based on our answer to the first question.
>
> **Limitations:**
>
> - “In the sequence modeling …  the parity task”. **Answer:**  We thank the reviewer for the suggestions regarding additional tasks. To satisfy the reviewer, in a subsequent version of the paper, we will include ORGaNICs’ performance on the parity task. However, we stress that the primary focus of this paper is to demonstrate the stability, trainability, and interpretability of ORGaNICs compared to other RNNs, which we have achieved through a thorough analytical and numerical analysis, and not to provide an extensive benchmark of the model on ML tasks. A more comprehensive test of ORGaNICs on additional long sequence modeling tasks, along with hyperparameter optimization, will be pursued in future work.

---

### Official Review · Reviewer_NGE1 · 2024-07-23

**Soundness:** 3
**Presentation:** 3
**Contribution:** 3
**Rating:** 6
**Confidence:** 2

**Summary:**

Recurrent neural networks are widely applied in both solving machine learning tasks and modeling neural recordings.  However, they can be difficult to train due to exploding/vanishing gradients and other sources of instability, often requiring hyperparameter optimization.  In this paper, the authors argue that the ORGaNICs model is capable of solving sequential tasks without the need for hyperprameter optimization due to the conjectured property of ``unconditional stability,'' which guarantees the existence of a stable fixed point in the dynamical system for any model parameters.  The biological plausibility of the ORGaNICs dynamics also suggests that a similar dynamical system could guarantee stability in biological neural circuits for processing sensory information and generating motor sequences.

The stability property is proven rigorously in the case where the recurrent weights are equal to the identity and in the case where only a single recurrent and inhibitory unit are included in the model.  The authors provide new insights into the behavior of ORGaNICs by providing a Lyapunov function for the dynamics near the fixed point in the case where $\mathbf{W}_r = \mathbf{I}$.  An algorithm for quickly finding a stable fixed point of the ORGaNICs dynamical system in general is provided.  This algorithm is used to probe empirically for the fixed points of the ORGaNICs dynamics with general recurrent weights $\mathbf{W}_r$.

The authors test their prediction empirically in numerical experiments where they train an ORGaNICs network on two tasks: Static and sequential MNIST classification.  In both cases, they show that ORGaNICs obtains competitive performance without hyperparameter optimization.

**Strengths:**

This paper includes many interesting insights and clever mathematical arguments.  The proof of unconditional stability through the indirect method of Lyapunov and by relating the problem to a second-order dynamical system is mathematically sound and provides interesting insights through the analogy to a damped harmonic oscillator.  The finding that ORGaNICs achieves competitive performance without the need for hyperparameter optimization  on the static and sequential MNIST tasks is also very promising.

**Weaknesses:**

The main theorem makes the very restrictive assumption that the recurrent weights $\mathbf{W}_r = \mathbf{I}$, such that they exactly cancel the intrinsic decay rate of the recurrent neurons.  While there are neural circuits where all recurrent interactions take place through inhibitory interneurons (e.g. cerebellar granule cells interacting through inhibitory Golgi cells), the intrinsic decay must also be taken into account.  If the assumption can be relaxed to say that $\mathbf{W}_r = \alpha \mathbf{I}$ for some $\alpha \in [0, 1]$ then I would be inclined to raise my score.

Also, the argument for unconditional stability of the ORGaNICs model is not well-supported in the paper's current form.  Beyond the case $\mathbf{W}_r = \mathbf{I}$ and the two-dimensional case, there this is tested only for the MNIST task with constant inputs.  In these tests, it also appears that the fixed-point-finding algorithm is used to quickly discover the stable fixed point.  However, it would be stronger to show that the dynamics naturally converge to the fixed point in a reasonable amount of time when started from randomly generated initial configuration.  In fact, it is possible that the dynamics naturally lead the system to a stable limit cycle and never converge to the stable fixed point, even if it is guaranteed to exist.  We ask the authors to clarify the methods used to locate the fixed point of the ORGaNICs network in their experiments and show that the system's dynamics tend toward this fixed point and not another limit cycle in a reasonable amount of time.  The numerical training also enforces a maximum singular value of $\mathbf{W}_r$, which is a strong condition required for stability.  ``Towards Unconditional Stability...'' would be a more appropriate title given the strong constraints required for stability guarantees in both the analytical proofs and numerical tests.

For the sequential MNIST task, the authors claim that they are able to achieve LSTM-like performance without hyperparameter optimization or gradient clipping/scaling.  During these experiments, the neural time constants are treated as parameters that can be learned.  This result is different than showing that for any fixed values of the time constants, ORGaNICs can be trained without issue.  The authors should verify that fixing the time constants and then training also leads stable training and no need for gradient clipping.  This is important to maintain a claim of biological plausibility where time constants are intrinsic properties of the neurons and not trained.

The static and sequential MNIST tasks also represent a rather limited scope in which to test for unconditional stability in practice.  It's possible (though I agree, unlikely) that for other more unwieldy tasks unconditional stability will not hold.  Either a proof of unconditional stability for general recurrent weight matrices or a wider range of numerical tests are necessary to make the case for unconditional stability of the ORGaNICs dynamics.

**Questions:**

What hyperparameters are present in the ORGaNICs model as trained on the static and sequential MNIST tasks?  While the time constants are learned, how is the learning rate for naive BPTT selected?  Would this hyperparameter plausibly need to be optimized to achieve performance competitive with a well-tuned LSTM?

In the verification of the iterative fixed-point finding algorithm, how is the ``true'' fixed point determined?  By running the dynamics from a random starting point to the fixed point, or by running the dynamics from an initial estimate obtained using the iterative fixed point finding algorithm?

**Limitations:**

Even for static inputs, there are scenarios in which a biological circuit may want to produce a non-static output.  For example, in central pattern generators of motor sequences. If indeed ORGaNICs always converges to a stable fixed point, this may represent too constrained of a circuit to explain many interesting neural phenomena.

---

> ### Author Rebuttal · Authors · 2024-08-07
>
> **Weaknesses:**
> - “The main … raise my score”. **Answer:** Our proof for the stability of the high-dimensional model (using the indirect method of Lyapunov) relies on the existence of analytical expressions for the fixed point. Such an expression exists when $\mathbf{W}_r=\mathbf{I}$, but does not exist when $\mathbf{W}_r \neq \mathbf{I}$. However, we were able to prove unconditional stability for the 2-dimensional model (see Fig.1, Theorem.5.1 and Theorem.5.2), for any positive value of $\alpha (w_r) \in (0,\infty)$ and any combination of choices for other parameters. Building on these two limiting cases, for which a rigorous proof of stability is tractable (i.e., for any $w_r$ in mean field, when d=2, and for $\mathbf{W}_r=\mathbf{I}$ when d>2, with no restrictions on other parameters), we conjecture at the end of Section 5 that the stability property holds in higher dimensions for systems where the largest singular value of $\mathbf{W}_r$​ is less than 1 (therefore covering the case when $\alpha \in [0, 1]$). We provide empirical evidence in support of this conjecture in Fig. 4 and Fig. 5. To satisfy the reviewer’s question directly, we simulated 10,000 random networks with  $\mathbf{W}_r=\alpha \mathbf{I}$ with $\alpha \in [0,1]$ and plot the distribution of the maximum real part of eigenvalues, all of which are found to be less than 0 (Fig.1 of pdf), indicating stability. We do not exclude that a statistical approach could be used to prove the stability of a general network “in expectation”, but 1) it would fall short of a rigorous proof of stability like ours; 2) it is sufficiently ambitious that it should be the subject of a subsequent paper.
> - “Also, the  … amount of time”. **Answer:** We would like to point out to the reviewer an additional experiment at the end of Section 5 where we mention empirical evidence of stability for a more general case of the recurrent matrix. where the maximum singular value of $\mathbf{W}_r$ is constrained to 1. This conjecture is supported by empirical evidence showing consistent stability, as ORGaNICs initialized with random parameters and inputs under these constraints have exhibited stability in 100\% of trials Fig.4.” In Fig. 4 the “true”  fixed point was found by simulating the network from a zero initial condition. In all of the trials where the conditions in the conjecture were met, the dynamics always converged to the stable fixed point found by the iterative scheme. To show that in all simulations there is no limit cycle and that we can start from a random initial condition, we re-performed this analysis (10,000 networks), starting from random initial conditions \in [0, 1] and fixed $\mathbf{W}_r=\alpha \mathbf{I}$ for some $\alpha \in [0,1]$. We find that the system is stable (Fig.1 of pdf) and the simulations always converge to the same fixed point as found by the iterative scheme. So, while we cannot prove the conjecture by the indirect method of Lyapunov, numerical evidence suggests that the conjecture is true almost surely.
> - “For the … and not trained”. **Answer:** This is an excellent point. We chose the time constants to be learnable to demonstrate that ORGaNICs is competitive with LSTMs on sequential MNIST tasks without hyperparameter optimization and without using ad hoc techniques for training, such as gradient clipping/scaling. As per the referee's suggestion, we have now conducted additional experiments where ORGaNICs is trained with fixed time constants (Table 1 of pdf). Our findings indicate that even with fixed time constants, ORGaNICs maintain stable training without the need for gradient clipping/scaling.
>
> - “The static and … ORGaNICs dynamics”. **Answer:** We have provided empirical evidence for unconditional stability for a general recurrent weight matrix, particularly when the maximum singular value of $\mathbf{W}_r$ ​ is less than 1 (Section 5, Fig. 4, and Fig. 5). We prove unconditional stability, using a non-trivial approach, in two specific cases of $\mathbf{W}_r$ ​: first, when $\mathbf{W}_r = \mathbf{I}$, and second, for the 2D model. While a proof for unconditional stability for a general $\mathbf{W}_r$​ would be most desirable, it is currently beyond reach as detailed in our response above. Therefore, we must rely on empirical evidence for the stability of a general $\mathbf{W}_r$. In a subsequent version of the paper, we will provide results for the formal language parity task to further corroborate our point.
>
> **Questions:**
>
> - “What hyperparameters … tasks?” **Answer:** Learning rate = 0.001 (static MNIST), 0.01 (sMNIST); Batch size = 256, Weight decay = $10^{-5}$ for both. This information has been added in the Appendix.
>
> - “While the … LSTM?” **Answer:** We do not have a specific scheme for selecting an optimal learning rate. Since the point of this paper is to showcase greater stability, trainability, and interpretability compared to other RNNs, and not to achieve SOTA on the benchmarks, we did not tune the hyperparameters. This will be explored in future work and should lead to better performance on ML benchmarks.
>
> - “In the verification … algorithm?” **Answer:** The “true” fixed point is found by simulating the network from a zero initial condition. However, in response to the reviewer, we have verified that for a random initialization as well, the simulation converges to the same fixed point.
>
> **Limitations:**
>
> - “Even for … phenomena”. **Answer:** This class of models has already been shown to produce non-static oscillatory activity as well as to reproduce a wide range of neural phenomena, including sustained activity, sequential activity, motor preparation and motor control (Heeger, D. J., & Mackey, W. E. (2019)); simulate the dynamics of V1 activity (Heeger, D. J., & Zemlianova, K. O. (2020), S. Rawat, D.J. Heeger, and S. Martiniani. Cosyne Abstracts 2024); predict the emergence of communication subspaces in interareal communication (S. Rawat, D.J. Heeger, and S. Martiniani. Cosyne Abstracts 2023).

---

> > ### Comment · Reviewer_NGE1 · 2024-08-09
> > **Raising Score**
> >
> > The authors have addressed most of my concerns with added experiments.  I believe that this is a significant contribution even without a proof of unconditional stability in the most general case.  I've raised my score to a 6.

---

> > > ### Author Response · Authors · 2024-08-11
> > > **Comment**
> > >
> > > We thank the reviewer for their feedback and for increasing the score.

---

### Author Rebuttal · Authors · 2024-08-07

We thank all the reviewers for dedicating their time and providing valuable feedback on our submission. Your insights and comments are greatly appreciated and will help improve the quality of our work.

In response to the reviewers' comments, we have conducted additional experiments and analyses. Please find the following new results in the attached PDF:
- $\textbf{Table 1:}$ Based on the suggestion from Reviewer 1, we demonstrate that ORGaNICs can be trained with fixed time constants on both sequential and permuted MNIST datasets without requiring gradient clipping or scaling.
- $\textbf{Figure 1:}$ In response to Reviewer 1's inquiry about ORGaNICs' stability when the recurrent weight matrix defined as $\mathbf{W}_r=\alpha \mathbf{I}$ for $\alpha \in [0,1]$, we simulated 10,000 networks with random weights, inputs and values of $\alpha$, and with random initial values of $\mathbf{y}$ and $\mathbf{a}$. We found that all simulations converge to the fixed point identified by the iterative algorithm. These fixed points are stable, as the largest real part of the eigenvalues, for all the networks, is strictly negative.
- $\textbf{Figure 2:}$ Based on the insight from Reviewer 5 that ORGaNICs, when trained on static input classification task (Section 6.1), acts as a deep equilibrium model (DEQ) (Bai, S., Kolter, J. Z., & Koltun, V. (2019)). DEQs are known to be prone to instabilities during training. We demonstrate that since ORGaNICs is intrinsically stable for all parameters and input in the conditions assumed in Section 6.1, it stays stable throughout the training. Specifically, the fixed points remain attractive as all of the eigenvalues, across all the test samples, have negative real parts throughout the training.

---

### Decision · Program_Chairs · 2024-09-25

**Decision:**

Accept (poster)

**Comment:**

This paper proves the unconditional stability of the ORGaNICs model, a biologically plausible RNN that implements divisive normalization. The authors provide rigorous mathematical analysis, particularly under specific conditions like the identity recurrent weight matrix, and demonstrate empirical stability across various tasks, including static and sequential classification benchmarks. Reviewers raised concerns about the broader applicability of the stability proof and the reliance on empirical evidence for more general cases. However, the authors addressed these concerns with additional experiments. While all reviewers voted for acceptance, the paper could benefit from clearer exposition and more comprehensive comparisons.